# Adaptive Gradient-Based Meta-Learning Methods

**Mikhail Khodak**
Carnegie Mellon University
`khodak@cmu.edu`

**Maria-Florina Balcan**
Carnegie Mellon University
`ninamf@cs.cmu.edu`

**Ameet Talwalkar**
Carnegie Mellon University
& Determined AI
`talwalkar@cmu.edu`

## Abstract

We build a theoretical framework for designing and understanding practical meta-learning methods that integrates sophisticated formalizations of task-similarity with the extensive literature on online convex optimization and sequential prediction algorithms. Our approach enables the task-similarity to be learned adaptively, provides sharper transfer-risk bounds in the setting of statistical learning-to-learn, and leads to straightforward derivations of average-case regret bounds for efficient algorithms in settings where the task-environment changes dynamically or the tasks share a certain geometric structure. We use our theory to modify several popular meta-learning algorithms and improve their meta-test-time performance on standard problems in few-shot learning and federated learning.

## 1   Introduction

*Meta-learning*, or *learning-to-learn* (LTL) [52], has recently re-emerged as an important direction for developing algorithms for multi-task learning, dynamic environments, and federated settings. By using the data of numerous training tasks, meta-learning methods seek to perform well on new, potentially related test tasks without using many samples. Successful modern approaches have also focused on exploiting the capabilities of deep neural networks, whether by learning multi-task embeddings passed to simple classifiers [51] or by neural control of optimization algorithms [46].

Because of its simplicity and flexibility, a common approach is *parameter-transfer*, where all tasks use the same class of $\Theta$-parameterized functions $f_\theta : \mathcal{X} \mapsto \mathcal{Y}$; often a shared model $\phi \in \Theta$ is learned that is used to train within-task models. In *gradient-based meta-learning* (GBML) [23], $\phi$ is a meta-initialization for a gradient descent method over samples from a new task. GBML is used in a variety of LTL domains such as vision [38, 44, 35], federated learning [16], and robotics [20, 1]. Its simplicity also raises many practical and theoretical questions about the task-relations it can exploit and the settings in which it can succeed. Addressing these issues has naturally led several authors to online convex optimization (OCO) [55], either directly [24, 34] or from online-to-batch conversion [34, 19]. These efforts study how to find a meta-initialization, either by proving algorithmic learnability [24] or giving meta-test-time performance guarantees [34, 19].

However, this recent line of work has so far considered a very restricted, if natural, notion of task-similarity – closeness to a single fixed point in the parameter space. We introduce a new theoretical framework, **Average Regret-Upper-Bound Analysis (ARUBA)**, that enables the derivation of meta-learning algorithms that can provably take advantage of much more sophisticated structure. ARUBA treats meta-learning as the online learning of a sequence of losses that each upper bounds the regret on a single task. These bounds often have convenient functional forms that are (a) sufficiently nice, so that we can draw upon the existing OCO literature, and (b) strongly dependent on both the task-data and the meta-initialization, thus encoding task-similarity in a mathematically accessible way. Using ARUBA we introduce or dramatically improve upon GBML results in the following settings:

- **Adapting to the Task-Similarity:** A major drawback of previous work is a reliance on knowing the task-similarity beforehand to set the learning rate [24] or regularization [19], or the use of a sub-optimal guess-and-tune approach using the doubling trick [34]. ARUBA yields a simple gradient-based algorithm that eliminates the need to guess the similarity by learning it on-the-fly.

- **Adapting to Dynamic Environments:** While previous theoretical work has largely considered a fixed initialization [24, 34], in many practical applications of GBML the optimal initialization varies over time due to a changing environment [1]. We show how ARUBA reduces the problem of meta-learning in dynamic environments to a dynamic regret-minimization problem, for which there exists a vast array of online algorithms with provable guarantees that can be directly applied.

- **Adapting to the Inter-Task Geometry:** A recurring notion in LTL is that certain model weights, such as feature extractors, are shared, whereas others, such as classification layers, vary between tasks. By only learning a fixed initialization we must re-learn this structure on every task. Using ARUBA we provide a method that adapts to this structure and determines which directions in $\Theta$ need to be updated by learning a Mahalanobis-norm regularizer for online mirror descent (OMD). We show how a variant of this can be used to meta-learn a per-coordinate learning-rate for certain GBML methods, such as MAML [23] and Reptile [44], as well as for FedAvg, a popular federated learning algorithm [41]. This leads to improved meta-test-time performance on few-shot learning and a simple, tuning-free approach to effectively add user-personalization to FedAvg.

- **Statistical Learning-to-Learn:** ARUBA allows us to leverage powerful results in online-to-batch conversion [54, 33] to derive new bounds on the transfer risk when using GBML for statistical LTL [8], including fast rates in the number of tasks when the task-similarity is known and high-probability guarantees for a class of losses that includes linear regression. This improves upon the guarantees of Khodak et al. [34] and Denevi et al. [19] for similar or identical GBML methods.

## 1.1 Related Work

**Theoretical LTL:** The statistical analysis of LTL was formalized by Baxter [8]. Several works have built upon this theory for modern LTL, such as via a PAC-Bayesian perspective [3] or by learning the kernel for the ridge regression [18]. However, much effort has also been devoted to the online setting, often through the framework of lifelong learning [45, 5, 2]. Alquier et al. [2] consider a many-task notion of regret similar to the one we study in order to learn a shared data representation, although our algorithms are much more practical. Recently, Bullins et al. [11] developed an efficient online approach to learning a linear data embedding, but such a setting is distinct from GBML and more closely related to popular shared-representation methods such as ProtoNets [51]. Nevertheless, our approach does strongly rely on online learning through the study of data-dependent regret-upper-bounds, which has a long history of use in deriving adaptive single-task methods [40, 21]; however, in meta-learning there is typically not enough data to adapt to without considering multi-task data. Analyzing regret-upper-bounds was done implicitly by Khodak et al. [34], but their approach is largely restricted to using Follow-the-Leader (FTL) as the meta-algorithm. Similarly, Finn et al. [24] use FTL to show learnability of the MAML meta-initialization. In contrast, the ARUBA framework can handle general classes of meta-algorithms, which leads not only to new and improved results in static, dynamic, and statistical settings but also to significantly more practical LTL methods.

**GBML:** GBML stems from the Model-Agnostic Meta-Learning (MAML) algorithm [23] and has been widely used in practice [1, 44, 31]. An expressivity result was shown for MAML by Finn and Levine [22], proving that the meta-learner can approximate any permutation-invariant learner given enough data and a specific neural architecture. Under strong-convexity and smoothness assumptions and using a fixed learning rate, Finn et al. [24] show that the MAML meta-initialization is learnable, albeit via an impractical FTL method. In contrast to these efforts, Khodak et al. [34] and Denevi et al. [19] focus on providing finite-sample meta-test-time performance guarantees in the convex setting, the former for the SGD-based Reptile algorithm of Nichol et al. [44] and the latter for a regularized variant. Our work improves upon these analyses by considering the case when the learning rate, a proxy for the task-similarity, is not known beforehand as in Finn et al. [24] and Denevi et al. [19] but must be learned online; Khodak et al. [34] do consider an unknown task-similarity but use a doubling-trick-based approach that considers the absolute deviation of the task-parameters from the meta-initialization and is thus average-case suboptimal and sensitive to outliers. Furthermore, ARUBA can handle more sophisticated and dynamic notions of task-similarity and in certain settings can provide better statistical guarantees than those of Khodak et al. [34] and Denevi et al. [19].

## 2 Average Regret-Upper-Bound Analysis

Our main contribution is ARUBA, a framework for analyzing the learning of $\mathcal{X}$-parameterized learning algorithms via reduction to the online learning of a sequence of functions $\mathbf{U}_t : \mathcal{X} \mapsto \mathbb{R}$ upper-bounding their regret on task $t$. We consider a meta-learner facing a sequence of online learning tasks $t = 1, \ldots, T$, each with $m_t$ loss functions $\ell_{t,i} : \Theta \mapsto \mathbb{R}$ over action-space $\Theta \subset \mathbb{R}^d$. The learner has access to a set of learning algorithms parameterized by $x \in \mathcal{X}$ that can be used to determine the action $\theta_{t,i} \in \Theta$ on each round $i \in [m_t]$ of task $t$. Thus on each task $t$ the meta-learner chooses $x_t \in \mathcal{X}$, runs the corresponding algorithm, and suffers regret $\mathbf{R}_t(x_t) = \sum_{i=1}^{m_t} \ell_{t,i}(\theta_{t,i}) - \min_\theta \sum_{i=1}^{m_t} \ell_{t,i}(\theta)$. We propose to analyze the meta-learner's performance by studying the online learning of a sequence of regret-upper-bounds $\mathbf{U}_t(x_t) \geq \mathbf{R}_t(x_t)$, specifically by bounding the **average regret-upper-bound** $\bar{\mathbf{U}}_T = \frac{1}{T} \sum_{t=1}^T \mathbf{U}_t(x_t)$. The following two observations highlight why we care about this quantity:

1. **Generality:** Many algorithms of interest in meta-learning have regret guarantees $\mathbf{U}_t(x)$ with nice, e.g. smooth and convex, functional forms that depend strongly on both their parameterizations $x \in \mathcal{X}$ and the task-data. This data-dependence lets us adaptively set the parameterization $x_t \in \mathcal{X}$.

2. **Consequences:** By definition of $\mathbf{U}_t$ we have that $\bar{\mathbf{U}}_T$ bounds the **task-averaged regret (TAR)** $\bar{\mathbf{R}}_T = \frac{1}{T} \sum_{t=1}^T \mathbf{R}_t(x_t)$ [34]. Thus if the average regret-upper-bound is small then the meta-learner will perform well on-average across tasks. In Section 5 we further show that a low average regret-upper-bound will also lead to strong statistical guarantees in the batch setting.

ARUBA's applicability depends only on finding a low-regret algorithm over the functions $\mathbf{U}_t$; then by observation 2 we get a task-averaged regret bound where the first term vanishes as $T \to \infty$ while by observation 1 the second term can be made small due to the data-dependent task-similarity:

$$\bar{\mathbf{R}}_T \leq \bar{\mathbf{U}}_T \leq o_T(1) + \min_x \frac{1}{T} \sum_{t=1}^T \mathbf{U}_t(x)$$

**The Case of Online Gradient Descent:** Suppose the meta-learner uses online gradient descent (OGD) as the within-task learning algorithm, as is done by Reptile [44]. OGD can be parameterized by an initialization $\phi \in \Theta$ and a learning rate $\eta > 0$, so that $\mathcal{X} = \{(\phi, \eta) : \phi \in \Theta, \eta > 0\}$. Using the notation $v_{a:b} = \sum_{i=a}^b v_i$ and $\nabla_{t,j} = \nabla \ell_{t,j}(\theta_{t,j})$, at each round $i$ of task $t$ OGD plays $\theta_{t,i} = \arg\min_{\theta \in \Theta} \frac{1}{2} \|\theta - \phi\|_2^2 + \eta \langle \nabla_{t,1:i-1}, \theta \rangle$. The regret of this procedure when run on $m$ convex $G$-Lipschitz losses has a well-known upper-bound [48, Theorem 2.11]

$$\mathbf{U}_t(x) = \mathbf{U}_t(\phi, \eta) = \frac{1}{2\eta} \|\theta_t^* - \phi\|_2^2 + \eta G^2 m \geq \sum_{i=1}^m \ell_{t,i}(\theta_t) - \ell_{t,i}(\theta_t^*) = \mathbf{R}_t(x) \qquad (1)$$

which is convex in the learning rate $\eta$ and the initialization $\phi$. Note the strong data dependence via $\theta_t^* \in \arg\min_\theta \sum_{i=1}^{m_t} \ell_{t,i}(\theta)$, the optimal action in hindsight. To apply ARUBA, first note that if $\bar{\theta}^* = \frac{1}{T} \theta_{1:T}^*$ is the mean of the optimal actions $\theta_t^*$ on each task and $V^2 = \frac{1}{T} \sum_{t=1}^T \|\theta_t^* - \bar{\theta}^*\|_2^2$ is their empirical variance, then $\min_{\phi, \eta} \frac{1}{T} \sum_{t=1}^T \mathbf{U}_t(\phi, \eta) = \mathcal{O}(GV\sqrt{m})$. Thus by running a low-regret algorithm on the regret-upper-bounds $\mathbf{U}_t$ the meta-learner will suffer task-averaged regret at most $o_T(1) + \mathcal{O}(GV\sqrt{m})$, which can be much better than the single-task regret $\mathcal{O}(GD\sqrt{m})$, where $D$ is the $\ell_2$-radius of $\Theta$, if $V \ll D$, i.e. if the optimal actions $\theta_t^*$ are close together. See Theorem 3.2 for the result yielded by ARUBA in this simple setting.

## 3 Adapting to Similar Tasks and Dynamic Environments

We now demonstrate the effectiveness of ARUBA for analyzing GBML by using it to prove a general bound for a class of algorithms that can adapt to both *task-similarity*, i.e. when the optimal actions $\theta_t^*$ for each task are close to some good initialization, and to *changing environments*, i.e. when this initialization changes over time. The task-similarity will be measured using the **Bregman divergence** $\mathcal{B}_R(\theta \| \phi) = R(\theta) - R(\phi) - \langle \nabla R(\phi), \theta - \phi \rangle$ of a 1-strongly-convex function $R : \Theta \mapsto \mathbb{R}$ [10], a generalized notion of distance. Note that for $R(\cdot) = \frac{1}{2} \| \cdot \|_2^2$ we have $\mathcal{B}_R(\theta \| \phi) = \frac{1}{2} \|\theta - \phi\|_2^2$. A changing environment will be studied by analyzing **dynamic regret**, which for a sequence of actions $\{\phi_t\}_t \subset \Theta$ taken by some online algorithm over a sequence of loss functions $\{f_t : \Theta \mapsto \mathbb{R}\}_t$ is defined w.r.t. a reference sequence $\Psi = \{\psi_t\}_t \subset \Theta$ as $\mathbf{R}_T(\Psi) = \sum_{t=1}^T f_t(\phi_t) - f_t(\psi_t)$. Dynamic regret measures the performance of an online algorithm taking actions $\phi_t$ relative to a potentially time-varying comparator taking actions $\psi_t$. Note that when we fix $\psi_t = \psi^* \in \arg\min_{\psi \in \Theta} \sum_{t=1}^T f_t(\psi)$ we recover the standard **static regret**, in which the comparator always uses the same action.

**Algorithm 1:** Generic online algorithm for gradient-based parameter-transfer meta-learning. To run OGD within-task set $R(\cdot) = \frac{1}{2}\|\cdot\|_2^2$. To run FTRL within-task substitute $\ell_{t,j}(\theta)$ for $\langle \nabla_{t,j}, \theta \rangle$.

---

Set meta-initialization $\phi_1 \in \Theta$ and learning rate $\eta_1 > 0$.
**for** task $t \in [T]$ **do**
    **for** round $i \in [m_t]$ **do**
        $\theta_{t,i} \leftarrow \arg\min_{\theta \in \Theta} \mathcal{B}_R(\theta||\phi_t) + \eta_t \langle \nabla_{t,1:i-1}, \theta \rangle$   `// online mirror descent step`
        Suffer loss $\ell_{t,i}(\theta_{t,i})$
    Update $\phi_{t+1}, \eta_{t+1}$   `// meta-update of OMD initialization and learning rate`

---

Putting these together, we seek to define variants of Algorithm 1 for which as $T \to \infty$ the average regret scales with $V_\Psi$, where $V_\Psi^2 = \frac{1}{T}\sum_{t=1}^{T} \mathcal{B}_R(\theta_t^*||\psi_t)$, without knowing this quantity in advance. Note for fixed $\psi_t = \bar{\theta}^* = \frac{1}{T}\theta_{1:T}^*$ this measures the empirical standard deviation of the optimal task-actions $\theta_t^*$. Thus achieving our goal implies that average performance improves with task-similarity.

On each task $t$ Algorithm 1 runs online mirror descent with regularizer $\frac{1}{\eta_t}\mathcal{B}_R(\cdot||\phi_t)$ for initialization $\phi_t \in \Theta$ and learning rate $\eta_t > 0$. It is well-known that OMD and the related Follow-the-Regularized-Leader (FTRL), for which our results also hold, generalize many important online methods, e.g. OGD and multiplicative weights [26]. For $m_t$ convex losses with mean squared Lipschitz constant $G_t^2$ they also share a convenient, data-dependent regret-upper-bound for any $\theta_t^* \in \Theta$ [48, Theorem 2.15]:

$$\mathbf{R}_t \leq \mathbf{U}_t(\phi_t, \eta_t) = \frac{1}{\eta_t}\mathcal{B}_R(\theta_t^*||\phi_t) + \eta_t G_t^2 m_t \qquad (2)$$

All that remains is to come up with update rules for the meta-initialization $\phi_t \in \Theta$ and the learning rate $\eta_t > 0$ in Algorithm 1 so that the average over $T$ of these upper-bounds $\mathbf{U}_t(\phi_t, \eta_t)$ is small. While this can be viewed as a single online learning problem to determine actions $x_t = (\phi_t, \eta_t) \in \Theta \times (0, \infty)$, it is easier to decouple $\phi$ and $\eta$ by first defining two function sequences $\{f_t^{\text{init}}\}_t$ and $\{f_t^{\text{sim}}\}_t$:

$$f_t^{\text{init}}(\phi) = \mathcal{B}_R(\theta_t^*||\phi)G_t\sqrt{m_t} \qquad\qquad f_t^{\text{sim}}(v) = \left(\frac{\mathcal{B}_R(\theta_t^*||\phi_t)}{v} + v\right)G_t\sqrt{m_t} \qquad (3)$$

We show in Theorem 3.1 that to get an adaptive algorithm it suffices to specify two OCO algorithms, INIT and SIM, such that the actions $\phi_t = \text{INIT}(t)$ achieve good (dynamic) regret over $f_t^{\text{init}}$ and the actions $v_t = \text{SIM}(t)$ achieve low (static) regret over $f_t^{\text{sim}}$; these actions then determine the update rules of $\phi_t$ and $\eta_t = v_t/(G_t\sqrt{m_t})$. We will specialize Theorem 3.1 to derive algorithms that provably adapt to task similarity (Theorem 3.2) and to dynamic environments (Theorem 3.3).

To understand the formulation of $f_t^{\text{init}}$ and $f_t^{\text{sim}}$, first note that $f_t^{\text{sim}}(v) = \mathbf{U}_t(\phi_t, v/(G_t\sqrt{m_t}))$, so the online algorithm SIM over $f_t^{\text{sim}}$ corresponds to an online algorithm over the regret-upper-bounds $\mathbf{U}_t$ when the sequence of initializations $\phi_t$ is chosen adversarially. Once we have shown that SIM is low-regret we can compare its losses $f_t^{\text{sim}}(v_t)$ to those of an arbitrary fixed $v > 0$; this is the first line in the proof of Theorem 3.1 (below). For fixed $v$, each $f_t^{\text{init}}(\phi_t)$ is an affine transformation of $f_t^{\text{sim}}(v)$, so the algorithm INIT with low dynamic regret over $f_t^{\text{init}}$ corresponds to an algorithm with low dynamic regret over the regret-upper-bounds $\mathbf{U}_t$ when $\eta_t = v/(G_t\sqrt{m_t}) \; \forall \; t$. Thus once we have shown a dynamic regret guarantee for INIT we can compare its losses $f_t^{\text{init}}(\phi_t)$ to those of an arbitrary comparator sequence $\{\psi_t\}_t \subset \Theta$; this is the second line in the proof of Theorem 3.1.

**Theorem 3.1.** *Assume $\Theta \subset \mathbb{R}^d$ is convex, each task $t \in [T]$ is a sequence of $m_t$ convex losses $\ell_{t,i} : \Theta \mapsto \mathbb{R}$ with mean squared Lipschitz constant $G_t^2$, and $R : \Theta \mapsto \mathbb{R}$ is 1-strongly-convex.*

- *Let* INIT *be an algorithm whose dynamic regret over functions $\{f_t^{init}\}_t$ w.r.t. any reference sequence $\Psi = \{\psi_t\}_{t=1}^T \subset \Theta$ is upper-bounded by $\mathbf{U}_T^{init}(\Psi)$.*
- *Let* SIM *be an algorithm whose static regret over functions $\{f_t^{sim}\}_t$ w.r.t. any $v > 0$ is upper-bounded by a non-increasing function $\mathbf{U}_T^{sim}(v)$ of $v$.*

*If Algorithm 1 sets $\phi_t = \text{INIT}(t)$ and $\eta_t = \frac{\text{SIM}(t)}{G_t\sqrt{m_t}}$ then for $V_\Psi^2 = \frac{\sum_{t=1}^T \mathcal{B}_R(\theta_t^*||\psi_t)G_t\sqrt{m_t}}{\sum_{t=1}^T G_t\sqrt{m_t}}$ it will achieve average regret*

$$\bar{\mathbf{R}}_T \leq \bar{\mathbf{U}}_T \leq \frac{\mathbf{U}_T^{sim}(V_\Psi)}{T} + \frac{1}{T}\min\left\{\frac{\mathbf{U}_T^{init}(\Psi)}{V_\Psi}, 2\sqrt{\mathbf{U}_T^{init}(\Psi)\sum_{t=1}^T G_t\sqrt{m_t}}\right\} + \frac{2V_\Psi}{T}\sum_{t=1}^T G_t\sqrt{m_t}$$

*Proof.* For $\sigma_t = G_t\sqrt{m_t}$ we have by the regret bound on OMD/FTRL (2) that

$$\bar{\mathbf{U}}_T\,T = \sum_{t=1}^T \left(\frac{\mathcal{B}_R(\theta_t^*||\phi_t)}{v_t} + v_t\right)\sigma_t \le \min_{v>0}\mathbf{U}_T^{\text{sim}}(v) + \sum_{t=1}^T \left(\frac{\mathcal{B}_R(\theta_t^*||\phi_t)}{v} + v\right)\sigma_t$$

$$\le \min_{v>0}\mathbf{U}_T^{\text{sim}}(v) + \frac{\mathbf{U}_T^{\text{init}}(\Psi)}{v} + \sum_{t=1}^T \left(\frac{\mathcal{B}_R(\theta_t^*||\psi_t)}{v} + v\right)\sigma_t$$

$$\le \mathbf{U}_T^{\text{sim}}(V_\Psi) + \min\left\{\frac{\mathbf{U}_T^{\text{init}}(\Psi)}{V_\Psi}, 2\sqrt{\mathbf{U}_T^{\text{init}}(\Psi)\sigma_{1:T}}\right\} + 2V_\Psi\sigma_{1:T}$$

where the last line follows by substituting $v = \max\left\{V_\Psi, \sqrt{\mathbf{U}_T^{\text{init}}(\Psi)/\sigma_{1:T}}\right\}$. $\qquad\square$

**Similar Tasks in Static Environments:**   By Theorem 3.1, if we can specify algorithms INIT and SIM with sublinear regret over $f_t^{\text{init}}$ and $f_t^{\text{sim}}$ (3), respectively, then the average regret will converge to $\mathcal{O}(V_\Psi\sqrt{m})$ as desired. We first show an approach in the case when the optimal actions $\theta_t^*$ are close to a fixed point in $\Theta$, i.e. for fixed $\psi_t = \bar{\theta}^* = \frac{1}{T}\theta_{1:T}^*$. Henceforth we assume the Lipschitz constant $G$ and number of rounds $m$ are the same across tasks; detailed statements are in the supplement.

Note that if $R(\cdot) = \frac{1}{2}\|\cdot\|_2^2$ then $\{f_t^{\text{init}}\}_t$ are quadratic functions, so playing $\phi_{t+1} = \frac{1}{t}\theta_{1:t}^*$ has logarithmic regret [48, Corollary 2.2]. We use a novel strongly convex coupling argument to show that this holds for any such sequence of Bregman divergences, *even for nonconvex* $\mathcal{B}_R(\theta_t^*||\cdot)$. The second sequence $\{f_t^{\text{sim}}\}_t$ is harder because it is not smooth near 0 and not strongly convex if $\theta_t^* = \phi_t$. We study a regularized sequence $\tilde{f}_t^{\text{sim}}(v) = f_t^{\text{sim}}(v) + \varepsilon^2/v$ for $\varepsilon \ge 0$. Assuming a bound of $D^2$ on the Bregman divergence and setting $\varepsilon = 1/\sqrt[4]{T}$, we achieve $\tilde{\mathcal{O}}(\sqrt{T})$ regret on the original sequence by running exponentially-weighted online-optimization (EWOO) [28] on the regularized sequence:

$$v_t = \frac{\int_0^{\sqrt{D^2+\varepsilon^2}} v\exp(-\gamma\sum_{s<t}\tilde{f}_s^{\text{sim}}(v))dv}{\int_0^{\sqrt{D^2+\varepsilon^2}} \exp(-\gamma\sum_{s<t}\tilde{f}_s^{\text{sim}}(v))dv} \qquad\text{for}\qquad \gamma = \frac{2}{DG\sqrt{m}}\min\left\{\frac{\varepsilon^2}{D^2}, 1\right\} \qquad (4)$$

Note that while EWOO is inefficient in high dimensions, we require only single-dimensional integrals. In the supplement we also show that simply setting $v_{t+1}^2 = \varepsilon^2 t + \sum_{s\le t}\mathcal{B}_R(\theta_s^*||\phi_t)$ has only a slightly worse regret of $\tilde{\mathcal{O}}(T^{3/5})$. These guarantees suffice to show the following:

**Theorem 3.2.** *Under the assumptions of Theorem 3.1 and boundedness of $\mathcal{B}_R$ over $\Theta$, if* INIT *plays $\phi_{t+1} = \frac{1}{t}\theta_{1:t}^*$ and* SIM *uses $\varepsilon$-EWOO (4) with $\varepsilon = 1/\sqrt[4]{T}$ then Algorithm 1 achieves average regret*

$$\bar{\mathbf{R}}_T \le \bar{\mathbf{U}}_T = \tilde{\mathcal{O}}\left(\min\left\{\frac{1+\frac{1}{V}}{\sqrt{T}}, \frac{1}{\sqrt[4]{T}}\right\} + V\right)\sqrt{m} \qquad\text{for}\qquad V^2 = \min_{\phi\in\Theta}\frac{1}{T}\sum_{t=1}^T\mathcal{B}_R(\theta_t^*||\phi)$$

Observe that if $V$, the average deviation of $\theta_t^*$, is $\Omega_T(1)$ then the bound becomes $\mathcal{O}(V\sqrt{m})$ at rate $\tilde{\mathcal{O}}(1/\sqrt{T})$, while if $V = o_T(1)$ the bound tends to zero. Theorem 3.1 can be compared to the main result of Khodak et al. [34], who set the learning rate via a doubling trick. We improve upon their result in two aspects. First, their asymptotic regret is $\mathcal{O}(D^*\sqrt{m})$, where $D^*$ is the maximum distance between *any two optimal actions*. Note that $V$ is always at most $D^*$, and indeed may be much smaller in the presence of outliers. Second, our result is more general, as we do not need convex $\mathcal{B}_R(\theta_t^*||\cdot)$.

**Remark 3.1.** *We assume an oracle giving a unique $\theta^* \in \arg\min_{\theta\in\Theta}\sum_{\ell\in S}\ell(\theta)$ for any finite loss sequence $S$, which may be inefficient or undesirable. One can instead use the last or average iterate of within-task OMD/FTRL for the meta-update; in the supplement we show that this incurs an additional $o(\sqrt{m})$ regret term under a quadratic growth assumption that holds in many practical settings [34].*

**Related Tasks in Changing Environments:**   In many settings we have a changing environment and so it is natural to study dynamic regret. This has been widely analyzed by the online learning community [15, 30], often by showing a dynamic regret bound consisting of a sublinear term plus a bound on the variation in the action or function space. Using Theorem 3.1 we can show dynamic guarantees for GBML via reduction to such bounds. We provide an example in the Euclidean geometry using the popular path-length-bound $P_\Psi = \sum_{t=2}^T\|\psi_t - \psi_{t-1}\|_2$ for reference actions $\Psi = \{\psi_t\}_{t=1}^T$ [55]. We use a result showing that OGD with learning rate $\eta \le 1/\beta$ over $\alpha$-strongly-convex, $\beta$-strongly-smooth, and $L$-Lipschitz functions has a bound of $\mathcal{O}(L(1+P_\Psi))$ on its dynamic regret [42, Corollary 1]. Observe that in the case of $R(\cdot) = \frac{1}{2}\|\cdot\|_2^2$ the sequence $f_t^{\text{init}}$ in Theorem 3.1 consists of $DG\sqrt{m}$-Lipschitz quadratic functions. Thus using Theorem 3.1 we achieve the following:

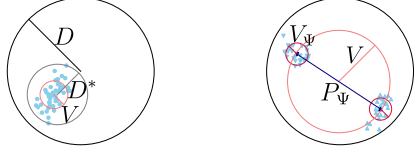

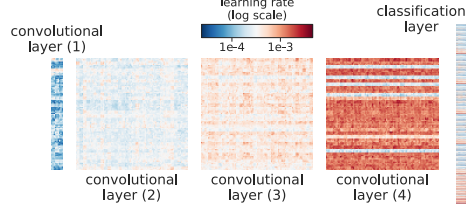

Figure 1: Left - Theorem 3.2 improves upon [34, Theorem 2.1] via its dependence on the average deviation $V$ rather than the maximal deviation $D^*$ of the optimal task-parameters $\theta_t^*$ (light blue). Right - a case where Theorem 3.3 yields a strong task-similarity-based guarantee via a dynamic comparator $\Psi$ despite the deviation $V$ being large.

Figure 2: Learning rate variation across layers of a convolutional net trained on Mini-ImageNet using Algorithm 2. Following intuition outlined in Section 6, shared feature extractors are not updated much if at all compared to higher layers.

**Theorem 3.3.** *Under Theorem 3.1 assumptions, bounded $\Theta$, and $R(\cdot) = \frac{1}{2}\|\cdot\|_2^2$, if INIT is OGD with learning rate $\frac{1}{G\sqrt{m}}$ and SIM uses $\varepsilon$-EWOO (4) with $\varepsilon = 1/\sqrt[4]{T}$ then by using OGD within-task Algorithm 1 will achieve for any fixed comparator sequence $\Psi = \{\psi_t\}_{t\in[T]} \subset \Theta$ the average regret*

$$\bar{\mathbf{R}}_T \le \bar{\mathbf{U}}_T = \tilde{\mathcal{O}}\left(\min\left\{\frac{1 + \frac{1}{V_\Psi}}{\sqrt{T}}, \frac{1}{\sqrt[4]{T}}\right\} + \min\left\{\frac{1 + P_\Psi}{V_\Psi T}, \sqrt{\frac{1 + P_\Psi}{T}}\right\} + V_\Psi\right)\sqrt{m}$$

*for $V_\Psi^2 = \frac{1}{2T}\sum_{t=1}^T \|\theta_t^* - \psi_t\|_2^2$ and $P_\Psi = \sum_{t=2}^T \|\psi_t - \psi_{t-1}\|_2$.*

This bound controls the average regret across tasks using the deviation $V_\Phi$ of the optimal task parameters $\theta_t^*$ from some reference sequence $\Phi$, which is assumed to vary slowly or sparsely so that the path length $P_\Phi$ is small. Figure 1 illustrates when such a guarantee improves over Theorem 3.2. Note also that Theorem 3.3 specifies OGD as the meta-update algorithm INIT, so under the approximation that each task $t$'s last iterate is close to $\theta_t^*$ this suggests that simple GBML methods such as Reptile [44] or FedAvg [41] are adaptive. The generality of ARUBA also allows for the incorporation of other dynamic regret bounds [25, 53] and other non-static notions of regret [27].

## 4 Adapting to the Inter-Task Geometry

Previously we gave improved guarantees for learning OMD under a simple notion of task-similarity: closeness of the optimal actions $\theta_t^*$. We now turn to new algorithms that can adapt to a more sophisticated task-similarity structure. Specifically, we study a class of learning algorithms parameterized by an initialization $\phi \in \Theta$ and a symmetric positive-definite matrix $H \in \mathcal{M} \subset \mathbb{R}^{d\times d}$ which plays

$$\theta_{t,i} = \arg\min_{\theta\in\Theta} \frac{1}{2}\|\theta - \phi\|_{H^{-1}}^2 + \langle \nabla_{t,1:i-1}, \theta \rangle \tag{5}$$

This corresponds $\theta_{t,i+1} = \theta_{t,i} - H\nabla_{t,i}$, so if the optimal actions $\theta_t^*$ vary strongly in certain directions, a matrix emphasizing those directions improves within-task performance. By strong-convexity of $\frac{1}{2}\|\theta - \phi\|_{H^{-1}}^2$ w.r.t. $\|\cdot\|_{H^{-1}}$, the regret-upper-bound is $\mathbf{U}_t(\phi, H) = \frac{1}{2}\|\theta_t^* - \phi\|_{H^{-1}}^2 + \sum_{i=1}^m \|\nabla_{t,i}\|_H^2$ [48, Theorem 2.15]. We first study the diagonal case, i.e. learning a per-coordinate learning rate $\eta \in \mathbb{R}^d$ to get iteration $\theta_{t,i+1} = \theta_{t,i} - \eta_t \odot \nabla_{t,i}$. We propose to set $\eta_t$ at each task $t$ as follows:

$$\eta_t = \sqrt{\frac{\sum_{s<t}\varepsilon_s^2 + \frac{1}{2}(\theta_s^* - \phi_s)^2}{\sum_{s<t}\zeta_s^2 + \sum_{i=1}^{m_s}\nabla_{s,i}^2}} \text{ for } \varepsilon_t^2 = \frac{\varepsilon^2}{(t+1)^p}, \zeta_t^2 = \frac{\zeta^2}{(t+1)^p} \forall t \ge 0, \text{ where } \varepsilon, \zeta, p > 0 \tag{6}$$

Observe the similarity between this update AdaGrad [21], which is also inversely related to the sum of the element-wise squares of all gradients seen so far. Our method adds multi-task information by setting the numerator to depend on the sum of squared distances between the initializations $\phi_t$ set by the algorithm and that task's optimal action $\theta_t^*$. This algorithm has the following guarantee:

**Theorem 4.1.** *Let $\Theta$ be a bounded convex subset of $\mathbb{R}^d$, let $\mathcal{D} \subset \mathbb{R}^{d\times d}$ be the set of positive definite diagonal matrices, and let each task $t \in [T]$ consist of a sequence of $m$ convex Lipschitz loss functions $\ell_{t,i} : \Theta \mapsto \mathbb{R}$. Suppose for each task $t$ we run the iteration in Equation 5 setting $\phi = \frac{1}{t-1}\theta_{1:t-1}^*$ and setting $H = \text{Diag}(\eta_t)$ via Equation 6 for $\varepsilon = 1, \zeta = \sqrt{m}$, and $p = \frac{2}{5}$. Then we achieve*

$$\bar{\mathbf{R}}_T \le \bar{\mathbf{U}}_T = \min_{\substack{\phi\in\Theta \\ H\in\mathcal{D}}} \tilde{\mathcal{O}}\left(\sum_{j=1}^d \min\left\{\frac{\frac{1}{H_{jj}} + H_{jj}}{T^{\frac{2}{5}}}, \frac{1}{\sqrt[5]{T}}\right\}\right)\sqrt{m} + \frac{1}{T}\sum_{t=1}^T \frac{\|\theta_t^* - \phi\|_{H^{-1}}^2}{2} + \sum_{i=1}^m \|\nabla_{t,i}\|_H^2$$

As $T \to \infty$ the average regret converges to the minimum over $\phi, H$ of the last two terms, which corresponds to running OMD with the optimal initialization and per-coordinate learning rate on every task. The rate of convergence of $T^{-2/5}$ is slightly slower than the usual $1/\sqrt{T}$ achieved in the previous section; this is due to the algorithm's adaptivity to within-task gradients, whereas previously we simply assumed a known Lipschitz bound $G_t$ when setting $\eta_t$. This adaptivity makes the algorithm much more practical, leading to a method for adaptively learning a within-task learning rate using multi-task information; this is outlined in Algorithm 2 and shown to significantly improve GBML performance in Section 6. Note also the per-coordinate separation of the left term, which shows that the algorithm converges more quickly on non-degenerate coordinates. The per-coordinate specification of $\eta_t$ (6) can be further generalized to learning a full-matrix adaptive regularizer, for which we show guarantees in Theorem 4.2. However, the rate is much slower, and without further assumptions such methods will have $\Omega(d^2)$ computation and memory requirements.

**Theorem 4.2.** *Let $\Theta$ be a bounded convex subset of $\mathbb{R}^d$ and let each task $t \in [T]$ consist of a sequence of $m$ convex Lipschitz loss functions $\ell_{t,i} : \Theta \mapsto \mathbb{R}$. Suppose for each task $t$ we run the iteration in Equation 5 with $\phi = \frac{1}{t-1}\theta^*_{1:t-1}$ and $H$ the unique positive definite solution of $B_t^2 = HG_t^2H$ for*

$$B_t^2 = t\varepsilon^2 I_d + \frac{1}{2}\sum_{s<t}(\theta_s^* - \phi_s)(\theta_s^* - \phi_s)^T \quad and \quad G_t^2 = t\zeta^2 I_d + \sum_{s<t}\sum_{i=1}^m \nabla_{s,i}\nabla_{s,i}^T$$

*for $\varepsilon = 1/\sqrt[8]{T}$ and $\zeta = \sqrt{m}/\sqrt[8]{T}$. Then for $\lambda_j$ corresponding to the $j$th largest eigenvalue we have*

$$\bar{\mathbf{R}}_T \le \bar{\mathbf{U}}_T = \tilde{\mathcal{O}}\left(\frac{1}{\sqrt[8]{T}}\right)\sqrt{m} + \min_{\substack{\phi \in \Theta \\ H \succ 0}} \frac{2\lambda_1^2(H)}{\lambda_d(H)}\frac{1+\log T}{T} + \sum_{t=1}^T \frac{\|\theta_t^* - \phi^*\|_{H^{-1}}^2}{2} + \sum_{i=1}^m \|\nabla_{t,i}\|_H^2$$

## 5 Fast Rates and High Probability Bounds for Statistical Learning-to-Learn

Batch-setting transfer risk bounds have been an important motivation for studying LTL via online learning [2, 34, 19]. If the regret-upper-bounds are convex, which is true for most practical variants of OMD/FTRL, ARUBA yields several new results in the classical distribution over task-distributions setup of Baxter [8]. In Theorem 5.1 we present bounds on the risk $\ell_{\mathcal{P}}(\bar{\theta})$ of the parameter $\bar{\theta}$ obtained by running OMD/FTRL on i.i.d. samples from a new task distribution $\mathcal{P}$ and averaging the iterates.

**Theorem 5.1.** *Assume $\Theta, \mathcal{X}$ are convex Euclidean subsets. Let convex losses $\ell_{t,i} : \Theta \mapsto [0,1]$ be drawn i.i.d. $\mathcal{P}_t \sim \mathcal{Q}, \{\ell_{t,i}\}_i \sim \mathcal{P}_t^m$ for distribution $\mathcal{Q}$ over tasks. Suppose they are passed to an algorithm with average regret upper-bound $\bar{\mathbf{U}}_T$ that at each $t$ picks $x_t \in \mathcal{X}$ to initialize a within-task method with convex regret upper-bound $\mathbf{U}_t : \mathcal{X} \mapsto [0, B\sqrt{m}]$, for $B \ge 0$. If the within-task algorithm is initialized by $\bar{x} = \frac{1}{T}x_{1:T}$ and it takes actions $\theta_1, \dots, \theta_m$ on $m$ i.i.d. losses from new task $\mathcal{P} \sim \mathcal{Q}$ then $\bar{\theta} = \frac{1}{m}\theta_{1:m}$ satisfies the following transfer risk bounds for any $\theta^* \in \Theta$ (all w.p. $1 - \delta$):*

1. **general case:** $\quad \mathbb{E}_{\mathcal{P}\sim\mathcal{Q}}\mathbb{E}_{\mathcal{P}^m}\ell_{\mathcal{P}}(\bar{\theta}) \le \mathbb{E}_{\mathcal{P}\sim\mathcal{Q}}\ell_{\mathcal{P}}(\theta^*) + \mathcal{L}_T \quad for \quad \mathcal{L}_T = \frac{\bar{\mathbf{U}}}{m} + B\sqrt{\frac{8}{mT}\log\frac{1}{\delta}}.$

2. **$\rho$-self-bounded losses $\ell$:** *if $\exists \rho > 0$ s.t. $\rho\mathbb{E}_{\ell\sim\mathcal{P}}\Delta\ell(\theta) \ge \mathbb{E}_{\ell\sim\mathcal{P}}(\Delta\ell(\theta) - \mathbb{E}_{\ell\sim\mathcal{P}}\Delta\ell(\theta))^2$ for all distributions $\mathcal{P} \sim \mathcal{Q}$, where $\Delta\ell(\theta) = \ell(\theta) - \ell(\theta^*)$ for any $\theta^* \in \arg\min_{\theta\in\Theta}\ell_{\mathcal{P}}(\theta)$, then for*
   $\mathcal{L}_T$ *as above we have* $\quad \mathbb{E}_{\mathcal{P}\sim\mathcal{Q}}\ell_{\mathcal{P}}(\bar{\theta}) \le \mathbb{E}_{\mathcal{P}\sim\mathcal{Q}}\ell_{\mathcal{P}}(\theta^*) + \mathcal{L}_T + \sqrt{\frac{2\rho\mathcal{L}_T}{m}\log\frac{2}{\delta}} + \frac{3\rho+2}{m}\log\frac{2}{\delta}.$

3. **$\alpha$-strongly-convex, $G$-Lipschitz regret-upper-bounds $\mathbf{U}_t$:** *in parts 1 and 2 above we can substitute* $\quad \mathcal{L}_T = \frac{\bar{\mathbf{U}} + \min_x \mathbb{E}_{\mathcal{P}\sim\mathcal{Q}}\mathbf{U}(x)}{m} + \frac{4G}{T}\sqrt{\frac{\bar{\mathbf{U}}}{\alpha m}\log\frac{8\log T}{\delta}} + \frac{\max\{16G^2, 6\alpha B\sqrt{m}\}}{\alpha mT}\log\frac{8\log T}{\delta}.$

In the **general case**, Theorem 5.1 provides bounds on the excess transfer risk decreasing with $\bar{\mathbf{U}}/m$ and $1/\sqrt{mT}$. Thus if $\bar{\mathbf{U}}$ improves with task-similarity so will the transfer risk as $T \to \infty$. Note that the second term is $1/\sqrt{mT}$ rather than $1/\sqrt{T}$ as in most-analyses [34, 19]; this is because regret is $m$-bounded but the OMD regret-upper-bound is $\mathcal{O}(\sqrt{m})$-bounded. The results also demonstrate ARUBA's ability to utilize specialized results from the online-to-batch conversion literature. This is witnessed by the guarantee for **self-bounded losses**, a class which Zhang [54] shows includes linear regression; we use a result by the same author to obtain high-probability bounds, whereas previous GBML bounds are in-expectation [34, 19]. We also apply a result due to Kakade and Tewari [33] for the case of **strongly-convex regret-upper-bounds**, enabling fast rates in the number of tasks $T$. The strongly-convex case is especially relevant for GBML since it holds for OGD with fixed learning rate.

**Algorithm 2:** ARUBA: an approach for modifying a generic batch GBML method to learn a per-coordinate learning rate. Two specialized variants provided below.

---

**Input:** $T$ tasks, update method for meta-initialization, within-task descent method, settings $\varepsilon, \zeta, p > 0$

Initialize $b_1 \leftarrow \varepsilon^2 1_d$, $g_1 \leftarrow \zeta^2 1_d$

**for** task $t = 1, 2, \ldots, T$ **do**

    Set $\phi_t$ according to update method, $\eta_t \leftarrow \sqrt{b_t/g_t}$

    Run descent method from $\phi_t$ with learning rate $\eta_t$:

        observe gradients $\nabla_{t,1}, \ldots, \nabla_{t,m_t}$

        obtain within-task parameter $\hat{\theta}_t$

    $b_{t+1} \leftarrow b_t + \frac{\varepsilon^2 1_d}{(t+1)^p} + \frac{1}{2}(\phi_t - \hat{\theta}_t)^2$

    $g_{t+1} \leftarrow g_t + \frac{\zeta^2 1_d}{(t+1)^p} + \sum_{i=1}^{m_t} \nabla_{t,i}^2$

**Result:** initialization $\phi_T$, learning rate $\eta_T = \sqrt{b_T/g_T}$

---

**ARUBA++:** starting with $\eta_{T,1} = \eta_T$ and $g_{T,1} = g_T$, adaptively reset the learning rate by setting $\hat{g}_{T,i+1} \leftarrow \hat{g}_{T,i} + c\nabla_i^2$ for some $c > 0$ and then updating $\eta_{T,i+1} \leftarrow \sqrt{b_T/g_{T,i+1}}$.
**Isotropic:** $b_t$ and $g_t$ are scalars tracking the sum of squared distances and sum of squared gradient norms, respectively.

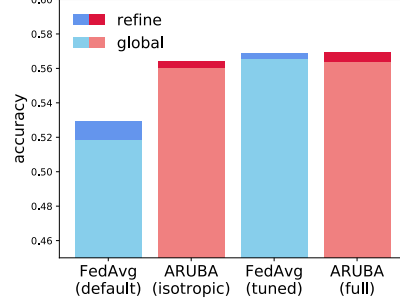

Figure 3: Next-character prediction performance for recurrent networks trained on the Shakespeare dataset [12] using FedAvg [41] and its modifications by Algorithm 2. Note that the two ARUBA methods require no learning rate tuning when personalizing the model (refine), unlike *both* FedAvg methods; this is a critical improvement in federated settings. Furthermore, isotropic ARUBA has negligible overhead by only communicating scalars.

We present two consequences of these results for the algorithms from Section 3 when run on i.i.d. data. To measure task-similarity we use the **variance** $V_Q^2 = \min_{\phi \in \Theta} \mathbb{E}_{\mathcal{P} \sim Q} \mathbb{E}_{\mathcal{P}^m} \|\theta^* - \phi\|_2^2$ of the empirical risk minimizer $\theta^*$ of an $m$-sample task drawn from $Q$. If $V_Q$ is known we can use strong-convexity of the regret-upper-bounds to obtain a fast rate for learning the initialization, as shown in the first part of Corollary 5.1. The result can be loosely compared to Denevi et al. [19], who provide a similar asymptotic improvement but with a slower rate of $\mathcal{O}(1/\sqrt{T})$ in the second term. However, their task-similarity measures the deviation of the true, not empirical, risk-minimizers, so the results are not directly comparable. Corollary 5.1 also gives a guarantee for when we do *not* know $V_Q$ and must learn the learning rate $\eta$ in addition to the initialization; here we match the rate of Denevi et al. [19], who do not learn $\eta$, up to some additional fast $o(1/\sqrt{m})$ terms.

**Corollary 5.1.** *In the setting of Theorems 3.2 & 5.1, if $\delta \leq 1/e$ and Algorithm 1 uses within-task OGD with initialization $\phi_{t+1} = \frac{1}{t}\theta_{1:t}^*$ and step-size $\eta_t = \frac{V_Q + 1/\sqrt{T}}{G\sqrt{m}}$ for $V_Q$ as above, then w.p. $1 - \delta$*

$$\mathbb{E}_{\mathcal{P} \sim Q} \mathbb{E}_{\mathcal{P}^m} \ell_{\mathcal{P}}(\bar{\theta}) \leq \mathbb{E}_{\mathcal{P} \sim Q} \ell_{\mathcal{P}}(\theta^*) + \tilde{\mathcal{O}}\left( \frac{V_Q}{\sqrt{m}} + \left( \frac{1}{\sqrt{mT}} + \frac{1}{T} \right) \log \frac{1}{\delta} \right)$$

*If $\eta_t$ is set adaptively using $\varepsilon$-EWOO as in Theorem 3.2 for $\varepsilon = 1/\sqrt[4]{mT} + 1/\sqrt{m}$ then w.p. $1 - \delta$*

$$\mathbb{E}_{\mathcal{P} \sim Q} \mathbb{E}_{\mathcal{P}^m} \ell_{\mathcal{P}}(\bar{\theta}) \leq \mathbb{E}_{\mathcal{P} \sim Q} \ell_{\mathcal{P}}(\theta^*) + \tilde{\mathcal{O}}\left( \frac{V_Q}{\sqrt{m}} + \min\left\{ \frac{\frac{1}{\sqrt{m}} + \frac{1}{\sqrt{T}}}{V_Q m}, \frac{1}{\sqrt[4]{m^3 T}} + \frac{1}{m} \right\} + \sqrt{\frac{1}{T} \log \frac{1}{\delta}} \right)$$

## 6 Empirical Results: Adaptive Methods for Few-Shot & Federated Learning

A generic GBML method does the following at iteration $t$: (1) initialize a descent method at $\phi_t$; (2) take gradient steps with learning rate $\eta$ to get task-parameter $\hat{\theta}_t$; (3) update meta-initialization to $\phi_{t+1}$. Motivated by Section 4, in Algorithm 2 we outline a generic way of replacing $\eta$ by a per-coordinate rate learned on-the-fly. This entails keeping track of two quantities: (1) $b_t \in \mathbb{R}^d$, a per-coordinate sum over $s < t$ of the squared distances from the initialization $\phi_s$ to within-task parameter $\hat{\theta}_s$; (2) $g_t \in \mathbb{R}^d$, a per-coordinate sum of the squared gradients seen so far. At task $t$ we set $\eta$ to be the element-wise square root of $b_t/g_t$, allowing multi-task information to inform the trajectory. For example, if along coordinate $j$ the $\hat{\theta}_{t,j}$ is usually not far from initialization then $b_j$ will be small and thus so will $\eta_j$; then if on a new task we get a high noisy gradient along coordinate $j$ the performance will be less adversely affected because it will be down-weighted by the learning rate. Single-task algorithms such as AdaGrad [21] and Adam [36] also work by reducing the learning rate along frequent directions.

|  |  | 20-way Omniglot | | 5-way Mini-ImageNet | |
|  |  | 1-shot | 5-shot | 1-shot | 5-shot |
|---|---|---|---|---|---|
| 1st Order | 1st-Order MAML [23] | $89.4 \pm 0.5$ | $\underline{97.9} \pm 0.1$ | $48.07 \pm 1.75$ | $63.15 \pm 0.91$ |
|  | Reptile [44] w. Adam [36] | $89.43 \pm 0.14$ | $97.12 \pm 0.32$ | $49.97 \pm 0.32$ | $\mathbf{65.99} \pm 0.58$ |
|  | Reptile w. ARUBA | $86.67 \pm 0.17$ | $96.61 \pm 0.13$ | $\mathbf{50.73} \pm 0.32$ | $65.69 \pm 0.61$ |
|  | Reptile w. ARUBA++ | $\underline{89.66} \pm 0.3$ | $97.49 \pm 0.28$ | $50.35 \pm 0.74$ | $65.89 \pm 0.34$ |
| 2nd Order | 2nd-Order MAML | $95.8 \pm 0.3$ | $98.9 \pm 0.2$ | $48.7 \pm 1.84$ | $63.11 \pm 0.92$ |
|  | Meta-SGD [38] | $\mathbf{95.93} \pm 0.38$ | $\mathbf{98.97} \pm 0.19$ | $\underline{50.47} \pm 1.87$ | $64.03 \pm 0.94$ |

Table 1: Meta-test-time performance of GBML algorithms on few-shot classification benchmarks. 1st-order and 2nd-order results obtained from Nichol et al. [44] and Li et al. [38], respectively.

However, in meta-learning some coordinates may be frequently updated during meta-training because good task-weights vary strongly from the best initialization along them, and thus their gradients should not be downweighted; ARUBA encodes this intuition in the numerator using the distance-traveled per-task along each direction, which increases the learning rate along high-variance directions. We show in Figure 2 that this is realized in practice, as ARUBA assigns a faster rate to deeper layers than to lower-level feature extractors, following standard intuition in parameter-transfer meta-learning. As described in Algorithm 2, we also consider two variants: ARUBA++, which updates the meta-learned learning-rate at meta-test-time in a manner similar to AdaGrad, and Isotropic ARUBA, which only tracks scalar quantities and is thus useful for communication-constrained settings.

**Few-Shot Classification:** We first examine if Algorithm 2 can improve performance on Omniglot [37] and Mini-ImageNet [46], two standard few-shot learning benchmarks, when used to modify Reptile, a simple meta-learning method [44]. In its serial form Reptile is roughly the algorithm we study in Section 3 when OGD is used within-task and $\eta$ is fixed. Thus we can set Reptile+ARUBA to be Algorithm 2 with $\hat{\theta}_t$ the last iterate of OGD and the meta-update a weighted sum of $\hat{\theta}_t$ and $\phi_t$. In practice, however, Reptile uses Adam [36] to exploit multi-task gradient information. As shown in Table 1, ARUBA matches or exceeds this baseline on Mini-ImageNet, although on Omniglot it requires the additional within-task updating of ARUBA++ to show improvement.

It is less clear how ARUBA can be applied to MAML [23], as by only taking one step the distance traveled will be proportional to the gradient, so $\eta$ will stay fixed. We also do not find that ARUBA improves multi-step MAML – perhaps not surprising as it is further removed from our theory due to its use of held-out data. In Table 1 we compare to Meta-SGD [38], which does learn a per-coordinate learning rate for MAML by automatic differentiation. This requires more computation but does lead to consistent improvement. As with the original Reptile, our modification performs better on Mini-ImageNet but worse on Omniglot compared to MAML and its modification Meta-SGD.

**Federated Learning:** A main goal in this setting is to use data on heterogeneous nodes to learn a global model without much communication; leveraging this to get a personalized model is an auxiliary goal [50], with a common application being next-character prediction on mobile devices. A popular method is FedAvg [41], where at each communication round $r$ the server sends a global model $\phi_r$ to a batch of nodes, which then run local OGD; the server then sets $\phi_{r+1}$ to the average of the returned models. This can be seen as a GBML method with each node a task, making it easy to apply ARUBA: each node simply sends its accumulated squared gradients to the server together with its model. The server can use this information and the squared difference between $\phi_r$ and $\phi_{r+1}$ to compute a learning rate $\eta_{r+1}$ via Algorithm 2 and send it to each node in the next round. We use FedAvg with ARUBA to train a character LSTM [29] on the Shakespeare dataset, a standard benchmark of a thousand users with varying amounts of non-i.i.d. data [41, 12]. Figure 3 shows that ARUBA significantly improves over non-tuned FedAvg and matches the performance of FedAvg with a tuned learning rate schedule. Unlike both baselines we also do not require step-size tuning when refining the global model for personalization. This reduced need for hyperparameter optimization is crucial in federated settings, where the number of user-data accesses are extremely limited.

## 7 Conclusion

In this paper we introduced ARUBA, a framework for analyzing GBML that is both flexible and consequential, yielding new guarantees for adaptive, dynamic, and statistical LTL via online learning. As a result we devised a novel per-coordinate learning rate applicable to generic GBML procedures, improving their training and meta-test-time performance on few-shot and federated learning. We see great potential for applying ARUBA to derive many other new LTL methods in a similar manner.

## Acknowledgments

We thank Jeremy Cohen, Travis Dick, Nikunj Saunshi, Dravyansh Sharma, Ellen Vitercik, and our three anonymous reviewers for helpful feedback. This work was supported in part by DARPA FA875017C0141, National Science Foundation grants CCF-1535967, CCF-1910321, IIS-1618714, IIS-1705121, IIS-1838017, and IIS-1901403, a Microsoft Research Faculty Fellowship, a Bloomberg Data Science research grant, an Amazon Research Award, an Amazon Web Services Award, an Okawa Grant, a Google Faculty Award, a JP Morgan AI Research Faculty Award, and a Carnegie Bosch Institute Research Award. Any opinions, findings and conclusions, or recommendations expressed in this material are those of the authors and do not necessarily reflect the views of DARPA, the National Science Foundation, or any other funding agency.

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
