[Supplementary Material]

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

# A  Background and Results for Online Convex Optimization

Throughout the appendix we assume all subsets are convex and in a finite-dimensional real vector space with inner product $\langle \cdot, \cdot \rangle$ unless explicitly stated. Let $\| \cdot \|_*$ be the dual norm of $\| \cdot \|$ and note that the dual norm of $\| \cdot \|_2$ is itself. For sequences of scalars $\sigma_1, \ldots, \sigma_T \in \mathbb{R}$ we will use the notation $\sigma_{1:t}$ to refer to the sum of the first $t$ of them. In the online learning setting, we will use the shorthand $\nabla_t$ to denote the subgradient of $\ell_t : \Theta \mapsto \mathbb{R}$ evaluated at action $\theta_t \in \Theta$. We will use $\mathrm{Conv}(S)$ to refer to the convex hull of a set of points $S$ and $\mathrm{Proj}_S(\cdot)$ to be the projection to any convex subset $S$.

## A.1  Convex Functions

We first state the related definitions of *strong convexity* and *strong smoothness*:

**Definition A.1.** *An everywhere sub-differentiable function $f : S \mapsto \mathbb{R}$ is $\alpha$-**strongly-convex** w.r.t. norm $\| \cdot \|$ if*

$$f(y) \geq f(x) + \langle \nabla f(x), y - x \rangle + \frac{\alpha}{2} \|y - x\|^2 \,\forall\, x, y \in S$$

**Definition A.2.** *An everywhere sub-differentiable function $f : S \mapsto \mathbb{R}$ is $\beta$-**strongly-smooth** w.r.t. norm $\| \cdot \|$ if*

$$f(y) \leq f(x) + \langle \nabla f(x), y - x \rangle + \frac{\beta}{2} \|y - x\|^2 \,\forall\, x, y \in S$$

Finally, we will also consider functions that are exp-concave [28]:

**Definition A.3.** *An everywhere sub-differentiable function $f : S \mapsto \mathbb{R}$ is $\gamma$-**exp-concave** if $\exp(-\gamma f(x))$ is concave. For $S \subset \mathbb{R}$ we have that $\frac{\partial_{xx} f(x)}{(\partial_x f(x))^2} \geq \gamma \,\forall\, x \in S \implies f$ is $\gamma$-exp-concave.*

We now turn to the *Bregman divergence* and a discussion of several useful properties [10, 6]:

**Definition A.4.** *Let $f : S \mapsto \mathbb{R}$ be an everywhere sub-differentiable strictly convex function. Its **Bregman divergence** is defined as*

$$\mathcal{B}_f(x\|y) = f(x) - f(y) - \langle \nabla f(y), x - y \rangle$$

*The definition directly implies that $\mathcal{B}_f(\cdot\|y)$ preserves the (strong or strict) convexity of $f$ for any fixed $y \in S$. Strict convexity further implies $\mathcal{B}_f(x\|y) \geq 0 \,\forall\, x, y \in S$, with equality iff $x = y$. Finally, if $f$ is $\alpha$-strongly-convex, or $\beta$-strongly-smooth, w.r.t. $\| \cdot \|$ then Definitions A.1 and A.2 imply $\mathcal{B}_f(x\|y) \geq \frac{\alpha}{2} \|x - y\|^2$ or $\mathcal{B}_f(x\|y) \leq \frac{\beta}{2} \|x - y\|^2$, respectively.*

**Claim A.1.** *Let $f : S \mapsto \mathbb{R}$ be a strictly convex function on $S$, $\alpha_1, \ldots, \alpha_n \in \mathbb{R}$ be a sequence satisfying $\alpha_{1:n} > 0$, and $x_1, \ldots, x_n \in S$. Then*

$$\bar{x} = \frac{1}{\alpha_{1:n}} \sum_{i=1}^{n} \alpha_i x_i = \arg \min_{y \in S} \sum_{i=1}^{n} \alpha_i \mathcal{B}_f(x_i\|y)$$

*Proof.* $\forall\, y \in S$ we have

$$\sum_{i=1}^{n} \alpha_i \left( \mathcal{B}_f(x_i\|y) - \mathcal{B}_f(x_i\|\bar{x}) \right)$$

$$= \sum_{i=1}^{n} \alpha_i \left( f(x_i) - f(y) - \langle \nabla f(y), x_i - y \rangle - f(x_i) + f(\bar{x}) + \langle \nabla f(\bar{x}), x_i - \bar{x} \rangle \right)$$

$$= (f(\bar{x}) - f(y) + \langle \nabla f(y), y \rangle) \alpha_{1:n} + \sum_{i=1}^{n} \alpha_i \left( -\langle \nabla f(\bar{x}), \bar{x} \rangle + \langle \nabla f(\bar{x}) - \nabla f(y), x_i \rangle \right)$$

$$= (f(\bar{x}) - f(y) - \langle \nabla f(y), \bar{x} - y \rangle) \alpha_{1:n}$$

$$= \alpha_{1:n} \mathcal{B}_f(\bar{x}\|y)$$

By Definition A.4 the last expression has a unique minimum at $y = \bar{x}$. ☐

## A.2 Online Algorithms

Here we provide a review of the online algorithms we use. Recall that in this setting our goal is minimizing regret:

**Definition A.5.** *The **regret** of an agent playing actions $\{\theta_t \in \Theta\}_{t\in[T]}$ on a sequence of loss functions $\{\ell_t : \Theta \mapsto \mathbb{R}\}_{t\in[T]}$ is*

$$\mathbf{R}_T = \sum_{t=1}^{T} \ell_t(\theta_t) - \min_{\theta\in\Theta} \sum_{t=1}^{T} \ell_t(\theta)$$

Within-task our focus is on two closely related meta-algorithms, Follow-the-Regularized-Leader (FTRL) and (linearized lazy) Online Mirror Descent (OMD).

**Definition A.6.** *Given a strictly convex function $R : \Theta \mapsto \mathbb{R}$, starting point $\phi \in \Theta$, fixed learning rate $\eta > 0$, and a sequence of functions $\{\ell_t : \Theta \mapsto \mathbb{R}\}_{t\geq1}$, **Follow-the-Regularized Leader** $(\mathrm{FTRL}_{\phi,\eta}^{(R)})$ plays*

$$\theta_t = \operatorname*{arg\,min}_{\theta\in\Theta} \mathcal{B}_R(\theta||\phi) + \eta \sum_{s<t} \ell_s(\theta)$$

**Definition A.7.** *Given a strictly convex function $R : \Theta \mapsto \mathbb{R}$, starting point $\phi \in \Theta$, fixed learning rate $\eta > 0$, and a sequence of functions $\{\ell_t : \Theta \mapsto \mathbb{R}\}_{t\geq1}$, **lazy linearized Online Mirror Descent** $(\mathrm{OMD}_{\phi,\eta}^{(R)})$ plays*

$$\theta_t = \operatorname*{arg\,min}_{\theta\in\Theta} \mathcal{B}_R(\theta||\phi) + \eta \sum_{s<t} \langle \nabla_s, \theta \rangle$$

These formulations make the connection between the two algorithms – their equivalence in the linear case $\ell_s(\cdot) = \langle \nabla_s, \cdot \rangle$ – very explicit. There exists a more standard formulation of OMD that is used to highlight its generalization of OGD – the case of $R(\cdot) = \frac{1}{2}\|\cdot\|_2^2$ – and the fact that the update is carried out in the dual space induced by $R$ [26, Section 5.3]. However, we will only need the following regret bound satisfied by both [48, Theorems 2.11 and 2.15]

**Theorem A.1.** *Let $\{\ell_t : \Theta \mapsto \mathbb{R}\}_{t\in[T]}$ be a sequence of convex functions that are $G_t$-Lipschitz w.r.t. $\|\cdot\|$ and let $R : S \mapsto \mathbb{R}$ be 1-strongly-convex. Then the regret of both $\mathrm{FTRL}_{\eta,\phi}^{(R)}$ and $\mathrm{OMD}_{\eta,\phi}^{(R)}$ is bounded by*

$$\mathbf{R}_T \leq \frac{\mathcal{B}_R(\theta^*||\phi)}{\eta} + \eta G^2 T$$

*for all $\theta^* \in \Theta$ and $G^2 \geq \frac{1}{T}\sum_{t=1}^{T} G_t^2$.*

We next review the online algorithms we use for the meta-update. The main requirement here is logarithmic regret guarantees for the case of strongly convex loss functions, which is satisfied by two well-known algorithms:

**Definition A.8.** *Given a sequence of strictly convex functions $\{\ell_t : \Theta \mapsto \mathbb{R}\}_{t\geq1}$, **Follow-the-Leader** (FTL) plays arbitrary $\theta_1 \in \Theta$ and for $t > 1$ plays*

$$\theta_t = \operatorname*{arg\,min}_{\theta\in\Theta} \sum_{s<t} \ell_s(\theta)$$

**Definition A.9.** *Given a sequence of functions $\{\ell_t : \Theta \mapsto \mathbb{R}\}_{t\geq1}$ that are $\alpha_t$-strongly-convex w.r.t. $\|\cdot\|_2$, **Adaptive OGD (AOGD)** plays arbitrary $\theta_1 \in \Theta$ and for $t > 1$ plays*

$$\theta_{t+1} = \mathrm{Proj}_\Theta \left( \theta_t - \frac{1}{\alpha_{1:t}} \nabla f(\theta_t) \right)$$

Kakade and Shalev-Shwartz [32, Theorem 2] and Bartlett et al. [7, Theorem 2.1] provide for FTL and AOGD, respectively, the following regret bound:

**Theorem A.2.** *Let $\{\ell_t : \Theta \mapsto \mathbb{R}\}_{t\in[T]}$ be a sequence of convex functions that are $G_t$-Lipschitz and $\alpha_t$-strongly-convex w.r.t. $\|\cdot\|$. Then the regret of both FTL and AOGD is bounded by*

$$\mathbf{R}_T \leq \frac{1}{2}\sum_{t=1}^{T} \frac{G_t^2}{\alpha_{1:t}}$$

Finally, we state the EWOO algorithm due to Hazan et al. [28]. While difficult to run in high-dimensions, we will be running this method in single dimensions, when computing it requires only one integral.

**Definition A.10.** *Given a sequence of $\gamma$-exp-concave functions $\{\ell_t : \Theta \mapsto \mathbb{R}\}$,* **Exponentially Weighted Online Optimization (EWOO)** *plays*

$$\theta_t = \frac{\int_\Theta \theta \exp(-\gamma \sum_{s<t} \ell_s(\theta)) d\theta}{\int_\Theta \exp(-\gamma \sum_{s<t} \ell_s(\theta)) d\theta}$$

Hazan et al. [28, Theorem 7] provide the following guarantee for EWOO, which is notable for its lack of explicit dependence on the Lipschitz constant.

**Theorem A.3.** *Let $\{\ell_t : \Theta \mapsto \mathbb{R}\}$ be a sequence of $\gamma$-exp-concave functions. Then the regret of EWOO is bounded by*

$$\mathbf{R}_T \leq \frac{d}{\gamma}(1 + \log(T+1))$$

## A.3   Online-to-Batch Conversion

Finally, as we are also interested in distributional meta-learning, we discuss some techniques for converting regret guarantees into generalization bounds, which are usually named *online-to-batch conversions*. We first state some standard results.

**Proposition A.1.** *If a sequence of bounded convex loss functions $\{\ell_t : \Theta \mapsto \mathbb{R}\}_{t \in [T]}$ drawn i.i.d. from some distribution $\mathcal{D}$ is given to an online algorithm with regret bound $\mathbf{R}_T$ that generates a sequence of actions $\{\theta_t \in \Theta\}_{t \in [T]}$ then*

$$\mathop{\mathbb{E}}_{\mathcal{D}^T} \mathop{\mathbb{E}}_{\ell \sim \mathcal{D}} \ell(\bar{\theta}) \leq \mathop{\mathbb{E}}_{\ell \sim \mathcal{D}} \ell(\theta^*) + \frac{\mathbf{R}_T}{T}$$

*for $\bar{\theta} = \frac{1}{T}\theta_{1:T}$ and any $\theta^* \in \Theta$.*

*Proof.* Applying Jensen's inequality yields

$$\mathop{\mathbb{E}}_{\mathcal{D}^T} \mathop{\mathbb{E}}_{\ell \sim \mathcal{D}} \ell(\bar{\theta}) \leq \frac{1}{T} \mathop{\mathbb{E}}_{\mathcal{D}^T} \sum_{t=1}^T \mathop{\mathbb{E}}_{\ell_t' \sim \mathcal{D}} \ell_t'(\theta_t)$$

$$= \frac{1}{T} \mathop{\mathbb{E}}_{\{\ell_t\} \sim \mathcal{D}^T} \left( \sum_{t=1}^T \mathop{\mathbb{E}}_{\ell_t' \sim \mathcal{D}} \ell_t'(\theta_t) - \ell_t(\theta_t) \right) + \frac{1}{T} \mathop{\mathbb{E}}_{\{\ell_t\} \sim \mathcal{D}^T} \left( \sum_{t=1}^T \ell_t(\theta_t) \right)$$

$$\leq \frac{1}{T} \sum_{t=1}^T \mathop{\mathbb{E}}_{\{\ell_s\}_{s<t} \sim \mathcal{D}^{t-1}} \left( \mathop{\mathbb{E}}_{\ell_t' \sim \mathcal{D}} \ell_t'(\theta_t) - \mathop{\mathbb{E}}_{\ell_t \sim \mathcal{D}} \ell_t(\theta_t) \right) + \frac{\mathbf{R}_T}{T} + \frac{1}{T} \sum_{t=1}^T \mathop{\mathbb{E}}_{\ell \sim \mathcal{D}} \ell(\theta^*)$$

$$= \frac{\mathbf{R}_T}{T} + \mathop{\mathbb{E}}_{\ell \sim \mathcal{D}} \ell(\theta^*)$$

where we used the fact that $\theta_t$ only depends on $\ell_1, \ldots, \ell_{t-1}$. $\qquad \square$

For nonnegative bounded losses we have the following fact [14, Proposition 1]:

**Proposition A.2.** *If a sequence of loss functions $\{\ell_t : \Theta \mapsto [0,1]\}_{t \in [T]}$ drawn i.i.d. from some distribution $\mathcal{D}$ is given to an online algorithm that generates a sequence of actions $\{\theta_t \in \Theta\}_{t \in [T]}$ then*

$$\frac{1}{T} \sum_{t=1}^T \mathop{\mathbb{E}}_{\ell \sim \mathcal{D}} \ell(\theta_t) \leq \frac{1}{T} \sum_{t=1}^T \ell_t(\theta_t) + \sqrt{\frac{2}{T} \log \frac{1}{\delta}} \qquad w.p.\ 1 - \delta$$

$$\frac{1}{T} \sum_{t=1}^T \mathop{\mathbb{E}}_{\ell \sim \mathcal{D}} \ell(\theta_t) \geq \frac{1}{T} \sum_{t=1}^T \ell_t(\theta_t) - \sqrt{\frac{2}{T} \log \frac{1}{\delta}} \qquad w.p.\ 1 - \delta$$

Note that Cesa-Bianchi et al. [14] only prove the first inequality; the second follows via the same argument but applying the symmetric version of the Azuma-Hoeffding inequality [4]. The inequalities above can be easily used to derive the following competitive bounds:

**Corollary A.1.** *If a sequence of loss functions $\{\ell_t : \Theta \mapsto [0,1]\}_{t \in [T]}$ drawn i.i.d. from some distribution $\mathcal{D}$ is given to an online algorithm with regret bound $\mathbf{R}_T$ that generates a sequence of actions $\{\theta_t \in \Theta\}_{t \in [T]}$ then*

$$\underset{t \sim \mathcal{U}[T]}{\mathbb{E}} \underset{\ell \sim \mathcal{D}}{\mathbb{E}} \ell(\theta_t) \leq \underset{\ell \sim \mathcal{D}}{\mathbb{E}} \ell(\theta^*) + \frac{\mathbf{R}_T}{T} + \sqrt{\frac{8}{T} \log \frac{1}{\delta}} \qquad \text{w.p. } 1 - \delta$$

*for any $\theta^* \in \Theta$. If the losses are also convex then for $\bar{\theta} = \frac{1}{T}\theta_{1:T}$ we have*

$$\underset{\ell \sim \mathcal{D}}{\mathbb{E}} \ell(\bar{\theta}) \leq \underset{\ell \sim \mathcal{D}}{\mathbb{E}} \ell(\theta^*) + \frac{\mathbf{R}_T}{T} + \sqrt{\frac{8}{T} \log \frac{1}{\delta}} \qquad \text{w.p. } 1 - \delta$$

*Proof.* By Proposition A.2 we have

$$\frac{1}{T}\sum_{t=1}^{T} \underset{\ell \sim \mathcal{D}}{\mathbb{E}} \ell(\theta_t) \leq \frac{1}{T}\sum_{t=1}^{T} \ell_t(\theta^*) + \frac{\mathbf{R}_T}{T} + \sqrt{\frac{2}{T}\log\frac{1}{\delta}} \leq \underset{\ell \sim \mathcal{D}}{\mathbb{E}} \ell(\theta^*) + \frac{\mathbf{R}_T}{T} + \sqrt{\frac{8}{T}\log\frac{1}{\delta}}$$

Apply linearity of expectations to get the first inequality and Jensen's inequality to get the second. $\square$

We now discuss some stronger guarantees for certain classes of loss functions. The first, due to Kakade and Tewari [33, Theorem 2], yields faster rates for strongly convex losses:

**Theorem A.4.** *Let $\mathcal{D}$ be some distribution over loss functions $\ell : \Theta \mapsto [0, B]$ for some $B > 0$ that are $G$-Lipschitz w.r.t. $\|\cdot\|$ for some $G > 0$ and $\alpha$-strongly-convex w.r.t $\|\cdot\|$ for some $\alpha > 0$. If a sequence of loss functions $\{\ell_t\}_{t \in [T]}$ is drawn i.i.d. from $\mathcal{D}$ and given to an online algorithm with regret bound $\mathbf{R}_T$ that generates a sequence of actions $\{\theta_t \in \Theta\}_{t \in [T]}$ then w.p. $1 - \delta$ we have for $\bar{\theta} = \frac{1}{T}\theta_{1:T}$ and any $\theta^* \in \Theta$ that*

$$\underset{\ell \sim \mathcal{D}}{\mathbb{E}} \ell(\bar{\theta}) \leq \underset{\ell \sim \mathcal{D}}{\mathbb{E}} \ell(\theta^*) + \frac{\mathbf{R}_T}{T} + \frac{4G}{T}\sqrt{\frac{\mathbf{R}_T}{\alpha}\log\frac{4\log T}{\delta}} + \frac{\max\{16G^2, 6\alpha B\}}{\alpha T}\log\frac{4\log T}{\delta}$$

We can also obtain a data-dependent bound using a result of Zhang [54] under a self-bounding property. Cesa-Bianchi and Gentile [13, Proposition 2] show a similar but less general result.

**Definition A.11.** *A distribution $\mathcal{D}$ over $\ell : \Theta \mapsto \mathbb{R}$ has $\rho$-**self-bounding** losses if $\forall \theta \in \Theta$ we have*

$$\rho \underset{\ell \sim \mathcal{D}}{\mathbb{E}} \ell(\theta) \geq \underset{\ell \sim \mathcal{D}}{\mathbb{E}} (\ell(\theta) - \underset{\ell \sim \mathcal{D}}{\mathbb{E}} \ell(\theta))^2$$

**Theorem A.5.** *Let $\mathcal{D}$ be some distribution over $\rho$-self-bounding convex loss functions $\ell : \Theta \mapsto [-1, 1]$ for some $\rho > 0$. If a sequence of loss functions $\{\ell_t\}_{t \in [T]}$ is drawn i.i.d. from $\mathcal{D}$ and given to an online algorithm with regret bound $\mathbf{R}_T$ that generates a sequence of actions $\{\theta_t \in \Theta\}_{t \in [T]}$ then w.p. $1 - \delta$ we have*

$$\underset{\ell \sim \mathcal{D}}{\mathbb{E}} \ell(\bar{\theta}) \leq \bar{L}_T + \sqrt{\frac{2\rho\max\{0, \bar{L}_T\}}{T}\log\frac{1}{\delta}} + \frac{3\rho + 2}{T}\log\frac{1}{\delta}$$

*where $\bar{\theta} = \frac{1}{T}\theta_{1:T}$ and $\bar{L}_T = \frac{1}{T}\sum_{t=1}^{T}\ell_t(\theta_t)$ is the average loss suffered by the agent.*

*Proof.* Apply Jensen's inequality and Zhang [54, Theorem 4]. $\square$

Note that nonnegative 1-bounded convex losses satisfy the conditions of Theorem A.5 with $\rho = 1$. However, we are interested in a different result that can yield a data-dependent competitive bound:

**Corollary A.2.** *Let $\mathcal{D}$ be some distribution over convex loss functions $\ell : \Theta \mapsto [0, 1]$ such that the functions $\ell(\theta) - \ell(\theta^*)$ are $\rho$-self-bounded for some $\theta^* \in \arg\min_{\theta \in \Theta} \mathbb{E}_{\ell \sim \mathcal{D}} \ell(\theta)$. If a sequence of loss functions $\{\ell_t\}_{t \in [T]}$ is drawn i.i.d. from $\mathcal{D}$ and given to an online algorithm with regret bound $\mathbf{R}_T$ that generates a sequence of actions $\{\theta_t \in \Theta\}_{t \in [T]}$ then w.p. $1 - \delta$ we have*

$$\underset{\ell \sim \mathcal{D}}{\mathbb{E}} \ell(\bar{\theta}) \leq \underset{\ell \sim \mathcal{D}}{\mathbb{E}} \ell(\theta^*) + \frac{\mathbf{R}_T}{T} + \frac{1}{T}\sqrt{2\rho\mathbf{R}_T\log\frac{1}{\delta}} + \frac{3\rho + 2}{T}\log\frac{1}{\delta}$$

*where $\bar{\theta} = \frac{1}{T}\theta_{1:T}$ and $\mathcal{E}^* = \arg\min_{\theta \in \Theta} \mathbb{E}\,\ell(\theta)$.*

*Proof.* Apply Theorem A.5 over the sequence of functions $\{\ell_t(\theta) - \ell_t(\theta^*)\}_{t \in [T]}$ and by definition of regret substitute $\bar{L}_T = \frac{1}{T}\sum_{t=1}^{T}\ell_t(\theta) - \ell_t(\theta^*) \leq \frac{\mathbf{R}_T}{T}$. $\square$

Zhang [54, Lemma 7] shows that the conditions are satisfied for $\rho = 4$ by least-squares regression.

### A.4 Dynamic Regret Guarantees

Here we review several results for optimizing dynamic regret. We first define this quantity:

**Definition A.12.** *The **dynamic regret** of an agent playing actions $\{\theta_t \in \Theta\}_{t \in [T]}$ on a sequence of loss functions $\{\ell_t : \Theta \mapsto \mathbb{R}\}$ w.r.t. a sequence of reference parameters $\Psi = \{\psi_t\}_{t \in [T]}$ is*

$$\mathbf{R}_T(\Psi) = \sum_{t=1}^{T} \ell_t(\theta_t) - \sum_{t=1}^{T} \ell_t(\psi_t)$$

Mokhtari et al. [42, Corollary 1] show the following guarantee for OGD over strongly convex functions:

**Theorem A.6.** *Let $\{\ell_t : \Theta \mapsto \mathbb{R}\}_{t \in [T]}$ be a sequence of $\alpha$-strongly-convex, $\beta$-strongly-smooth, and $G$-Lipschitz functions w.r.t. $\|\cdot\|_2$. Then OGD with step-size $\eta \leq \frac{1}{\beta}$ achieves dynamic regret*

$$\mathbf{R}_T(\Psi) \leq \frac{GD}{1-\rho}\left(1 + \sum_{t=2}^{T} \|\psi_t - \psi_{t-1}\|_2\right)$$

*w.r.t. reference sequence $\Psi = \{\psi_t\}_{t \in [T]}$ for $\rho = \sqrt{1 - \frac{h\alpha}{\eta}}$ for any $h \in (0,1]$ and $D$ the $\ell_2$-diameter of $\Theta$.*

# B  Strongly Convex Coupling

Our first result is a simple trick that we believe may be of independent interest. It allows us to bound the regret of FTL on any (possibly non-convex) sequence of Lipschitz functions so long as the actions played are identical to those played on a different strongly-convex sequence of Lipschitz functions. The result is formalized in Theorem B.1.

## B.1  Derivation

We start with some standard facts about convex functions.

**Claim B.1.** *Let $f : S \mapsto \mathbb{R}$ be an everywhere sub-differentiable convex function. Then for any norm $\| \cdot \|$ we have*

$$f(x) - f(y) \leq \|\nabla f(x)\|_* \|x - y\| \ \forall \ x, y \in S$$

**Claim B.2.** *Let $f : S \mapsto \mathbb{R}$ be $\alpha$-strongly-convex w.r.t. $\| \cdot \|$ with minimum $x^* \in \arg\min_{x \in S} f(x)$. Then $x^*$ is unique and for all $x \in S$ we have*

$$f(x) \geq f(x^*) + \frac{\alpha}{2}\|x - x^*\|^2$$

Next we state some technical results, starting with the well-known be-the-leader lemma [48, Lemma 2.1].

**Lemma B.1.** *Let $\theta_1, \ldots, \theta_{T+1} \in \Theta$ be the sequence of actions of FTL on the function sequence $\{\ell_t : \Theta \mapsto \mathbb{R}\}_{t \in [T]}$. Then*

$$\sum_{t=1}^{T} \ell_t(\theta_t) - \ell_t(\theta^*) \leq \sum_{t=1}^{T} \ell_t(\theta_t) - \ell_t(\theta_{t+1})$$

*for all $\theta^* \in \Theta$.*

The final result depends on a stability argument for FTL on strongly-convex functions adapted from Saha et al. [47]:

**Lemma B.2.** *Let $\{\ell_t : \Theta \mapsto \mathbb{R}\}_{t \in [T]}$ be a sequence of functions that are $\alpha_t$-strongly-convex w.r.t. $\| \cdot \|$ and let $\theta_1, \ldots, \theta_{T+1} \in \Theta$ be the corresponding sequence of actions of FTL. Then*

$$\|\theta_t - \theta_{t+1}\| \leq \frac{2\|\nabla_t\|_*}{\alpha_t + 2\alpha_{1:t-1}}$$

*for all $t \in [T]$.*

*Proof.* The proof slightly generalizes an argument in Saha et al. [47, Theorem 6]. For each $t \in [T]$ we have by Claim B.2 and the $\alpha_{1:t}$-strong-convexity of $\sum_{s=1}^{t} \ell_s(\cdot)$ that

$$\sum_{s=1}^{t} \ell_s(\theta_t) \geq \sum_{s=1}^{t} \ell_s(\theta_{t+1}) + \frac{\alpha_{1:t}}{2}\|\theta_t - \theta_{t+1}\|^2$$

We similarly have

$$\sum_{s=1}^{t-1} \ell_s(\theta_{t+1}) \geq \sum_{s=1}^{t-1} \ell_s(\theta_t) + \frac{\alpha_{1:t-1}}{2}\|\theta_{t+1} - \theta_t\|^2$$

Adding these two inequalities and applying Claim B.1 yields

$$\left(\frac{\alpha_t}{2} + \alpha_{1:t-1}\right)\|\theta_t - \theta_{t+1}\|^2 \leq \ell_t(\theta_t) - \ell_t(\theta_{t+1}) \leq \|\nabla_t\|_* \|\theta_t - \theta_{t+1}\|$$

Dividing by $\|\theta_t - \theta_{t+1}\|$ yields the result. □

**Theorem B.1.** *Let $\{\ell_t : \Theta \mapsto \mathbb{R}\}_{t \in [T]}$ be a sequence of functions that are $G_t$-Lipschitz in $\|\cdot\|_A$ and let $\theta_1, \ldots, \theta_{T+1}$ be the sequence of actions produced by FTL. Let $\{\ell'_t : \Theta \mapsto \mathbb{R}\}_{t \in [T]}$ be a sequence of functions on which FTL also plays $\theta_1, \ldots, \theta_{T+1}$ but which are $G'_t$-Lipschitz and $\alpha_t$-strongly-convex in $\|\cdot\|_B$. Then*

$$\sum_{t=1}^{T} \ell_t(\theta_t) - \ell_t(\theta^*) \le 2C \sum_{t=1}^{T} \frac{G_t G'_t}{\alpha_t + 2\alpha_{1:t-1}}$$

*for all $\theta^* \in \Theta$ and some constant $C$ s.t. $\|\theta\|_A \le C\|\theta\|_B \; \forall \, \theta \in \Theta$. If the functions $\ell_t$ are also convex then we have*

$$\sum_{t=1}^{T} \ell_t(\theta_t) - \ell_t(\theta^*) \le 2C \sum_{t=1}^{T} \frac{\|\nabla_t\|_{A,*}\|\nabla'_t\|_{B,*}}{\alpha_t + 2\alpha_{1:t-1}}$$

*or all $\theta^* \in \Theta$*

*Proof.* By Lemma B.2,

$$\|\theta_t - \theta_{t+1}\|_A \le C\|\theta_t - \theta_{t+1}\|_B \le \frac{2CG'_t}{\alpha_t + 2\alpha_{1:t-1}}$$

for all $t \in [T]$. Then by Lemma B.1 and the $G_t$-Lipschitzness of $\ell_t$ we have for all $\theta^* \in \Theta$ that

$$\sum_{t=1}^{T} \ell_t(\theta_t) - \ell(\theta^*) \le \sum_{t=1}^{T} \ell_t(\theta_t) - \ell(\theta_{t+1}) \le \sum_{t=1}^{T} G_t\|\theta_t - \theta_{t+1}\|_A \le 2C \sum_{t=1}^{T} \frac{G_t G'_t}{\alpha_t + 2\alpha_{1:t-1}}$$

In the convex case we instead apply Claim B.1 and Lemma B.2 to get

$$\sum_{t=1}^{T} \ell_t(\theta_t) - \ell_t(\theta^*) \le \sum_{t=1}^{T} \ell_t(\theta_t) - \ell(\theta_{t+1}) \le \sum_{t=1}^{T} \|\nabla_t\|_{A,*}\|\theta_t - \theta_{t+1}\|_A \le 2C \sum_{t=1}^{T} \frac{\|\nabla_t\|_{A,*}\|\nabla'_t\|_{B,*}}{\alpha_t + 2\alpha_{1:t-1}}$$

$\square$

## B.2 Applications

We now show two applications of strongly convex coupling. The first shows logarithmic regret for FTL run on a sequence of Bregman regularizers. Note that these functions are nonconvex in general.

**Proposition B.1.** *Let $R : \Theta \mapsto \mathbb{R}$ be 1-strongly-convex w.r.t. $\|\cdot\|$ and consider any $\theta_1, \ldots, \theta_T \in \Theta$. Then when run on the loss sequence $\alpha_1 \mathcal{B}_R(\theta_1\|\cdot), \ldots, \alpha_T \mathcal{B}_R(\theta_T\|\cdot)$ for any positive scalars $\alpha_1, \ldots, \alpha_T \in \mathbb{R}_+$, FTL obtains regret*

$$\mathbf{R}_T \le 2CD \sum_{t=1}^{T} \frac{\alpha_t^2 G_t}{\alpha_t + 2\alpha_{1:t-1}}$$

*for $C$ s.t. $\|\theta\| \le C\|\theta\|_2 \; \forall \, \theta \in \Theta$, $D = \max_{\theta,\phi \in \Theta} \|\theta - \phi\|_2$ the $\ell_2$-diameter of $\Theta$, and $G_t$ the Lipschitz constant of $\mathcal{B}_R(\theta_t\|\cdot)$ over $\Theta$ w.r.t. $\|\cdot\|$. Note that for $\|\cdot\| = \|\cdot\|_2$ we have $C = 1$ and $G_t \le D \; \forall \, t \in [T]$.*

*Proof.* Note that $\alpha_t \mathcal{B}_R(\theta_t\|\cdot)$ is $\alpha_t G_t$-Lipschitz w.r.t. $\|\cdot\|$. Let $R'(\cdot) = \frac{1}{2}\|\cdot\|_2^2$, so $\mathcal{B}_{R'}(\theta_t\|\phi) = \frac{1}{2}\|\theta_t - \phi\|_2^2 \; \forall \, \phi \in \Theta, t \in [T]$. The function $\alpha_t \mathcal{B}_{R'}(\theta_t\|\cdot)$ is thus $\alpha_t$-strongly-convex and $D$-Lipschitz w.r.t. $\|\cdot\|_2$. Now by Claim A.1 FTL run on this new sequence plays the same actions as FTL run on the original sequence. Applying Theorem B.1 yields the result. $\square$

In the next application we use coupling to give a $\tilde{\mathcal{O}}(T^{\frac{3}{5}})$-regret algorithm for a sequence of non-Lipschitz convex functions.

**Proposition B.2.** *Let $\{\ell_t : \mathbb{R}_+ \mapsto \mathbb{R}\}_{t \geq 1}$ be a sequence of functions of form $\ell_t(x) = \left(\frac{B_t^2}{x} + x\right)\alpha_t$ for any positive scalars $\alpha_1, \ldots, \alpha_T \in \mathbb{R}_+$ and adversarially chosen $B_t \in [0, D]$. Then the $\varepsilon$-FTL algorithm, which for $\varepsilon > 0$ uses the actions of FTL run on the functions $\tilde{\ell}_t(x) = \left(\frac{B_t^2 + \varepsilon^2}{x} + x\right)\alpha_t$ over the domain $[\varepsilon, \sqrt{D^2 + \varepsilon^2}]$ to determine $x_t$, achieves regret*

$$\mathbf{R}_T \leq \min\left\{\frac{\varepsilon^2}{x^*}, \varepsilon\right\}\alpha_{1:T} + 2D \max\left\{\frac{D^3}{\varepsilon^3}, 1\right\}\sum_{t=1}^T \frac{\alpha_t^2}{\alpha_t + 2\alpha_{1:t-1}}$$

*for all $x^* > 0$.*

*Proof.* Define $\tilde{B}_t^2 = B_t^2 + \varepsilon^2$ and note that FTL run on the functions $\tilde{\ell}_t'(x) = \left(\frac{x^2}{2} - \tilde{B}_t^2 \log x\right)\alpha_t$ plays the exact same actions $x_t^2 = \frac{\sum_{s<t} \alpha_s \tilde{B}_s^2}{\alpha_{1:t-1}}$ as FTL run on $\tilde{\ell}_t$. We have that

$$|\partial_x \tilde{\ell}_t| = \alpha_t \left|1 - \frac{\tilde{B}_t^2}{x^2}\right| \leq \frac{\alpha_t D^2}{\varepsilon^2}$$

$$|\partial_x \tilde{\ell}_t'| = \alpha_t \left|x - \frac{\tilde{B}_t^2}{x}\right| \leq \alpha_t \max\left\{D, \frac{D^2}{\varepsilon}\right\} \qquad \partial_{xx}\tilde{\ell}_t' = \alpha_t\left(1 + \frac{\tilde{B}_t^2}{x^2}\right) \geq \alpha_t$$

so the functions $\tilde{\ell}_t$ are $\frac{\alpha_t D^2}{\varepsilon^2}$-Lipschitz while the functions $\tilde{\ell}_t'$ are $\alpha_t D \max\left\{\frac{D}{\varepsilon}, 1\right\}$-Lipschitz and $\alpha_t$-strongly-convex. Therefore by Theorem B.1 we have that

$$\sum_{t=1}^T \tilde{\ell}_t(x_t) - \tilde{\ell}_t(x^*) \leq 2D \max\left\{\frac{D^3}{\varepsilon^3}, 1\right\}\sum_{t=1}^T \frac{\alpha_t^2}{\alpha_t + 2\alpha_{1:t-1}}$$

for any $x^* \in [\varepsilon, \sqrt{D^2 + \varepsilon^2}]$. Since $\sum_{t=1}^T \tilde{\ell}_t$ is minimized on $[\varepsilon, \sqrt{D^2 + \varepsilon^2}]$, the above also holds for all $x^* > 0$. Therefore we have that

$$\sum_{t=1}^T \ell_t(x_t) \leq \sum_{t=1}^T \left(\frac{B_t^2 + \varepsilon^2}{x_t} + x_t\right)\alpha_t$$

$$= \sum_{t=1}^T \tilde{\ell}_t(x_t)$$

$$\leq \min_{x^*>0} 2D \max\left\{\frac{D^3}{\varepsilon^3}, 1\right\}\sum_{t=1}^T \frac{\alpha_t^2}{\alpha_t + 2\alpha_{1:t-1}} + \sum_{t=1}^T \tilde{\ell}_t(x^*)$$

$$= \min_{x^*>0} 2D \max\left\{\frac{D^3}{\varepsilon^3}, 1\right\}\sum_{t=1}^T \frac{\alpha_t^2}{\alpha_t + 2\alpha_{1:t-1}} + \sum_{t=1}^T \left(\frac{B_t^2 + \varepsilon^2}{x^*} + x^*\right)\alpha_t$$

$$= \min_{x^*>0} \frac{\varepsilon^2}{x^*}\alpha_{1:T} + 2D \max\left\{\frac{D^3}{\varepsilon^3}, 1\right\}\sum_{t=1}^T \frac{\alpha_t^2}{\alpha_t + 2\alpha_{1:t-1}} + \sum_{t=1}^T \ell_t(x^*)$$

Note that substituting $x^* = \sqrt{\frac{\sum_{t=1}^T \alpha_t \tilde{B}_t^2}{\alpha_{1:T}}}$ into the second-to-last line yields

$$\min_{x^*>0} \sum_{t=1}^T \left(\frac{B_t^2 + \varepsilon^2}{x^*} + x^*\right)\alpha_t \leq 2\sqrt{\alpha_{1:T}\sum_{t=1}^T \alpha_t \tilde{B}_t^2} \leq 2\varepsilon\alpha_{1:T} + \min_{x^*>0}\sum_{t=1}^T \ell_t(x^*)$$

completing the proof. $\qquad\square$

## C  Adaptive and Dynamic Guarantees

Throughout Appendices C, D, and E we assume that $\arg\min_{\theta\in\Theta}\sum_{\ell\in\mathcal{S}}\ell(\theta)$ returns a unique minimizer of the sum of the loss functions in the sequence $\mathcal{S}$. Formally, this can be defined to be the one minimizing an appropriate Bregman divergence $\mathcal{B}_R(\cdot|\phi_R)$ from some fixed $\phi_R\in\Theta$, e.g. the origin in Euclidean space or the uniform distribution over the simplex, which is unique by strong-convexity of $\mathcal{B}_R(\cdot|\phi_R)$ and convexity of the set of optimizers of a convex function.

**Theorem C.1.** *Let each task $t\in[T]$ consist of a sequence of $m_t$ convex loss functions $\ell_{t,i}:\Theta\mapsto\mathbb{R}$ that are $G_{t,i}$-Lipschitz w.r.t. $\|\cdot\|$. For $G_t^2=G_{1:m_t}^2/m_t$ and $R:\Theta\mapsto\mathbb{R}$ a 1-strongly-convex function w.r.t. $\|\cdot\|$ define the following online algorithms:*

1. INIT*: a method that has dynamic regret $\mathbf{U}_T^{init}(\Psi)=\sum_{t=1}^T f_t^{init}(\phi_t)-f_t^{init}(\psi_t)$ w.r.t. reference actions $\Psi=\{\psi_t\}_{t=1}^T\subset\Theta$ over the sequence $f_t^{init}(\cdot)=\mathcal{B}_R(\theta_t^*\|\cdot)G_t\sqrt{m_t}$ .*

2. SIM*: a method that has (static) regret $\mathbf{U}_T^{sim}(x)$ decreasing in $x>0$ over the sequence of functions $f_t^{sim}(x)=\left(\frac{\mathcal{B}_R(\theta_t^*\|\phi_t)}{x}+x\right)G_t\sqrt{m_t}$.*

*Then if Algorithm 1 sets $\phi_t=\mathrm{INIT}(t)$ and $\eta_t=\frac{\mathrm{SIM}(t)}{G_t\sqrt{m_t}}$ it will achieve*

$$\bar{\mathbf{R}}_T\leq\bar{\mathbf{U}}_T\leq\frac{\mathbf{U}_T^{sim}(V_\Psi)}{T}+\frac{1}{T}\min\left\{\frac{\mathbf{U}_T^{init}(\Psi)}{V_\Psi},2\sqrt{\mathbf{U}_T^{init}(\Psi)\sum_{t=1}^T G_t\sqrt{m_t}}\right\}+\frac{2V_\Psi}{T}\sum_{t=1}^T G_t\sqrt{m_t}$$

*for $V_\Psi^2=\frac{1}{\sum_{t=1}^T G_t\sqrt{m_t}}\sum_{t=1}^T\mathcal{B}_R(\theta_t^*\|\psi_t)G_t\sqrt{m_t}$.*

*Proof.* Letting $x_t=\mathrm{SIM}(t)$ be the output of SIM at time $t$, defining $\sigma_t=G_t\sqrt{m_t}$ and $\sigma_{1:T}=\sum_{t=1}^T\sigma_t$, and substituting into the regret-upper-bound of OMD/FTRL (2), we have that

$$\bar{\mathbf{U}}_T T=\sum_{t=1}^T\left(\frac{\mathcal{B}_R(\theta_t^*\|\phi_t)}{x_t}+x_t\right)\sigma_t\leq\min_{x>0}\mathbf{U}_T^{sim}(x)+\sum_{t=1}^T\left(\frac{\mathcal{B}_R(\theta_t^*\|\phi_t)}{x}+x\right)\sigma_t$$

$$\leq\min_{x>0}\mathbf{U}_T^{sim}(x)+\frac{\mathbf{U}_T^{init}(\Psi)}{x}+\sum_{t=1}^T\left(\frac{\mathcal{B}_R(\theta_t^*\|\psi_t)}{x}+x\right)\sigma_t$$

$$\leq\mathbf{U}_T^{sim}(V_\Psi)+\min\left\{\frac{\mathbf{U}_T^{init}(\Psi)}{V_\Psi},2\sqrt{\mathbf{U}_T^{init}(\Psi)\sigma_{1:T}}\right\}+2V_\Psi\sigma_{1:T}$$

where the last line follows by substituting $x=\max\left\{V_\Psi,\sqrt{\frac{\mathbf{U}_T^{init}(\Psi)}{\sigma_{1:T}}}\right\}$. $\qquad\square$

**Corollary C.1.** *Under the assumptions of Theorem C.1 and boundedness of $\mathcal{B}_R$ over $\Theta$, if INIT uses FTL, or AOGD in the case of $R(\cdot)=\frac{1}{2}\|\cdot\|_2^2$, and SIM uses $\varepsilon$-FTL as defined in Proposition B.2, then Algorithm 1 achieves*

$$\bar{\mathbf{U}}_T T\leq\min\left\{\frac{\varepsilon^2}{V},\varepsilon\right\}\sigma_{1:T}+2D\max\left\{\frac{D^3}{\varepsilon^3},1\right\}\sum_{t=1}^T\frac{\sigma_t^2}{\sigma_{1:t}}+\sqrt{8CD\sigma_{1:T}\sum_{t=1}^T\frac{\sigma_t^2}{\sigma_{1:t}}}+2V\sigma_{1:T}$$

*for $V^2=\min_{\phi\in\Theta}\sum_{t=1}^T\sigma_t\mathcal{B}_R(\theta_t^*\|\phi)$ and constant $C$ the product of the constant $C$ from Proposition B.1 and the bound on the gradient of the Bregman divergence. Assuming $\sigma_t=G\sqrt{m}\ \forall\ t$ and substituting $\varepsilon=\frac{1}{\sqrt[5]{T}}$ yields*

$$\bar{\mathbf{R}}_T\leq\bar{\mathbf{U}}_T=\tilde{\mathcal{O}}\left(\min\left\{\frac{1}{VT^{\frac{2}{5}}}+\frac{1}{\sqrt{T}},\frac{1}{\sqrt[5]{T}}\right\}+V\right)\sqrt{m}$$

*Proof.* Substitute Propositions B.1 and B.2 into Theorem C.1. $\qquad\square$

**Proposition C.1.** *Let $\{\ell_t : \mathbb{R}_+ \mapsto \mathbb{R}\}_{t \geq 1}$ be a sequence of functions of form $\ell_t(x) = \left(\frac{B_t^2}{x} + x\right)\alpha_t$ for any positive scalars $\alpha_1, \ldots, \alpha_T \in \mathbb{R}_+$ and adversarially chosen $B_t \in [0, D]$. Then the losses $\tilde{\ell}_t(x) = \left(\frac{B_t^2 + \varepsilon^2}{x} + x\right)\alpha_t$ over the domain $[\varepsilon, \sqrt{D^2 + \varepsilon^2}]$ are $\frac{\alpha_t D^2}{\varepsilon^2}$-Lipschitz and $\frac{2}{\alpha_t D} \min\left\{\frac{\varepsilon^2}{D^2}, 1\right\}$-exp-concave.*

*Proof.* Lipschitzness follows by taking derivatives as in Proposition B.2. Define $\tilde{B}_t^2 = B_t^2 + \varepsilon^2$. We then have

$$\partial_x \tilde{\ell}_t = \alpha_t \left(1 - \frac{\tilde{B}_t^2}{x^2}\right) \qquad\qquad \partial_{xx}\tilde{\ell}_t = \frac{2\alpha_t \tilde{B}_t^2}{x^3}$$

The $\gamma$-exp-concavity of the functions $\tilde{\ell}_t$ can be determined by finding the largest $\gamma$ satisfying

$$\gamma \leq \frac{\partial_{xx}\tilde{\ell}_t}{(\partial_x \tilde{\ell}_t)^2} = \frac{2\tilde{B}_t^2 x}{\alpha_t (\tilde{B}_t^2 - x^2)^2}$$

for all $x \in [\varepsilon, \sqrt{D^2 + \varepsilon^2}]$ and all $t \in [T]$. We first minimize jointly over choice of $x, \tilde{B}_t \in [\varepsilon, \sqrt{D^2 + \varepsilon^2}]$. The derivatives of the objective w.r.t. $x$ and $\tilde{B}_t$, respectively, are

$$\frac{2\tilde{B}_t^2(\tilde{B}_t^2 + 3x^2)}{(\tilde{B}_t^2 - x^2)^3} \qquad\qquad -\frac{4\tilde{B}_t x(\tilde{B}_t^2 + x^2)}{(\tilde{B}_t^2 - x^2)^3}$$

Note that the objective approaches $\infty$ as the coordinates approach the line $x = \tilde{B}_t$. For $x < \tilde{B}_t$ the derivative w.r.t. $x$ is always positive while the derivative w.r.t. $\tilde{B}_t$ is always negative. Since we have the constraints $x \geq \varepsilon$ and $\tilde{B}_t^2 \leq D^2 + \varepsilon^2$, the optimum over $x < \tilde{B}_t$ is thus attained at $x = \varepsilon$ and $\tilde{B}_t^2 = D^2 + \varepsilon^2$. Substituting into the original objective yields

$$\frac{2(D^2 + \varepsilon^2)\varepsilon}{\alpha_t D^4} \geq \frac{2\varepsilon}{\alpha_t D^2}$$

For $x > \tilde{B}_t$ the derivative w.r.t. $x$ is always negative while the derivative w.r.t. $\tilde{B}_t$ is always positive. Since we have the constraints $x \leq \sqrt{D^2 + \varepsilon^2}$ and $\tilde{B}_t^2 \geq \varepsilon^2$, the optimum over $x > \tilde{B}_t$ is thus attained at $x = \sqrt{D^2 + \varepsilon^2}$ and $\tilde{B}_t^2 = \varepsilon^2$. Substituting into the original objective yields

$$\frac{2\varepsilon^2\sqrt{D^2 + \varepsilon^2}}{\alpha_t D^4} \geq \frac{2\varepsilon^2}{\alpha_t D^3}$$

Thus we have that the functions $\tilde{\ell}_t$ are $\frac{2}{\alpha_t D} \min\left\{\frac{\varepsilon^2}{D^2}, 1\right\}$-exp-concave. $\qquad\square$

**Corollary C.2.** *Let $\{\ell_t : \mathbb{R}_+ \mapsto \mathbb{R}\}_{t \geq 1}$ be a sequence of functions of form $\ell_t(x) = \left(\frac{B_t^2}{x} + x\right)\alpha_t$ for any positive scalars $\alpha_1, \ldots, \alpha_T \in \mathbb{R}_+$ and adversarially chosen $B_t \in [0, D]$. Then the $\varepsilon$-EWOO algorithm, which for $\varepsilon > 0$ uses the actions of EWOO run on the functions $\tilde{\ell}_t(x) = \left(\frac{B_t^2 + \varepsilon^2}{x} + x\right)\alpha_t$ over the domain $[\varepsilon, \sqrt{D^2 + \varepsilon^2}]$ to determine $x_t$, achieves regret*

$$\mathbf{R}_T \leq \min_{x^* > 0}\left\{\frac{\varepsilon^2}{x^*}, \varepsilon\right\}\alpha_{1:T} + \frac{D\alpha_{\max}}{2}\max\left\{\frac{D^2}{\varepsilon^2}, 1\right\}(1 + \log(T + 1))$$

*for all $x^* > 0$.*

*Proof.* Since $\sum_{t=1}^T \tilde{\ell}_t$ is minimized on $[\varepsilon, \sqrt{D^2 + \varepsilon^2}]$, we apply Theorem A.3 and follow a similar argument to that concluding Proposition B.2 to get

$$\sum_{t=1}^T \ell_t(x_t) \leq \frac{D\alpha_{\max}}{2}\max\left\{\frac{D^2}{\varepsilon^2}, 1\right\}(1 + \log(T + 1)) + \sum_{t=1}^T \tilde{\ell}_t(x^*)$$

$$= \min_{x^* > 0}\left\{\frac{\varepsilon^2}{x^*}, \varepsilon\right\}\alpha_{1:T} + \frac{D\alpha_{\max}}{2}\max\left\{\frac{D^2}{\varepsilon^2}, 1\right\}(1 + \log(T + 1)) + \sum_{t=1}^T \ell_t(x^*)$$

$\qquad\square$

**Corollary C.3.** *Under the assumptions of Theorem C.1 and boundedness of $\mathcal{B}_R$ over $\Theta$, if* INIT *uses FTL, or AOGD in the case of $R(\cdot) = \frac{1}{2}\|\cdot\|_2^2$, and* SIM *uses $\varepsilon$-EWOO as defined in Proposition C.2, then Algorithm 1 achieves*

$$\bar{\mathbf{U}}_T T \leq \min\left\{\frac{\varepsilon^2}{V}, \varepsilon\right\} \sigma_{1:T} + \frac{D\sigma_{\max}}{2}\max\left\{\frac{D^2}{\varepsilon^2}, 1\right\}(1+\log(T+1)) + \sqrt{8CD\sigma_{1:T}\sum_{t=1}^{T}\frac{\sigma_t^2}{\sigma_{1:t}}} + 2V\sigma_{1:T}$$

*for $V^2 = \min_{\phi \in \Theta}\sum_{t=1}^{T}\sigma_t \mathcal{B}_R(\theta_t^*\|\phi)$ and constant $C$ the product of the constant $C$ from Proposition B.1 and the bound on the gradient of the Bregman divergence. Assuming $\sigma_t = G\sqrt{m} \; \forall \, t$ and substituting $\varepsilon = \frac{1}{\sqrt[4]{T}}$ yields*

$$\bar{\mathbf{R}}_T \leq \bar{\mathbf{U}}_T = \tilde{\mathcal{O}}\left(\min\left\{\frac{1+\frac{1}{V}}{\sqrt{T}}, \frac{1}{\sqrt[4]{T}}\right\} + V\right)\sqrt{m}$$

*Proof.* Substitute Proposition B.1 and Corollary C.2 into Theorem C.1. $\square$

**Corollary C.4.** *Under the assumptions of Theorem 3.1 and boundedness of $\Theta$, if* INIT *is OGD with learning rate $\frac{1}{\sigma_{\max}}$ and* SIM *uses $\varepsilon$-EWOO as defined in Proposition C.2 then Algorithm 1 achieves*

$$\bar{\mathbf{U}}_T T \leq \min\left\{\frac{\varepsilon^2}{V_\Psi}, \varepsilon\right\}\sigma_{1:T} + \frac{D\sigma_{\max}}{2}\max\left\{\frac{D^2}{\varepsilon^2}, 1\right\}(1+\log(T+1))$$

$$+ 2D\min\left\{\frac{D\sigma_{\max}}{V_\Psi}(1+P_\Psi), \sqrt{2\sigma_{\max}\sigma_{1:T}(1+P_\Psi)}\right\} + 2V_\Psi\sigma_{1:T}$$

*for $P_T(\Psi) = \sum_{t=2}^{T}\|\psi_t - \psi_{t-1}\|_2$. Assuming $\sigma_t = G\sqrt{m} \; \forall \, t$ and substituting $\varepsilon = \frac{1}{\sqrt[4]{T}}$ yields*

$$\bar{\mathbf{R}}_T \leq \bar{\mathbf{U}}_T = \tilde{\mathcal{O}}\left(\min\left\{\frac{1+\frac{1}{V_\Psi}}{\sqrt{T}}, \frac{1}{\sqrt[4]{T}}\right\} + \min\left\{\frac{1+P_\Psi}{V_\Psi T}, \sqrt{\frac{1+P_\Psi}{T}}\right\} + V_\Psi\right)\sqrt{m}$$

*Proof.* Substitute Theorem 3.3 and Corollary C.2 into Theorem C.1. $\square$

# D Adapting to the Inter-Task Geometry

For clarity, vectors and matrices in this section will be **bolded**, although scalar regret quantities will continue to be as well. For any two vectors $\boldsymbol{x}, \boldsymbol{y} \in \mathbb{R}^d$, $\boldsymbol{x} \odot \boldsymbol{y}$ will denote element-wise multiplication, $\frac{\boldsymbol{x}}{\boldsymbol{y}}$ will denote element-wise division, $\boldsymbol{x}^p$ will denote raising each element of $\boldsymbol{x}$ to the power $p$, and $\max\{\boldsymbol{x}, \boldsymbol{y}\}$ and $\min\{\boldsymbol{x}, \boldsymbol{y}\}$ will denote element-wise maximum and minimum, respectively. For any nonnegative $\boldsymbol{a} \in \mathbb{R}^d$ we will use the notation $\|\cdot\|_{\boldsymbol{a}} = \langle \sqrt{\boldsymbol{a}}, \cdot \rangle$; note that if all elements of $\boldsymbol{a}$ are positive then $\|\cdot\|_{\boldsymbol{a}}$ is a norm on $\mathbb{R}^d$ with dual norm $\|\cdot\|_{\boldsymbol{a}^{-1}}$.

**Claim D.1.** *For $t \geq 1$ and $p \in (0,1)$ we have*

$$\sum_{s=0}^{t-1} \frac{1}{(s+1)^p} \geq \sum_{s=1}^{t} \frac{1}{(s+1)^p} \geq \underline{c}_p t^{1-p} \qquad and \qquad \sum_{s=1}^{t} \frac{1}{s^p} \leq \overline{c}_p t^{1-p}$$

*for $\underline{c}_p = \frac{1 - \left(\frac{2}{3}\right)^{1-p}}{1-p}$ and $\overline{c}_p = \frac{1}{1-p}$.*

*Proof.*

$$\sum_{s=0}^{t-1} \frac{1}{(s+1)^p} \geq \sum_{s=1}^{t} \frac{1}{(s+1)^p} \geq \int_1^{t+1} \frac{ds}{(s+1)^p} = \frac{(t+2)^{1-p} - 2^{1-p}}{1-p} \geq \underline{c}_p (t+2)^{1-p} \geq \underline{c}_p t^{1-p}$$

$$\sum_{s=1}^{t} \frac{1}{s^p} \leq 1 + \int_1^t \frac{ds}{s^p} = 1 + \frac{t^{1-p} - 1}{1-p} \leq \overline{c}_p t^{1-p}$$

$\square$

**Claim D.2.** *For any $\boldsymbol{x} \in \mathbb{R}^d$ we have $\|\boldsymbol{x}^2\|_2^2 \leq \|\boldsymbol{x}\|_2^4$.*

*Proof.*

$$\|\boldsymbol{x}^2\|_2^2 = \sum_{j=1}^d x_j^4 \leq \left( \sum_{j=1}^d x_j^2 \right)^2 = \|\boldsymbol{x}\|_2^4$$

$\square$

We now review some facts from matrix analysis. Throughout this section we will use matrices in $\mathbb{R}^{d \times d}$; we denote the subset of symmetric matrices by $\mathbb{S}^d$, the subset of symmetric PSD matrices by $\mathbb{S}^d_+$, and the subset of symmetric positive-definite matrices by $\mathbb{S}^d_{++}$. Note that every symmetric matrix $\boldsymbol{A} \in \mathbb{S}^d$ has diagonalization $\boldsymbol{A} = \boldsymbol{V}\boldsymbol{\Lambda}\boldsymbol{V}^{-1}$ for diagonal matrix $\boldsymbol{\Lambda} \in \mathbb{S}^d$ containing the eigenvalues of $\boldsymbol{A}$ along the diagonal and a matrix $\boldsymbol{V} \in \mathbb{R}^{d \times d}$ of orthogonal eigenvectors. For such matrices we will use $\lambda_j(\boldsymbol{A})$ to denote the $j$th largest eigenvalue of $\boldsymbol{A}$ and for any function $f : [\lambda_d(\boldsymbol{A}), \lambda_1(\boldsymbol{A})] \mapsto \mathbb{R}$ we will use the notation

$$f(\boldsymbol{A}) = \boldsymbol{V} \begin{pmatrix} f(\boldsymbol{\Lambda}_{11}) & & \\ & \ddots & \\ & & f(\boldsymbol{\Lambda}_{dd}) \end{pmatrix} \boldsymbol{V}^{-1}$$

We will denote the spectral norm by $\|\cdot\|_2$ and the Frobenius norm by $\|\cdot\|_F$.

**Claim D.3.** *[9, Section A.4.1] $f(\boldsymbol{X}) = \log \det \boldsymbol{X}$ has gradient $\nabla_{\boldsymbol{X}} f = \boldsymbol{X}^{-1}$ over $\mathbb{S}^d_{++}$*

**Claim D.4.** *[43, Theorem 3.1] The function $f(\boldsymbol{X}) = -\log \det \boldsymbol{X}$ is $\frac{1}{\sigma^2}$-strongly-convex w.r.t. $\|\cdot\|_2$ over the set of symmetric positive-definite matrices with spectral norm bounded by $\sigma$.*

**Definition D.1.** *A function* $f : (0, \infty) \mapsto \mathbb{R}$ *is* **operator convex** *if* $\forall\, \boldsymbol{X}, \boldsymbol{Y} \in \mathbb{S}^d_{++}$ *and any* $t \in [0, 1]$ *we have*

$$f(t\boldsymbol{X} + (1 - t)\boldsymbol{Y}) \preceq tf(\boldsymbol{X}) + (1 - t)f(\boldsymbol{Y})$$

**Claim D.5.** *If* $\boldsymbol{A} \in \mathbb{S}^d_+$ *and* $f : (0, \infty) \mapsto \mathbb{R}$ *is operator convex then* $\mathrm{Tr}(\boldsymbol{A}f(\boldsymbol{X}))$ *is convex on* $\mathbb{S}^d_{++}$.

*Proof.* Consider any $\boldsymbol{X}, \boldsymbol{Y} \in \mathbb{S}^d_{++}$ and any $t \in [0, 1]$. By the operator convexity of $f$, positive semi-definiteness of $\boldsymbol{A}$, and linearity of the trace functional we have that

$$
\begin{aligned}
0 &\preceq \mathrm{Tr}(\boldsymbol{A}(tf(\boldsymbol{X}) + (1 - t)f(\boldsymbol{Y}) - f(t\boldsymbol{X} + (1 - t)\boldsymbol{Y}))) \\
&= t\,\mathrm{Tr}(\boldsymbol{A}(f(\boldsymbol{X}))) + (1 - t)\,\mathrm{Tr}(\boldsymbol{A}f(\boldsymbol{Y})) - \mathrm{Tr}(\boldsymbol{A}(f(t\boldsymbol{X} + (1 - t)\boldsymbol{Y})))
\end{aligned}
$$

$\square$

**Corollary D.1.** *If* $\boldsymbol{A} \in \mathbb{S}^d_+$ *then* $\mathrm{Tr}(\boldsymbol{A}\boldsymbol{X}^{-1})$ *and* $\mathrm{Tr}(\boldsymbol{A}\boldsymbol{X})$ *are convex over* $\mathbb{S}^d_{++}$.

*Proof.* By the Löwner-Heinz theorem [17], $x^{-1}, x$, and $x^2$ are operator convex. The result follows by applying Claim D.5. $\square$

**Corollary D.2.** *[39, Corollary 1.1] If* $\boldsymbol{A}, \boldsymbol{B} \in \mathbb{S}^d_+$ *then* $\mathrm{Tr}(\boldsymbol{A}\boldsymbol{X}\boldsymbol{B}\boldsymbol{X})$ *is convex over* $\mathbb{S}^d_+$.

**Proposition D.1.** *Let* $\{\ell_t : \mathbb{R}_+ \mapsto \mathbb{R}\}_{t\geq 1}$ *be of form* $\ell_t(\boldsymbol{x}) = \left\|\frac{\boldsymbol{b}_t^2}{\boldsymbol{x}} + \boldsymbol{g}_t^2 \odot \boldsymbol{x}\right\|_1$ *for adversarially chosen* $\boldsymbol{b}_t, \boldsymbol{g}_t$ *satisfying* $\|\boldsymbol{b}_t\|_2 \leq D, \|\boldsymbol{g}_t\|_2 \leq G$. *Then the* $(\varepsilon, \zeta, p)$-*FTL algorithm, which for* $\varepsilon, \zeta > 0$ *and* $p \in (0, \frac{2}{3})$ *uses the actions of FTL run on the functions* $\tilde{\ell}_t(\boldsymbol{x}) = \left\|\frac{\boldsymbol{b}_t^2 + \varepsilon_t^2 \mathbf{1}_d}{\boldsymbol{x}} + (\boldsymbol{g}_t^2 + \zeta_t^2 \mathbf{1}_d) \odot \boldsymbol{x}\right\|_1$, *where* $\varepsilon_t^2 = \varepsilon^2(t+1)^{-p}, \zeta_t^2 = \zeta^2(t+1)^{-p}$ *for* $t \geq 0$ *and* $\boldsymbol{b}_0 = \boldsymbol{g}_0 = \mathbf{0}_d$, *to determine* $\boldsymbol{x}_t$, *has regret*

$$\mathbf{R}_T \leq C_p \sum_{j=1}^{d} \min\left\{\left(\frac{\varepsilon^2}{\boldsymbol{x}_j^*} + \zeta^2 \boldsymbol{x}_j^*\right) T^{1-p}, \sqrt{\zeta^2 \boldsymbol{b}_{j,1:T}^2 + \varepsilon^2 \boldsymbol{g}_{j,1:T}^2} T^{\frac{1-p}{2}} + 2\varepsilon\zeta T^{1-p}\right\}$$

$$+ C_p\left(\frac{D+\varepsilon}{\zeta^3}G^4 + \frac{G+\zeta}{\varepsilon^3}D^4\right)T^{\frac{3}{2}p} + C_p(D\zeta + G\varepsilon + \varepsilon\zeta)d$$

*for any* $\boldsymbol{x} > 0$ *and some constant* $C_p$ *depending only on* $p$.

*Proof.* Define $\tilde{\boldsymbol{b}}_t^2 = \boldsymbol{b}_t^2 + \varepsilon_t^2 \mathbf{1}_d, \tilde{\boldsymbol{g}}_t^2 = \boldsymbol{g}_t^2 + \zeta_t^2 \mathbf{1}_d$ and note that FTL run on the modified functions $\tilde{\ell}_t'(\boldsymbol{x}) = \left\|\frac{\tilde{\boldsymbol{g}}_t^2 \odot \boldsymbol{x}^2}{2} - \tilde{\boldsymbol{b}}_t^2 \odot \log(\boldsymbol{x})\right\|_1$ plays the exact same actions $\boldsymbol{x}_t^2 = \frac{\tilde{\boldsymbol{b}}_{0:t-1}^2}{\tilde{\boldsymbol{g}}_{0:t-1}^2}$ as FTL run $\tilde{\ell}_t$. Since both sequences of loss functions are separable across coordinates, we consider $d$ per-coordinate problems, with loss functions of form $\tilde{\ell}_t(x) = \frac{\tilde{b}_t^2}{x} + \tilde{g}_t^2 x$ and $\tilde{\ell}_t'(x) = \frac{\tilde{g}_t^2 x^2}{2} - \tilde{b}_t^2 \log x$. We have that

$$|\nabla_t| = \left|\tilde{g}_t^2 - \frac{\tilde{b}_t^2}{x_t^2}\right| = \frac{|\tilde{g}_t^2 x_t^2 - \tilde{b}_t^2|}{x_t^2} \quad |\nabla_t'| = \left|\tilde{g}_t^2 x_t - \frac{\tilde{b}_t^2}{x_t}\right| = \frac{|\tilde{g}_t^2 x_t^2 - \tilde{b}_t^2|}{x_t} \quad \partial_{xx}\tilde{\ell}_t' = \tilde{g}_t^2 + \frac{\tilde{b}_t^2}{x^2} \geq \tilde{g}_t^2$$

so by Theorem B.1 and substituting the action $x_t^2 = \frac{\tilde{b}_{0:t-1}^2}{\tilde{g}_{0:t-1}^2}$ we have per-coordinate regret

$$\sum_{t=1}^{T} \tilde{\ell}_t(x_t) - \tilde{\ell}_t(x^*) \leq 2\sum_{t=1}^{T} \frac{|\nabla_t||\nabla_t'|}{\tilde{g}_{1:t}^2} = 2\sum_{t=1}^{T} \frac{|\tilde{g}_t^2 x_t^2 - \tilde{b}_t^2|^2}{x_t^3 \tilde{g}_{1:t}^2}$$

$$\leq 2\sum_{t=1}^{T} \frac{\tilde{g}_t^4 x_t}{\tilde{g}_{1:t}^2} + \frac{\tilde{b}_t^4}{x_t^3 \tilde{g}_{1:t}^2}$$

$$= 2\sum_{t=1}^{T} \frac{\tilde{g}_t^4 \sqrt{\tilde{b}_{0:t-1}^2}}{\tilde{g}_{1:t}^2 \sqrt{\tilde{g}_{0:t-1}^2}} + \frac{\tilde{b}_t^4}{\tilde{g}_{1:t}^2 \left(\frac{\tilde{b}_{0:t-1}^2}{\tilde{g}_{0:t-1}^2}\right)^{\frac{3}{2}}}$$

$$\leq 2\sum_{t=1}^{T} \frac{\tilde{g}_t^4 \sqrt{\tilde{b}_{0:t-1}^2}}{\tilde{g}_{1:t}^2 \sqrt{\tilde{g}_{0:t-1}^2}} + \frac{\tilde{b}_t^4 \sqrt{2\tilde{g}_{1:t}^2}}{(\tilde{b}_{0:t-1}^2)^{\frac{3}{2}}} + \frac{\tilde{b}_t^4 \tilde{g}_0^3 \sqrt{2}}{\tilde{g}_{1:t}^2 (\tilde{b}_{0:t-1}^2)^{\frac{3}{2}}}$$

Taking the summation over the coordinates yields

$$\sum_{t=1}^{T} \tilde{\ell}_t(\boldsymbol{x}_t) - \tilde{\ell}_t(\boldsymbol{x}^*)$$

$$\leq 4\sum_{t=1}^{T} \left(\frac{(D+\varepsilon)(\|\boldsymbol{g}_t^2\|_2^2 + \zeta_t^4 d)}{\zeta_{1:t}^2 \sqrt{2\zeta_{0:t-1}^2}} + \frac{(G+\zeta)(\|\boldsymbol{b}_t^2\|_2^2 + \varepsilon_t^4 d)}{(\varepsilon_{0:t-1}^2)^{\frac{3}{2}}} + \frac{(\|\boldsymbol{b}_t^2\|_2^2 + \varepsilon_t^4 d)\zeta^3}{\tilde{\zeta}_{0:t-1}^2 (\tilde{\varepsilon}_{0:t-1}^2)^{\frac{3}{2}}}\right)\sqrt{2t}$$

$$\leq 4\sum_{t=1}^{T} \left(\frac{(D+\varepsilon)(G^4 + \zeta_t^4 d)}{(\mathfrak{c}_p\zeta^2 t^{1-p})^{\frac{3}{2}}\sqrt{2}} + \frac{(G+\zeta)(D^4 + \varepsilon_t^4 d)}{(\mathfrak{c}_p\varepsilon^2 t^{1-p})^{\frac{3}{2}}} + \frac{(D^4 + \varepsilon_t^4 d)\zeta}{\varepsilon^3(\mathfrak{c}_p t^{1-p})^{\frac{5}{2}}}\right)\sqrt{2t}$$

$$\leq 4\sqrt{2}\frac{1 + \frac{1}{\mathfrak{c}_p}}{\mathfrak{c}_p^{\frac{3}{2}}}\sum_{t=1}^{T} \left(\frac{D+\varepsilon}{\zeta^3}G^4 + \frac{G+\zeta}{\varepsilon^3}D^4\right)t^{\frac{3}{2}p-1} + \frac{D\zeta + G\varepsilon + 2\varepsilon\zeta}{t^{1+\frac{p}{2}}}d$$

$$\leq C_{p,1}\left(\frac{D+\varepsilon}{\zeta^3}G^4 + \frac{G+\zeta}{\varepsilon^3}D^4\right)T^{\frac{3}{2}p} + C_{p,2}(D\zeta + G\varepsilon + 2\varepsilon\zeta)d$$

for $C_{p,1} = 4\bar{c}_{1-\frac{3}{2}p}\sqrt{2}\left(1 + \frac{1}{\underline{c}_p}\right)/\underline{c}_p^{3/2}$ and $C_{p,2} = 4\sqrt{2}\left(1 + \frac{1}{\underline{c}_p}\right)\sum_{t=1}^{\infty}\frac{1}{t^{1+\frac{p}{2}}}/\underline{c}_p^{3/2}$. Thus we have

$$\sum_{t=1}^{T}\ell_t(\boldsymbol{x}_t) \leq \sum_{t=1}^{T}\tilde{\ell}_t(\boldsymbol{x}_t)$$

$$\leq \min_{\boldsymbol{x}^*>0} C_{p,1}\left(\frac{D+\varepsilon}{\zeta^3}G^4 + \frac{G+\zeta}{\varepsilon^3}D^4\right)T^{\frac{3}{2}p} + C_{p,2}(D\zeta + G\varepsilon + 2\varepsilon\zeta)d + \sum_{t=1}^{T}\tilde{\ell}_t(\boldsymbol{x}^*)$$

$$= C_{p,1}\left(\frac{D+\varepsilon}{\zeta^3}G^4 + \frac{G+\zeta}{\varepsilon^3}D^4\right)T^{\frac{3}{2}p} + C_{p,2}(D\zeta + G\varepsilon + 2\varepsilon\zeta)d$$

$$+ \min_{\boldsymbol{x}^*>0}\sum_{t=1}^{T}\left\|\frac{\boldsymbol{b}_t^2 + \varepsilon_t^2\mathbf{1}_d}{\boldsymbol{x}^*} + (\boldsymbol{g}_t^2 + \zeta_t^2\mathbf{1}_d)\odot\boldsymbol{x}^*\right\|_1$$

$$\leq C_{p,1}\left(\frac{D+\varepsilon}{\zeta^3}G^4 + \frac{G+\zeta}{\varepsilon^3}D^4\right)T^{\frac{3}{2}p} + C_{p,2}(D\zeta + G\varepsilon + 2\varepsilon\zeta)d$$

$$\min_{\boldsymbol{x}^*>0}\bar{c}_pT^{1-p}\sum_{j=1}^{d}\frac{\varepsilon^2}{\boldsymbol{x}_j^*} + \zeta^2\boldsymbol{x}_j^* + \sum_{t=1}^{T}\ell_t(\boldsymbol{x}^*)$$

Separating again per-coordinate we have that

$$\sum_{t=1}^{T}\frac{\tilde{b}_t^2}{x^*} + \tilde{g}_t^2x^* \leq \bar{c}_pT^{1-p}\frac{\varepsilon^2}{x^*} + \zeta^2x^* + \sum_{t=1}^{T}\ell_t(x^*)$$

However, substituting $x^* = \sqrt{\frac{\tilde{b}_{1:T}^2}{\tilde{g}_{1:T}}}$ also yields

$$\min_{x^*>0}\sum_{t=1}^{T}\frac{\tilde{b}_t^2}{x^*} + \tilde{g}_t^2x^* \leq 2\sqrt{\tilde{b}_{1:T}^2\tilde{g}_{1:T}^2}$$

$$\leq 2\sqrt{\bar{c}_p\left(\zeta^2b_{1:T}^2 + \varepsilon^2g_{1:T}^2\right)}T^{\frac{1-p}{2}} + 2\bar{c}_p\varepsilon\zeta T^{1-p} + \min_{x^*>0}\sum_{t=1}^{T}\ell_t(x^*)$$

completing the proof. $\qquad\square$

**Theorem D.1.** *Let $\Theta$ be a bounded convex subset of $\mathbb{R}^d$, let $\mathcal{D} \subset \mathbb{R}^{d \times d}$ be the set of positive definite diagonal matrices, and let each task $t \in [T]$ consist of a sequence of $m$ convex Lipschitz loss functions $\ell_{t,i} : \Theta \mapsto \mathbb{R}$. Suppose for each task $t$ we run the iteration in Equation 5 setting $\phi = \frac{1}{t-1} \boldsymbol{\theta}^*_{1:t-1}$ and setting $\boldsymbol{H} = \mathrm{Diag}(\boldsymbol{\eta}_t)$ via Equation 6 for $\varepsilon = 1, \zeta = \sqrt{m}$, and $p = \frac{2}{5}$. Then we achieve*

$$\bar{\mathbf{R}}_T \leq \bar{\mathbf{U}}_T = \min_{\substack{\boldsymbol{\phi} \in \Theta \\ \boldsymbol{H} \in \mathcal{D}}} \tilde{\mathcal{O}} \left( \sum_{j=1}^d \min \left\{ \frac{\frac{1}{\boldsymbol{H}_{jj}} + \boldsymbol{H}_{jj}}{T^{\frac{2}{5}}}, \frac{1}{\sqrt[5]{T}} \right\} \right) \sqrt{m} + \frac{1}{T} \sum_{t=1}^T \frac{\|\boldsymbol{\theta}^*_t - \boldsymbol{\phi}\|^2_{\boldsymbol{H}^{-1}}}{2} + \sum_{i=1}^m \|\boldsymbol{\nabla}_{t,i}\|^2_{\boldsymbol{H}}$$

*Proof.* Define $\boldsymbol{b}^2_t = \frac{1}{2}(\boldsymbol{\theta}^*_t - \boldsymbol{\phi}_t)^2$ and $\boldsymbol{g}^2_t = \boldsymbol{\nabla}^2_{1:m}$. Then applying Proposition D.1 yields

$$
\begin{aligned}
\bar{\mathbf{U}}_T T &= \sum_{t=1}^T \frac{\|\boldsymbol{\theta}^*_t - \boldsymbol{\phi}_t\|^2_{\boldsymbol{\eta}_t^{-1}}}{2} + \sum_{i=1}^m \|\boldsymbol{\nabla}_{t,i}\|^2_{\boldsymbol{\eta}_t} \\
&= \sum_{t=1}^T \left\| \frac{(\boldsymbol{\theta}^*_t - \boldsymbol{\phi}_t)^2}{2\boldsymbol{\eta}_t} + \boldsymbol{\eta}_t \odot \boldsymbol{\nabla}^2_{t,1:m} \right\|_1 \\
&\leq \min_{\boldsymbol{\eta} > 0} \sum_{t=1}^T \left\| \frac{(\boldsymbol{\theta}^*_t - \boldsymbol{\phi}_t)^2}{2\boldsymbol{\eta}} + \boldsymbol{\eta} \odot \boldsymbol{\nabla}^2_{t,1:m} \right\|_1 \\
&\quad + C_p \sum_{j=1}^d \min \left\{ \left( \frac{\varepsilon^2}{\boldsymbol{\eta}_j} + \zeta^2 \boldsymbol{\eta}_j \right) T^{1-p}, \sqrt{\zeta^2 \boldsymbol{b}^2_{j,1:T} + \varepsilon^2 \boldsymbol{g}^2_{j,1:T}} T^{\frac{1-p}{2}} + 2\varepsilon\zeta T^{1-p} \right\} \\
&\quad + C_p \left( \frac{D + \varepsilon}{\zeta^3} G^4 m^2 + \frac{G\sqrt{m} + \zeta}{\varepsilon^3} D^4 \right) T^{\frac{3}{2}p} + C_p (D\zeta + G\sqrt{m}\varepsilon + \varepsilon\zeta)d \\
&\leq \min_{\substack{\boldsymbol{\phi} \in \Theta \\ \boldsymbol{\eta} > 0}} \sum_{t=1}^T \frac{\|\boldsymbol{\theta}^*_t - \boldsymbol{\phi}\|^2_{\boldsymbol{\eta}^{-1}}}{2} + \sum_{i=1}^{m_t} \|\boldsymbol{\nabla}_{t,i}\|^2_{\boldsymbol{\eta}} + \frac{D^2_\infty}{2} \|\boldsymbol{\eta}^{-1}\|_1 (1 + \log T) \\
&\quad + C_p \sum_{j=1}^d \min \left\{ \left( \frac{\varepsilon^2}{\boldsymbol{\eta}_j} + \zeta^2 \boldsymbol{\eta}_j \right) T^{1-p}, \sqrt{\zeta^2 \boldsymbol{b}^2_{j,1:T} + \varepsilon^2 \boldsymbol{g}^2_{j,1:T}} T^{\frac{1-p}{2}} + 2\varepsilon\zeta T^{1-p} \right\} \\
&\quad + C_p \left( \frac{D + \varepsilon}{\zeta^3} G^4 m^2 + \frac{G\sqrt{m} + \zeta}{\varepsilon^3} D^4 \right) T^{\frac{3}{2}p} + C_p (D\zeta + G\sqrt{m}\varepsilon + \varepsilon\zeta)d
\end{aligned}
$$

Substituting $\boldsymbol{\eta} + \frac{\mathbf{1}_d}{\sqrt{mT}}$ for the optimum and the values of $\varepsilon, \zeta, p$ completes the proof. $\qquad \square$

**Proposition D.2.** *Let $\{\ell_t : \mathbb{R}_+ \mapsto \mathbb{R}\}_{t \geq 1}$ be of form $\ell_t(\boldsymbol{X}) = \mathrm{Tr}(\boldsymbol{X}^{-1}\boldsymbol{B}_t^2) + \mathrm{Tr}(\boldsymbol{X}\boldsymbol{G}_t^2)$ for adversarially chosen $\boldsymbol{B}_t, \boldsymbol{G}_t$ satisfying $\|\boldsymbol{B}_t\|_2 \leq \sigma_B, \|\boldsymbol{G}_t\|_2 \leq \sigma_G\sqrt{m}$ for $m \geq 1$. Then the $(\varepsilon, \zeta)$-FTL algorithm, which for $\varepsilon, \zeta > 0$ uses the actions of FTL on the alternate function sequence $\tilde{\ell}_t(\boldsymbol{X}) = \mathrm{Tr}((\boldsymbol{B}^2 + \varepsilon^2\boldsymbol{I}_d)\boldsymbol{X}^{-1}) + \mathrm{Tr}((\boldsymbol{G}^2 + \zeta^2\boldsymbol{I}_d)\boldsymbol{X})$, achieves regret*

$$\mathbf{R}_T \leq \frac{C_\sigma m^2}{\varepsilon^4 \zeta^3}(1 + \log T) + ((1 + \sigma_G^2)\varepsilon\sqrt{m} + (1 + \sigma_B^2)\zeta)T$$

*for constant $C_\sigma$ depending only on $\sigma_B, \sigma_G$.*

*Proof.* Define $\tilde{\boldsymbol{B}}_t^2 = \boldsymbol{B}_t^2 + \varepsilon^2\boldsymbol{I}_d, \tilde{\boldsymbol{G}}_t^2 = \boldsymbol{G}_t^2 + \zeta^2\boldsymbol{I}_d$ and note that FTL run on modified functions $\tilde{\ell}'_t(\boldsymbol{X}) = \frac{1}{2}\mathrm{Tr}(\tilde{\boldsymbol{B}}_t^{-2}\boldsymbol{X}\tilde{\boldsymbol{G}}_t^2\boldsymbol{X}) - \log\det\boldsymbol{X}$ has the same solution $\tilde{\boldsymbol{B}}_{1:T}^2 = \boldsymbol{X}\tilde{\boldsymbol{G}}_{1:T}^2\boldsymbol{X}$.

$$\|\nabla_{\boldsymbol{X}}\tilde{\ell}_t(\boldsymbol{X})\|_2 = \|\tilde{\boldsymbol{G}}_t^2 - \boldsymbol{X}^{-1}\tilde{\boldsymbol{B}}_t^2\boldsymbol{X}^{-1}\|_2 \leq \|\tilde{\boldsymbol{G}}_t\|_2^2 + \|\boldsymbol{X}^{-1}\|_2^2\|\tilde{\boldsymbol{B}}_t\|_2^2 \leq \frac{\sigma_B^2}{\varepsilon^2} + m\sigma_G^2 + \zeta^2$$

$$\|\nabla_{\boldsymbol{X}}\tilde{\ell}'_t(\boldsymbol{X})\|_2 = \|\tilde{\boldsymbol{G}}_t^2\boldsymbol{X}\tilde{\boldsymbol{B}}_t^{-2} - \boldsymbol{X}^{-1}\|_2 \leq \|\tilde{\boldsymbol{G}}_t\|_2^2\|\boldsymbol{X}\|_2\|\tilde{\boldsymbol{B}}_t^{-1}\|_2^2 + \|\boldsymbol{X}^{-1}\|_2$$
$$\leq \frac{(m\sigma_G^2 + \zeta^2)\sqrt{\sigma_B^2 + \varepsilon^2}}{\varepsilon^2\zeta} + \frac{\sqrt{m\sigma_G^2 + \zeta^2}}{\zeta}$$

Since by Claim D.4 $-\log\det|\boldsymbol{X}|$ is $\frac{\zeta^2}{\sigma_B^2 + \varepsilon^2}$-strongly-convex we have by Theorem B.1 that

$$\sum_{t=1}^{T}\tilde{\ell}_t(\boldsymbol{X}_t) - \tilde{\ell}_t(\boldsymbol{X}^*) \leq \frac{C_\sigma m^2}{\varepsilon^4 \zeta^3}(1 + \log T)$$

for some $C_\sigma$ depending on $\sigma_B^2, \sigma_G^2$. Therefore

$$\sum_{t=1}^{T}\ell_t(\boldsymbol{X}) \leq \sum_{t=1}^{T}\tilde{\ell}_t(\boldsymbol{X})$$

$$\leq \frac{C_\sigma m^2}{\varepsilon^4 \zeta^3}(1 + \log T) + \min_{\boldsymbol{X} \succ 0}\sum_{t=1}^{T}\tilde{\ell}_t(\boldsymbol{X})$$

$$\leq \frac{C_\sigma m^2}{\varepsilon^4 \zeta^3}(1 + \log T) + \min_{\boldsymbol{X} \succ 0}\varepsilon^2 T\,\mathrm{Tr}(\boldsymbol{X}^{-1}) + \zeta^2 T\,\mathrm{Tr}(\boldsymbol{X}) + \sum_{t=1}^{T}\ell_t(\boldsymbol{X})$$

$$\leq \frac{C_\sigma m^2}{\varepsilon^4 \zeta^3}(1 + \log T) + (1 + \sigma_G^2)\varepsilon T\sqrt{m} + \min_{\boldsymbol{X} \succ 0}\zeta^2 T\,\mathrm{Tr}(\boldsymbol{X}) + \sum_{t=1}^{T}\ell_t(\boldsymbol{X})$$

$$\leq \frac{C_\sigma m^2}{\varepsilon^4 \zeta^3}(1 + \log T) + ((1 + \sigma_G^2)\varepsilon\sqrt{m} + (1 + \sigma_B^2)\zeta)T + \min_{\boldsymbol{X} \succ 0}\sum_{t=1}^{T}\ell_t(\boldsymbol{X})$$

$\square$

**Theorem D.2.** *Let $\Theta$ be a bounded convex subset of $\mathbb{R}^d$ and let each task $t \in [T]$ consist of a sequence of $m$ convex Lipschitz loss functions $\ell_{t,i} : \Theta \mapsto \mathbb{R}$. Suppose for each task $t$ we run the iteration in Equation 5 with $\phi = \frac{1}{t-1}\boldsymbol{\theta}^*_{1:t-1}$ and $\boldsymbol{H}$ the unique positive definite solution of $\boldsymbol{B}_t^2 = \boldsymbol{H}\boldsymbol{G}_t^2\boldsymbol{H}$ for*

$$\boldsymbol{B}_t^2 = t\varepsilon^2 \boldsymbol{I}_d + \sum_{s<t}(\boldsymbol{\theta}^*_s - \boldsymbol{\phi}_s)(\boldsymbol{\theta}^*_s - \boldsymbol{\phi}_s)^T \qquad and \qquad \boldsymbol{G}_t^2 = t\varepsilon^2 \boldsymbol{I}_d + \sum_{s<t}\sum_{i=1}^{m}\boldsymbol{\nabla}_{s,i}\boldsymbol{\nabla}_{s,i}^T$$

*for $\varepsilon = 1/\sqrt[8]{T}$ and $\zeta = \sqrt{m}/\sqrt[8]{T}$. Then we achieve*

$$\bar{\mathbf{R}}_T \leq \bar{\mathbf{U}}_T = \tilde{\mathcal{O}}\left(\frac{1}{\sqrt[8]{T}}\right)\sqrt{m} + \min_{\substack{\boldsymbol{\phi}\in\Theta \\ \boldsymbol{H}\succ 0}} \frac{2\lambda_1^2(\boldsymbol{H})}{\lambda_d(\boldsymbol{H})}\frac{1+\log T}{T} + \sum_{t=1}^{T}\frac{\|\boldsymbol{\theta}^*_t - \boldsymbol{\phi}^*\|_{\boldsymbol{H}^{-1}}^2}{2} + \sum_{i=1}^{m}\|\boldsymbol{\nabla}_{t,i}\|_{\boldsymbol{H}}^2$$

*Proof.* Let $D$ and $G$ be the diameter of $\Theta$ and Lipschitz bound on the losses, respectively. Then applying Proposition D.2 yields

$$
\begin{aligned}
\bar{\mathbf{U}}_T\, T &= \sum_{t=1}^{T}\frac{\|\boldsymbol{\theta}^*_t - \boldsymbol{\phi}_t\|_{\boldsymbol{H}_t^{-1}}^2}{2} + \sum_{i=1}^{m}\|\boldsymbol{\nabla}_{t,i}\|_{\boldsymbol{H}_t}^2 \\
&= \sum_{t=1}^{T}\frac{1}{2}\operatorname{Tr}\left(\boldsymbol{H}_t^{-1}(\boldsymbol{\theta}^*_t - \boldsymbol{\phi}_t)(\boldsymbol{\theta}^*_t - \boldsymbol{\phi}_t)^T\right) + \operatorname{Tr}\left(\boldsymbol{H}_t\sum_{i=1}^{m}\boldsymbol{\nabla}_{t,i}\boldsymbol{\nabla}_{t,i}^T\right) \\
&\leq \min_{\boldsymbol{H}\succ 0}\sum_{t=1}^{T}\frac{1}{2}\operatorname{Tr}\left(\boldsymbol{H}^{-1}(\boldsymbol{\theta}^*_t - \boldsymbol{\phi}_t)(\boldsymbol{\theta}^*_t - \boldsymbol{\phi}_t)^T\right) + \operatorname{Tr}\left(\boldsymbol{H}\sum_{i=1}^{m}\boldsymbol{\nabla}_{t,i}\boldsymbol{\nabla}_{t,i}^T\right) \\
&\quad + \frac{C_\sigma m^2}{\varepsilon^4\zeta^3}(1+\log T) + ((1+G^2)\varepsilon\sqrt{m} + (1+D^2)\zeta)T \\
&= \min_{\boldsymbol{H}\succ 0}\sum_{t=1}^{T}\frac{\|\boldsymbol{\theta}^*_t - \boldsymbol{\phi}_t\|_{\boldsymbol{H}^{-1}}^2}{2} + \operatorname{Tr}\left(\boldsymbol{H}\sum_{i=1}^{m}\boldsymbol{\nabla}_{t,i}\boldsymbol{\nabla}_{t,i}^T\right) \\
&\quad + \frac{C_\sigma m^2}{\varepsilon^4\zeta^3}(1+\log T) + ((1+G^2)\varepsilon\sqrt{m} + (1+D^2)\zeta)T \\
&\leq \min_{\substack{\boldsymbol{\phi}\in\Theta \\ \boldsymbol{H}\succ 0}} \frac{2\lambda_1^2(\boldsymbol{H})}{\lambda_d(\boldsymbol{H})}\sum_{t=1}^{T}\frac{1}{t} + \sum_{t=1}^{T}\frac{\|\boldsymbol{\theta}^*_t - \boldsymbol{\phi}^*\|_{\boldsymbol{H}^{-1}}^2}{2} + \sum_{i=1}^{m}\|\boldsymbol{\nabla}_{t,i}\|_{\boldsymbol{H}}^2 \\
&\quad + \frac{C_\sigma m^2}{\varepsilon^4\zeta^3}(1+\log T) + ((1+G^2)\varepsilon\sqrt{m} + (1+D^2)\zeta)T \\
&= \min_{\substack{\boldsymbol{\phi}\in\Theta \\ \boldsymbol{H}\succ 0}} \frac{2\lambda_1^2(\boldsymbol{H})}{\lambda_d(\boldsymbol{H})}\sum_{t=1}^{T}\frac{1}{t} + \sum_{t=1}^{T}\frac{\|\boldsymbol{\theta}^*_t - \boldsymbol{\phi}^*\|_{\boldsymbol{H}^{-1}}^2}{2} + \sum_{i=1}^{m}\|\boldsymbol{\nabla}_{t,i}\|_{\boldsymbol{H}}^2 \\
&\quad + \frac{C_\sigma m^2}{\varepsilon^4\zeta^3}(1+\log T) + ((1+G^2)\varepsilon\sqrt{m} + (1+D^2)\zeta)T
\end{aligned}
$$

$\square$

# E  Online-to-Batch Conversion for Task-Averaged Regret

**Theorem E.1.** *Let $\mathcal{Q}$ be a distribution over distributions $\mathcal{P}$ over convex loss functions $\ell : \Theta \mapsto [0, 1]$. A sequence of sequences of loss functions $\{\ell_{t,i}\}_{t\in[T],i\in[m]}$ is generated by drawing $m$ loss functions i.i.d. from each in a sequence of distributions $\{\mathcal{P}_t\}_{t\in[T]}$ themselves drawn i.i.d. from $\mathcal{Q}$. If such a sequence is given to an meta-learning algorithm with task-averaged regret bound $\bar{\mathbf{R}}_T$ that has states $\{s_t\}_{t\in[T]}$ at the beginning of each task $t$ then we have w.p. $1 - \delta$ for any $\theta^* \in \Theta$ that*

$$\mathop{\mathbb{E}}_{t\sim\mathcal{U}[T]} \mathop{\mathbb{E}}_{\mathcal{P}\sim\mathcal{Q}} \mathop{\mathbb{E}}_{\mathcal{P}^m} \mathop{\mathbb{E}}_{\ell\sim\mathcal{P}} \ell(\bar{\theta}) \leq \mathop{\mathbb{E}}_{\mathcal{P}\sim\mathcal{Q}} \mathop{\mathbb{E}}_{\ell\sim\mathcal{P}} \ell(\theta^*) + \frac{\bar{\mathbf{R}}_T}{m} + \sqrt{\frac{8}{T}\log\frac{1}{\delta}}$$

*where $\bar{\theta} = \frac{1}{m}\theta_{1:m}$ is generated by randomly sampling $t \in \mathcal{U}[T]$, running the online algorithm with state $s_t$, and averaging the actions $\{\theta_i\}_{i\in[m]}$. If on each task the meta-learning algorithm runs an online algorithm with regret upper bound $\mathbf{U}_m(s_t)$ a convex, nonnegative, and $B\sqrt{m}$-bounded function of the state $s_t \in \mathcal{X}$, where $\mathcal{X}$ is a convex Euclidean subset, and the total regret upper bound is $\bar{\mathbf{U}}_T$, then we also have the bound*

$$\mathop{\mathbb{E}}_{\mathcal{P}\sim\mathcal{Q}} \mathop{\mathbb{E}}_{\mathcal{P}^m} \mathop{\mathbb{E}}_{\ell\sim\mathcal{P}} \ell(\bar{\theta}) \leq \mathop{\mathbb{E}}_{\mathcal{P}\sim\mathcal{Q}} \mathop{\mathbb{E}}_{\ell\sim\mathcal{P}} \ell(\theta^*) + \frac{\bar{\mathbf{U}}_T}{m} + B\sqrt{\frac{8}{mT}\log\frac{1}{\delta}}$$

*where $\bar{\theta} = \frac{1}{m}\theta_{1:m}$ is generated by running the online algorithm with state $\bar{s} = \frac{1}{T}s_{1:T}$ and averaging the actions $\{\theta_i\}_{i\in[m]}$.*

*Proof.* For the second inequality, applying Proposition A.1, Jensen's inequality, and Proposition A.2 yields

$$\mathop{\mathbb{E}}_{\mathcal{P}\sim\mathcal{Q}} \mathop{\mathbb{E}}_{\mathcal{P}^m} \mathop{\mathbb{E}}_{\ell\sim\mathcal{P}} \ell(\bar{\theta}) \leq \mathop{\mathbb{E}}_{\mathcal{P}\sim\mathcal{Q}} \left( \mathop{\mathbb{E}}_{\ell\sim\mathcal{P}} \ell(\theta^*) + \frac{\mathbf{U}_m(\bar{s})}{m} \right)$$

$$\leq \mathop{\mathbb{E}}_{\mathcal{P}\sim\mathcal{Q}} \mathop{\mathbb{E}}_{\ell\sim\mathcal{P}} \ell(\theta^*) + \frac{1}{T}\sum_{t=1}^{T} \mathop{\mathbb{E}}_{\mathcal{P}\sim\mathcal{Q}} \left( \frac{\mathbf{U}_m(s_t)}{m} \right)$$

$$= \mathop{\mathbb{E}}_{\mathcal{P}\sim\mathcal{Q}} \mathop{\mathbb{E}}_{\ell\sim\mathcal{P}} \ell(\theta^*) + \frac{2B}{T\sqrt{m}}\sum_{t=1}^{T} \mathop{\mathbb{E}}_{\mathcal{P}\sim\mathcal{Q}} \left( \frac{\mathbf{U}_m(s_t)}{2B\sqrt{m}} + \frac{\sqrt{m}}{2B} \right) - 1$$

$$\leq \mathop{\mathbb{E}}_{\mathcal{P}\sim\mathcal{Q}} \mathop{\mathbb{E}}_{\ell\sim\mathcal{P}} \ell(\theta^*) + \frac{\bar{\mathbf{U}}_T}{m} + B\sqrt{\frac{8}{mT}\log\frac{1}{\delta}}$$

The first inequality follows similarly except using $\mathbf{R}_m$ instead of $\mathbf{U}_m$, linearity of expectation instead of Jensen's inequality, 1 instead of $B$, and $\bar{\mathbf{R}}_T$ instead of $\bar{\mathbf{U}}_T$. $\qquad\square$

Note that since regret-upper-bounds are nonnegative one can easily replace 8 by 2 in the second inequality by simply multiplying and dividing by $B\sqrt{m}$ in the third line of the above proof.

**Claim E.1.** *In the setup of Theorem E.1, let $\theta_t^* \in \arg\min_{\theta\in\Theta} \sum_{i=1}^{m} \ell_{t,i}(\theta)$ and define the quantities $V_{\mathcal{Q}}^2 = \arg\min_{\phi\in\Theta} \mathbb{E}_{\mathcal{P}\sim\mathcal{Q}} \mathbb{E}_{\mathcal{P}^m} \|\theta^* - \phi\|_2^2$ and $D$ the $\ell_2$-radius of $\Theta$. Then w.p. $1 - \delta$ we have*

$$V^2 = \min_{\phi\in\Theta} \frac{1}{T}\sum_{t=1}^{T} \|\theta_t^* - \phi\|_2^2 \leq \mathcal{O}\left( V_{\mathcal{Q}}^2 + \frac{D^2}{T}\log\frac{1}{\delta} \right)$$

*Proof.* Define $\hat{\phi} = \arg\min_{\phi\in\Theta} \sum_{t=1}^{T} \|\theta_t^* - \phi\|_2^2$ and $\phi^* = \arg\min_{\phi\in\Theta} \mathbb{E}_{\mathcal{P}\sim\mathcal{Q}} \mathbb{E}_{\mathcal{P}^m} \|\theta^* - \phi\|_2^2$. Then by a multiplicative Chernoff's inequality w.p. at least $1 - \delta$ we have

$$TV^2 = \sum_{t=1}^{T} \|\theta_t^* - \hat{\phi}\|_2^2 \leq \sum_{t=1}^{T} \|\theta_t^* - \phi^*\|_2^2 \leq \left( 1 + \max\left\{ 1, \frac{3D^2}{V_{\mathcal{Q}}^2 T}\log\frac{1}{\delta} \right\} \right) T \mathop{\mathbb{E}}_{\mathcal{P}\sim\mathcal{Q}} \mathop{\mathbb{E}}_{\mathcal{P}^m} \|\theta^* - \phi^*\|_2^2$$

$$\leq 2TV_{\mathcal{Q}}^2 + 3D^2 \log\frac{1}{\delta}$$

$$\square$$

**Corollary E.1.** *Under the assumptions of Theorems 3.2 and 5.1, if the loss functions are Lipschitz and we use Algorithm 1 with $\eta_t$ also learned, using $\varepsilon$-EWOO as in Theorem 3.2 for $\varepsilon = 1/\sqrt[4]{mT} + 1/\sqrt{m}$, and set the initialization using $\phi_{t+1} = \frac{1}{t}\sum_{s \leq t}\theta_s^*$, then w.p. $1 - \delta$ we have*

$$\mathop{\mathbb{E}}_{\mathcal{P}\sim\mathcal{Q}}\mathop{\mathbb{E}}_{\mathcal{P}^m}\ell_{\mathcal{P}}(\bar{\theta}) \leq \mathop{\mathbb{E}}_{\mathcal{P}\sim\mathcal{Q}}\ell_{\mathcal{P}}(\theta^*) + \tilde{\mathcal{O}}\left(\frac{V_{\mathcal{Q}}}{\sqrt{m}} + \min\left\{\frac{\frac{1}{\sqrt{T}} + \frac{1}{\sqrt{m}}}{V_{\mathcal{Q}}m}, \frac{1}{\sqrt[4]{m^3T}} + \frac{1}{m}\right\} + \sqrt{\frac{1}{T}\log\frac{1}{\delta}}\right)$$

*where $V_{\mathcal{Q}}^2 = \min_{\phi\in\Theta}\mathbb{E}_{\mathcal{P}\sim\mathcal{Q}}\mathbb{E}_{\mathcal{P}^m}\|\theta^* - \phi\|_2^2$.*

*Proof.* Substitute Corollary C.3 into Theorem E.1 using the fact the the regret-upper-bounds are $\mathcal{O}(\frac{\sqrt{m}}{\varepsilon})$-bounded. Conclude by applying Claim E.1. $\qquad\square$

**Theorem E.2.** *Let $\mathcal{Q}$ be a distribution over distributions $\mathcal{P}$ over convex losses $\ell : \Theta \mapsto [0,1]$ such that the functions $\ell(\theta) - \ell(\theta^*)$ are $\rho$-self-bounded for some $\rho > 0$ and $\theta^* \in \arg\min_{\theta\in\Theta}\mathbb{E}_{\ell\sim\mathcal{P}}(\theta)$. A sequence of sequences of loss functions $\{\ell_{t,i}\}_{t\in[T],i\in[m]}$ is generated by drawing $m$ loss functions i.i.d. from each in a sequence of distributions $\{\mathcal{P}_t\}_{t\in[T]}$ themselves drawn i.i.d. from $\mathcal{Q}$. If such a sequence is given to an meta-learning algorithm with task-averaged regret bound $\bar{\mathbf{R}}_T$ that has states $\{s_t\}_{t\in[T]}$ at the beginning of each task $t$ then we have w.p. $1 - \delta$ for any $\theta^* \in \Theta$ that*

$$\mathop{\mathbb{E}}_{t\sim\mathcal{U}[T]}\mathop{\mathbb{E}}_{\mathcal{P}\sim\mathcal{Q}}\mathop{\mathbb{E}}_{\ell\sim\mathcal{P}}\ell(\bar{\theta}) \leq \mathop{\mathbb{E}}_{\mathcal{P}\sim\mathcal{Q}}\mathop{\mathbb{E}}_{\ell\sim\mathcal{P}}\ell(\theta^*) + \frac{\bar{\mathbf{R}}_T}{m} + \sqrt{\frac{2\rho}{m}\left(\frac{\bar{\mathbf{R}}_T}{m} + \sqrt{\frac{8}{T}\log\frac{2}{\delta}}\right)\log\frac{2}{\delta}}$$

$$+ \sqrt{\frac{8}{T}\log\frac{2}{\delta}} + \frac{3\rho+2}{m}\log\frac{2}{\delta}$$

*where $\bar{\theta} = \frac{1}{m}\theta_{1:m}$ is generated by randomly sampling $t \in \mathcal{U}[T]$, running the online algorithm with state $s_t$, and averaging the actions $\{\theta_i\}_{i\in[m]}$. If on each task the meta-learning algorithm runs an online algorithm with regret upper bound $\mathbf{U}_m(s_t)$ a convex, nonnegative, and $B\sqrt{m}$-bounded function of the state $s_t \in \mathcal{X}$, where $\mathcal{X}$ is a convex Euclidean subset, and the total regret upper bound is $\bar{\mathbf{U}}_T$, then we also have the bound*

$$\mathop{\mathbb{E}}_{\mathcal{P}\sim\mathcal{Q}}\mathop{\mathbb{E}}_{\ell\sim\mathcal{P}}\ell(\bar{\theta}) \leq \mathop{\mathbb{E}}_{\mathcal{P}\sim\mathcal{Q}}\mathop{\mathbb{E}}_{\ell\sim\mathcal{P}}\ell(\theta^*) + \frac{\bar{\mathbf{U}}_T}{m} + \sqrt{\frac{2\rho}{m}\left(\frac{\bar{\mathbf{U}}_T}{m} + B\sqrt{\frac{8}{mT}\log\frac{2}{\delta}}\right)\log\frac{2}{\delta}}$$

$$+ B\sqrt{\frac{8}{mT}\log\frac{2}{\delta}} + \frac{3\rho+2}{m}\log\frac{2}{\delta}$$

*where $\bar{\theta} = \frac{1}{m}\theta_{1:m}$ is generated by running the online algorithm with state $\bar{s} = \frac{1}{T}s_{1:T}$ and averaging the actions $\{\theta_i\}_{i\in[m]}$.*

*Proof.* By Corollary A.2 and Jensen's inequality we have w.p. $1 - \frac{\delta}{2}$ that

$$\mathop{\mathbb{E}}_{\mathcal{P}\sim\mathcal{Q}}\mathop{\mathbb{E}}_{\ell\sim\mathcal{P}}\ell(\bar{\theta}) \leq \mathop{\mathbb{E}}_{\mathcal{P}\sim\mathcal{Q}}\left(\mathop{\mathbb{E}}_{\ell\sim\mathcal{P}}\ell(\theta^*) + \frac{\mathbf{U}_m(\bar{s})}{m} + \frac{1}{m}\sqrt{2\rho\,\mathbf{U}_m(\bar{s})\log\frac{1}{\delta}} + \frac{3\rho+2}{m}\log\frac{1}{\delta}\right)$$

$$\leq \mathop{\mathbb{E}}_{\mathcal{P}\sim\mathcal{Q}}\mathop{\mathbb{E}}_{\ell\sim\mathcal{P}}\ell(\theta^*) + \frac{1}{T}\sum_{t=1}^{T}\mathop{\mathbb{E}}_{\mathcal{P}\sim\mathcal{Q}}\left(\frac{\mathbf{U}_m(s_t)}{m}\right)$$

$$+ \sqrt{\frac{2\rho}{mT}\sum_{t=1}^{T}\mathop{\mathbb{E}}_{\mathcal{P}\sim\mathcal{Q}}\left(\frac{\mathbf{U}_m(s_t)}{m}\right)\log\frac{2}{\delta}} + \frac{3\rho+2}{m}\log\frac{2}{\delta}$$

As in the proof of Theorem E.1, by Proposition A.2 we further have w.p. $1 - \frac{\delta}{2}$ that

$$\frac{1}{T}\sum_{t=1}^{T}\mathop{\mathbb{E}}_{\mathcal{P}\sim\mathcal{Q}}\left(\frac{\mathbf{U}_m(s_t)}{m}\right) \leq \frac{\bar{\mathbf{U}}_T}{m} + B\sqrt{\frac{8}{mT}\log\frac{2}{\delta}}$$

Substituting the second inequality into the first yields the second bound. The first bound follows similarly except using $\mathbf{R}_m$ instead of $\mathbf{U}_m$, linearity of expectation instead of Jensen's inequality, 1 instead of $B$, and $\bar{\mathbf{R}}_T$ instead of $\bar{\mathbf{U}}_T$. $\qquad\square$

**Theorem E.3.** *Let $\mathcal{Q}$ be a distribution over distributions $\mathcal{P}$ over convex loss functions $\ell : \Theta \mapsto [0,1]$. A sequence of sequences of loss functions $\{\ell_{t,i}\}_{t\in[T],i\in[m]}$ is generated by drawing $m$ loss functions i.i.d. from each in a sequence of distributions $\{\mathcal{P}_t\}_{t\in[T]}$ themselves drawn i.i.d. from $\mathcal{Q}$. If such a sequence is given to an meta-learning algorithm that on each task runs an online algorithm with regret upper bound $\mathbf{U}_m(s_t)$ a nonnegative, $B\sqrt{m}$-bounded, $G$-Lipschitz w.r.t. $\|\cdot\|$, and $\alpha$-strongly-convex w.r.t. $\|\cdot\|$ function of the state $s_t \in \mathcal{X}$ at the beginning of each task $t$, where $\mathcal{X}$ is a convex Euclidean subset, and the total regret upper bound is $\bar{\mathbf{U}}_T$, then we have w.p. $1-\delta$ for any $\theta^* \in \Theta$ that*

$$\mathop{\mathbb{E}}_{\mathcal{P}\sim\mathcal{Q}} \mathop{\mathbb{E}}_{\mathcal{P}^m} \mathop{\mathbb{E}}_{\ell\sim\mathcal{P}} \ell(\bar{\theta}) \leq \mathop{\mathbb{E}}_{\mathcal{P}\sim\mathcal{Q}} \mathop{\mathbb{E}}_{\ell\sim\mathcal{P}} \ell(\theta^*) + \mathcal{L}_T$$

*for*

$$\mathcal{L}_T = \frac{\mathbf{U}^* + \bar{\mathbf{U}}_T}{m} + \frac{4G}{T}\sqrt{\frac{\bar{\mathbf{U}}_T}{\alpha m}\log\frac{8\log T}{\delta}} + \frac{\max\{16G^2, 6\alpha B\sqrt{m}\}}{\alpha m T}\log\frac{8\log T}{\delta}$$

*where $\mathbf{U}^* = \mathbb{E}_{\mathcal{P}\sim\mathcal{Q}}\mathbf{U}_m(s^*)$ for any valid $s^*$ and $\bar{\theta} = \frac{1}{m}\theta_{1:m}$ is generated by running the online algorithm with state $\bar{s} = \frac{1}{T}s_{1:T}$ and averaging the actions $\{\theta_i\}_{i\in[m]}$. If we further assume that the functions $\ell(\theta) - \ell(\theta^*)$ are $\rho$-self-bounded for some $\rho > 0$ and $\theta^* \in \arg\min_{\theta\in\Theta}\mathbb{E}_{\ell\sim\mathcal{P}}(\theta)$ for all $\mathcal{P}$ in the support of $\mathcal{Q}$ then we also have the bound*

$$\mathop{\mathbb{E}}_{\mathcal{P}\sim\mathcal{Q}} \mathop{\mathbb{E}}_{\ell\sim\mathcal{P}} \ell(\bar{\theta}) \leq \mathop{\mathbb{E}}_{\mathcal{P}\sim\mathcal{Q}} \mathop{\mathbb{E}}_{\ell\sim\mathcal{P}} \ell(\theta^*) + \mathcal{L}_T + \sqrt{\frac{2\rho\mathcal{L}_T}{m}\log\frac{2}{\delta}} + \frac{3\rho+2}{m}\log\frac{2}{\delta}$$

*Proof.* Applying Proposition A.1 and Theorem A.4 we have w.p. $1-\frac{\delta}{2}$ that

$$\mathop{\mathbb{E}}_{\mathcal{P}\sim\mathcal{Q}} \mathop{\mathbb{E}}_{\mathcal{P}^m} \mathop{\mathbb{E}}_{\ell\sim\mathcal{P}} \ell(\bar{\theta}) \leq \mathop{\mathbb{E}}_{\mathcal{P}\sim\mathcal{Q}} \left( \mathop{\mathbb{E}}_{\ell\sim\mathcal{P}} \ell(\theta^*) + \frac{\mathbf{U}_m(\bar{s})}{m} \right)$$

$$\leq \mathop{\mathbb{E}}_{\mathcal{P}\sim\mathcal{Q}} \mathop{\mathbb{E}}_{\ell\sim\mathcal{P}} \ell(\theta^*) + \frac{1}{m} \mathop{\mathbb{E}}_{\mathcal{P}\sim\mathcal{Q}} \mathbf{U}_m(s^*) + \frac{\bar{\mathbf{U}}_T}{m}$$

$$+ \frac{4G}{T}\sqrt{\frac{\bar{\mathbf{U}}_T}{\alpha m}\log\frac{8\log T}{\delta}} + \frac{\max\{16G^2, 6\alpha B\sqrt{m}\}}{\alpha m T}\log\frac{8\log T}{\delta}$$

$$\leq \mathop{\mathbb{E}}_{\mathcal{P}\sim\mathcal{Q}} \mathop{\mathbb{E}}_{\ell\sim\mathcal{P}} \ell(\theta^*) + \mathcal{L}_T$$

This yields the first bound since. The second bound follows similarly except for the application of Corollary A.2 in the second step w.p. $1-\frac{\delta}{2}$. $\qquad\square$

**Corollary E.2.** *Under the assumptions of Theorem 5.1 and boundedness of $\Theta$, if the loss functions are $G$-Lipschitz and we use Algorithm 1 running OGD with fixed $\eta = \frac{V_\mathcal{Q}+1/\sqrt{T}}{G\sqrt{m}}$, where we have $V_\mathcal{Q}^2 = \min_{\phi\in\Theta}\mathbb{E}_{\mathcal{P}\sim\mathcal{Q}}\mathbb{E}_{\mathcal{P}^m}\|\theta^* - \phi\|_2^2$, and set the initialization using $\phi_{t+1} = \frac{1}{t}\theta_{1:t}^*$, then w.p. $1-\delta$ we have*

$$\mathop{\mathbb{E}}_{\mathcal{P}\sim\mathcal{Q}} \mathop{\mathbb{E}}_{\mathcal{P}^m} \ell_\mathcal{P}(\bar{\theta}) \leq \mathop{\mathbb{E}}_{\mathcal{P}\sim\mathcal{Q}} \ell_\mathcal{P}(\theta^*) + \tilde{\mathcal{O}}\left( \frac{V_\mathcal{Q}}{\sqrt{m}} + \left( \frac{1}{T} + \frac{1}{\sqrt{mT}} \right) \max\left\{ \log\frac{1}{\delta}, \sqrt{\log\frac{1}{\delta}} \right\} \right)$$

*Proof.* Apply Theorem C.1 with $V_\Phi = V_\mathcal{Q} + 1/\sqrt{T}$, $\mathbf{U}^{\text{sim}} = 0$ (because the learning rate is fixed), and $\mathbf{U}^{\text{init}} = \tilde{\mathcal{O}}\left( \hat{V}\sqrt{m} + 1/\sqrt{T} \right)$ (for $\hat{V}^2 = \min_{\phi\in\Theta}\frac{1}{T}\sum_{t=1}^{T}\|\theta_t^* - \phi\|_2^2$). Substitute the result into Theorem E.3 using the fact that $\mathbf{U}_m$ is $\mathcal{O}\left( \left(\frac{1}{\varepsilon} + \varepsilon\right)\sqrt{m} \right)$-bounded, $\mathcal{O}\left( \frac{\sqrt{m}}{\varepsilon} \right)$-Lipschitz, and $\Omega\left( \frac{\sqrt{m}}{\varepsilon} \right)$-strongly-convex. Conclude by applying Claim E.1 to bound $\hat{V}$. $\qquad\square$

# F  Adapting to Task-Similarity under Parameter Growth

In this appendix we cast the problem of adaptively learning the task-similarity in the framework of Khodak et al. [34]. We do this specifically to show that our basic results extend to approximate meta-updates under quadratic growth. We first provide a generalized version of their Ephemeral method in Algorithm 3. We then state the relevant approximation assumptions and proceed to prove guarantees on the average regret-upper-bound for the case of a fixed task-similarity in Theorem F.1 and for adaptively learning it in Theorem F.2. Then the quadratic-growth results of Khodak et al. [34], specifically Propositions B.1, B.2, and B.3, can be applied directly to show average regret-upper-bound guarantees of the same order as those in the main paper but with additional $o_m(1)$ terms inside the parentheses. Note that our results, especially in the batch-within-online setting, will in general be stronger because we do not incur the $\Delta_{\max}$-error term that is needed to account for the doubling trick in Khodak et al. [34].

---

**Algorithm 3:** Follow-the-Meta-Regularized-Leader (Ephemeral) meta-algorithm for meta-learning [34]. For the *Optimal Action* variant we assume $\arg\min_{\theta \in \Theta} L(\theta)$ returns $\theta$ minimizing $\mathcal{B}_R(\theta|\phi_R)$ over the set of all minimizers of $L$ over $\Theta$, where $\phi_R$ is some appropriate element of $\Phi$ such as the origin in Euclidean space or the uniform distribution over the simplex.

---

**Data:**

- action space $\Theta \subset \mathbb{R}^d$ with norm $\|\cdot\|$
- function $R : \Theta \mapsto \mathbb{R}$ that is 1-strongly-convex w.r.t. $\|\cdot\|$ and its corresponding Bregman divergence $\mathcal{B}_R$
- class of within-task algorithms $\{\text{TASK}_{\eta,\phi} : \eta > 0, \phi \in \Theta\}$
- meta-update algorithms INIT and SIM
- sequence of loss functions $\{\ell_{t,i} : \Theta \mapsto \mathbb{R}\}_{t \in [T], i \in [m_t]}$ where $\ell_{t,i}$ is $G_{t,i}$-Lipschitz w.r.t. $\|\cdot\|$

**for** $t \in [T]$ **do**

> // set learning rate and initialization using meta-update algorithms
> $D_t = \text{SIM}(\{\ell_{s,i}\}_{s<t,i\in[m_s]})$
> $G_t \leftarrow \sqrt{\frac{1}{m_t}\sum_{i=1}^{m_t} G_{t,i}^2}$
> $\eta_t \leftarrow \frac{D_t}{G_t\sqrt{m_t}}$
> $\phi_t = \text{INIT}(\{\ell_{s,i}\}_{s<t,i\in[m_s]})$
>
> // run within-task algorithm
> **for** $i \in [m_t]$ **do**
>> $\theta_{t,i} \leftarrow \text{TASK}_{\eta_t,\phi_t}(\ell_{t,1},...,\ell_{t,i-1})$
>> suffer loss $\ell_{t,i}(\theta_{t,i})$
>
> // compute meta-update vector $\theta_t$ according to Ephemeral variant
> **case** Optimal Action **do**
>> $\theta_t \leftarrow \arg\min_{\theta\in\Theta}\sum_{i=1}^{m_t}\ell_{t,i}(\theta)$
>
> **case** Last Iterate **do**
>> $\theta_t \leftarrow \text{TASK}_{\eta_t,\phi_t}(\ell_{t,1},...,\ell_{t,m_t})$
>
> **case** Average Iterate **do**
>> $\theta_t \leftarrow \frac{1}{m_t}\sum_{i=1}^{m_t}\theta_{t,i}$

---

**Assumption F.1.** *Assume the data given to Algorithm 3 and define the following quantities:*

- *convenience coefficients $\sigma_t = G_t \sqrt{m_t}$*

- *sequence of update parameters $\{\hat{\theta}_t \in \Theta\}_{t \in [T]}$ with average update $\hat{\phi} = \frac{1}{\sigma_{1:T}} \sum_{t=1}^{T} \sigma_t \hat{\theta}$*

- *a sequence of reference parameters $\{\theta'_t \in \Theta\}_{t \in [T]}$ with average reference parameter $\phi' = \frac{1}{\sigma_{1:T}} \sum_{t=1}^{T} \sigma_t \theta'_t$*

- *a sequence $\{\theta^*_t \in \Theta\}_{t \in [T]}$ of optimal parameters in hindsight*

- *we will say we are in the "Exact" case if $\hat{\theta}_t = \theta'_t = \theta^*_t \; \forall \, t$ and the "Approx" case otherwise*

- *$\kappa \geq 1, \Delta^*_t \geq 0$ s.t. $\sum_{t=1}^{T} \alpha_t \mathcal{B}_R(\theta^*_t || \phi_t) \leq \Delta^*_{1:T} + \kappa \sum_{t=1}^{T} \alpha_t \mathcal{B}_R(\hat{\theta}_t || \phi_t)$ for some $\alpha_t \geq 0$*

- *$\nu \geq 1, \Delta' \geq 0$ s.t. $\sum_{t=1}^{T} \sigma_t \mathcal{B}_R(\hat{\theta}_t || \hat{\phi}) \leq \Delta' + \nu \sum_{t=1}^{T} \sigma_t \mathcal{B}_R(\theta'_t || \phi')$*

- *average deviation $V^2 = \frac{1}{\sigma_{1:T}} \sum_{t=1}^{T} \sigma_t \mathcal{B}_R(\theta'_t || \phi')$ of the reference parameters*

- *action diameter $D^2 = \max\{D^{*2}, \max_{\theta \in \Theta} \mathcal{B}_R(\theta || \phi_1)\}$ in the Exact case or $\max_{\theta, \phi \in \Theta} \mathcal{B}_R(\theta || \phi)$ in the Approx case*

- *constant $C'$ s.t. $\|\theta\| \leq C' \|\theta\|_2 \; \forall \, \theta \in \Theta$ and $\ell_2$-diameter $D' = \max_{\theta, \phi} \|\theta - \phi\|_2$ of $\Theta$*

- *effective action space $\hat{\Theta} = \mathrm{Conv}(\{\hat{\theta}_t\}_{t \in [T]})$ if INIT is FTL or $\Theta$ if INIT is AOGD*

- *upper bound $G'$ on the Lipschitz constants of the functions $\{\mathcal{B}_R(\hat{\theta}_t || \cdot)\}_{t \in [T]}$ over $\hat{\Theta}$*

- *we will say we are in the "Nice" case if $\mathcal{B}_R(\theta || \cdot)$ is 1-strongly-convex and $\beta$-strongly-smooth w.r.t. $\| \cdot \| \; \forall \, \theta \in \Theta$*

- *in the general case INIT is FTL; in the Nice case INIT may instead be AOGD*

- *convenience indicator $\iota = 1_{\mathrm{INIT=FTL}}$*

- *$\mathrm{TASK}_{\eta, \phi} = \mathrm{FTRL}^{(R)}_{\eta, \phi}$ or $\mathrm{OMD}^{(R)}_{\eta, \phi}$*

*We make the following assumptions:*

- *the loss functions $\ell_{t,i}$ are convex $\forall \, t, i$*

- *at $t = 1$ the update algorithm INIT plays $\phi_1 \in \Theta$ satisfying $\max_{\theta \in \Theta} \mathcal{B}_R(\theta || \phi_1) < \infty$*

- *in the Approx case $R$ is $\beta$-strongly-smooth for some $\beta \geq 1$*

### F.1 Average Regret using Fixed Task Similarity

The following theorem does not appear in the main paper but is used in discussion. It shows guarantees for the case when the task-similarity is known in advance and so SIM always returns a constant.

**Theorem F.1.** *Make Assumption F.1 and suppose* SIM *always plays $D_t = \varepsilon$. Then Algorithm 3 has a regret upper-bound of*

$$\bar{U}_M \leq \frac{1}{T} \left( \left( \frac{\kappa D^2}{\varepsilon} + \varepsilon \right) \iota \sigma_1 + \frac{\kappa C}{\varepsilon} \sum_{t=1}^{T} \frac{\sigma_t^2}{\sigma_{1:t}} + \left( \frac{\kappa \nu V^2}{\varepsilon} + \varepsilon \right) \sigma_{1:T} + \frac{\Delta_{1:T}^*}{\varepsilon} + \frac{\kappa \Delta'}{\varepsilon} \right)$$

*for $C = \frac{G'^2}{2}$ in the Nice case or otherwise $C = 2C'D'G'$.*

*Proof.* Let $\{\tilde{\phi}_t\}_{t \in [T]}$ be a "cheating" sequences such that $\tilde{\phi}_t = \phi_t$ on all $t$ except if SIM is FTL and $t = 1$, in which case $\tilde{\phi}_1 = \hat{\theta}_1$. Note that by this definition all upper bounds of $\mathcal{B}_R(\hat{\theta}_t || \phi_t)$ also upper bound $\mathcal{B}_R(\hat{\theta}_t || \tilde{\phi}_t)$. We then use the fact that the actions of FTL at $t > 1$ do not depend on the action at time $t = 1$ to get

$$\bar{U}_M T$$

$$= \sum_{t=1}^{T} \frac{\mathcal{B}_R(\theta_t^* || \phi_t)}{\eta_t} + \eta_t G_t^2 m_t$$

$$= \frac{\Delta_{1:T}^*}{\varepsilon} + \sum_{t=1}^{T} \left( \frac{\kappa \mathcal{B}_R(\hat{\theta}_t || \phi_t)}{\varepsilon} + \varepsilon \right) \sigma_t \qquad \text{(substitute } \eta_t = \frac{D_t}{G_t \sqrt{m_t}} \text{ and } D_t = \varepsilon)$$

$$\leq \left( \frac{\kappa D^2}{\varepsilon} + \varepsilon \right) \iota \sigma_1 + \frac{\Delta_{1:T}^*}{\varepsilon} + \sum_{t=1}^{T} \left( \frac{\kappa \mathcal{B}_R(\hat{\theta}_t || \tilde{\phi}_t)}{\varepsilon} + \varepsilon \right) \sigma_t \qquad \text{(substitute cheating sequence)}$$

$$= \left( \frac{\kappa D^2}{\varepsilon} + \varepsilon \right) \iota \sigma_1 + \frac{\Delta_{1:T}^*}{\varepsilon} + \frac{\kappa}{\varepsilon} \sum_{t=1}^{T} \left( \mathcal{B}_R(\hat{\theta}_t || \tilde{\phi}_t) - \mathcal{B}_R(\hat{\theta}_t || \hat{\phi}) \right) \sigma_t + \sum_{t=1}^{T} \left( \frac{\kappa \mathcal{B}_R(\hat{\theta}_t || \hat{\phi})}{\varepsilon} + \varepsilon \right) \sigma_t$$

$$\leq \left( \frac{\kappa D^2}{\varepsilon} + \varepsilon \right) \iota \sigma_1 + \frac{\Delta_{1:T}^*}{\varepsilon} + \frac{\kappa C}{\varepsilon} \sum_{t=1}^{T} \frac{\sigma_t^2}{\sigma_{1:t}} + \frac{\kappa \Delta'}{\varepsilon}$$

$$+ \sum_{t=1}^{T} \left( \frac{\kappa \nu \mathcal{B}_R(\theta_t' || \phi')}{\varepsilon} + \varepsilon \right) \sigma_t \qquad \text{(Thm. A.2 and Prop. B.1)}$$

$$= \left( \frac{\kappa D^2}{\varepsilon} + \varepsilon \right) \iota \sigma_1 + \frac{\Delta_{1:T}^*}{\varepsilon} + \frac{\kappa C}{\varepsilon} \sum_{t=1}^{T} \frac{\sigma_t^2}{\sigma_{1:t}} + \frac{\kappa \Delta'}{\varepsilon} + \left( \frac{\kappa \nu V^2}{\varepsilon} + \varepsilon \right) \sigma_{1:T}$$

$\square$

## F.2 Average Regret when Learning Task Similarity

**Theorem F.2.** *Make Assumption F.1 and let* SIM *be an algorithm running on the sequence of pairs* $\{\mathcal{B}_R(\hat{\theta}_t||\phi_t), \sigma_t\}_{t\in[T]}$ *and at each time* $t$ *having as output the action of an OCO algorithm on the function sequence* $\{\ell_t(x) = (\mathcal{B}_R(\hat{\theta}_t||\phi_t)/x + x)\sigma_t\}_{t\in[T]}$. *Let* $\mathbf{R}_T$ *be the associated regret of this algorithm and suppose it has a parameter* $\varepsilon > 0$ *controlling the minimum action taken. For simplicity assume that at time* $t = 1$ SIM *plays* $D_1$ *s.t.* $\frac{1}{2}(\max_{\theta\in\Theta}\sqrt{\mathcal{B}_R(\theta||\phi_1)} + \varepsilon) \leq D_1 \leq \max_{\theta\in\Theta}\sqrt{\mathcal{B}_R(\theta||\phi_1)} + \varepsilon$. *Then Algorithm 3 has a regret upper-bound of*

$$\bar{\mathbf{U}}_M \leq \frac{1}{T}\left((2\kappa D + \varepsilon)\iota\sigma_1 + \kappa\,\mathbf{R}_T + \frac{\kappa C}{V}\sum_{t=1}^{T}\frac{\sigma_t^2}{\sigma_{1:t}} + \kappa(\nu+1)V\sigma_{1:T} + \frac{\Delta_{1:T}^*}{\varepsilon} + \frac{\kappa\Delta'}{V}\right)$$

*for* $C = \frac{G'^2}{2}$ *in the Nice case or otherwise* $C = 2C'D'G'$.

*Proof.* Let $\{\tilde{\phi}_t\}_{t\in[T]}$ be "cheating" sequence such that $\tilde{\phi}_t = \phi_t$ on all $t$ except if SIM is FTL and $t = 1$, in which case $\tilde{\phi}_1 = \hat{\theta}_1$. Note that by this definition all upper bounds of $\mathcal{B}_R(\hat{\theta}_t||\phi_t)$ also upper bound $\mathcal{B}_R(\hat{\theta}_t||\tilde{\phi}_t)$. We then have

$$\bar{\mathbf{U}}_M\,T = \sum_{t=1}^{T}\frac{\mathcal{B}_R(\theta_t^*||\phi_t)}{\eta_t} + \eta_t G_t^2 m_t$$

$$= \frac{\Delta_{1:T}^*}{\varepsilon} + \sum_{t=1}^{T}\left(\frac{\kappa\mathcal{B}_R(\hat{\theta}_t||\phi_t)}{D_t} + D_t\right)\sigma_t \qquad \text{(substitute } \eta_t = \frac{D_t}{G_t\sqrt{m_t}} \text{ and } D_t \geq \varepsilon\text{)}$$

$$\leq \left(\frac{\kappa\mathcal{B}_R(\hat{\theta}_t||\phi_t)}{D_1} + D_1\right)\iota\sigma_1 + \frac{\Delta_{1:T}^*}{\varepsilon}$$

$$\quad + \sum_{t=1}^{T}\left(\frac{\kappa\mathcal{B}_R(\hat{\theta}_t||\tilde{\phi}_t)}{D_t} + D_t\right)\sigma_t \qquad \text{(substitute cheating sequences)}$$

$$\leq ((\kappa+1)D + \varepsilon)\iota\sigma_1 + \frac{\Delta_{1:T}^*}{\varepsilon} + \kappa\,\mathbf{R}_T + \kappa\sum_{t=1}^{T}\left(\frac{\mathcal{B}_R(\hat{\theta}_t||\tilde{\phi}_t)}{V} + V\right)\sigma_t$$

$$\leq (2\kappa D + \varepsilon)\iota\sigma_1 + \frac{\Delta_{1:T}^*}{\varepsilon} + \kappa\,\mathbf{R}_T + \frac{\kappa C}{V}\sum_{t=1}^{T}\frac{\sigma_t^2}{\sigma_{1:t}}$$

$$\quad + \kappa\sum_{t=1}^{T}\left(\frac{\mathcal{B}_R(\hat{\theta}_t||\hat{\phi})}{V} + V\right)\sigma_t \qquad \text{(Thm. A.2 and Prop. B.1)}$$

$$\leq (2\kappa D + \varepsilon)\iota\sigma_1 + \frac{\Delta_{1:T}^*}{\varepsilon} + \kappa\,\mathbf{R}_T + \frac{\kappa C}{V}\sum_{t=1}^{T}\frac{\sigma_t^2}{\sigma_{1:t}} + \frac{\kappa\Delta'}{V} + \kappa\sum_{t=1}^{T}\left(\frac{\nu\mathcal{B}_R(\theta_t'||\phi')}{V} + V\right)\sigma_t$$

$$\leq (2\kappa D + \varepsilon)\iota\sigma_1 + \frac{\Delta_{1:T}^*}{\varepsilon} + \kappa\,\mathbf{R}_T + \frac{\kappa C}{V}\sum_{t=1}^{T}\frac{\sigma_t^2}{\sigma_{1:t}} + \frac{\kappa\Delta'}{V} + \kappa(\nu+1)V\sigma_{1:T}$$

$$\square$$

### F.3 Statistical Task-Similarity under Quadratic Growth

In this section we relate our task-similarity measure to that of Denevi et al. [19] under $\alpha$-QG.

**Proposition F.1.** *For some distribution $\mathcal{P} \sim \mathcal{Q}$ over losses $\ell : \Theta \mapsto \mathbb{R}_+$ let $\theta_{\mathcal{P}}^* = \arg\min_{\theta \in \Theta} \ell_{\mathcal{P}}(\theta)$ and $\hat{\theta}_m = \arg\min_{\theta \in \Theta} \sum_{i=1}^{m} \ell_i(\theta)$ for $m$ i.i.d. samples $\ell_i \sim \mathcal{P}$. Define task-similarity measures $V^2 = \min_{\phi \in \Theta} \mathbb{E}_{\mathcal{P} \sim \mathcal{Q}} \|\theta_{\mathcal{P}}^* - \phi\|_2^2$ and $\hat{V}_m^2 = \min_{\phi \in \Theta} \mathbb{E}_{\mathcal{P} \sim \mathcal{Q}} \mathbb{E}_{\mathcal{P}^m} \|\hat{\theta}_m - \phi\|_2^2$. If both $\ell_{\mathcal{P}}$ and $\frac{1}{m} \sum_{i=1}^{m} \ell_i$ are G-Lipschitz and $\alpha$-QG a.s. then we have*

$$V^2 \le 2\hat{V}_m^2 + \frac{16G^2}{\alpha^2 m} \qquad \text{and} \qquad \hat{V}_m^2 \le 2V^2 + \frac{16G^2}{\alpha^2 m}$$

*Proof.* Following the argument of Shalev-Shwartz et al. [49, Theorem 2] but applying $\alpha$-QG instead of strong-convexity in Equation 8, which holds by definition of $\alpha$-QG, we obtain

$$\mathbb{E}_{\mathcal{P}^m} \left( \ell_{\mathcal{P}}(\hat{\theta}_m) - \ell_{\mathcal{P}}(\theta_{\mathcal{P}}^*) \right) \le \frac{4G^2}{\alpha m}$$

Then for $\phi^* = \arg\min \mathbb{E}_{\mathcal{P} \sim \mathcal{Q}} \|\theta_{\mathcal{P}}^* - \phi\|_2^2$ and $\phi_m^* = \arg\min_{\phi \in \Theta} \mathbb{E}_{\mathcal{P} \sim \mathcal{Q}} \mathbb{E}_{\mathcal{P}^m} \|\hat{\theta}_m - \phi\|_2^2$ we have by these definitions, the triangle inequality, Jensen's inequality, $\alpha$-QG of $\frac{1}{m} \sum_{i=1}^{m} \ell_i$, and the above inequality we have

$$\hat{V}_m^2 = \mathbb{E}_{\mathcal{P} \sim \mathcal{Q}} \mathbb{E}_{\mathcal{P}^m} \|\hat{\theta}_m - \phi_m^*\|_2^2 \le \mathbb{E}_{\mathcal{P} \sim \mathcal{Q}} \mathbb{E}_{\mathcal{P}^m} \|\hat{\theta}_m - \phi^*\|_2^2$$

$$\le 2 \mathbb{E}_{\mathcal{P} \sim \mathcal{Q}} \mathbb{E}_{\mathcal{P}^m} \left( \|\hat{\theta}_m - \theta_{\mathcal{P}}^*\|_2^2 + \|\theta_{\mathcal{P}}^* - \phi^*\|_2^2 \right)$$

$$\le \frac{4}{\alpha} \mathbb{E}_{\mathcal{P} \sim \mathcal{Q}} \mathbb{E}_{\mathcal{P}^m} \left( \ell_{\mathcal{P}}(\hat{\theta}_m) - \ell_{\mathcal{P}}(\theta_{\mathcal{P}}^*) \right) + 2V^2$$

$$\le \frac{16G^2}{\alpha^2 m} + 2V^2$$

Similarly,

$$V^2 = \mathbb{E}_{\mathcal{P} \sim \mathcal{Q}} \mathbb{E}_{\mathcal{P}^m} \|\theta_{\mathcal{P}} - \phi^*\|_2^2 \le \mathbb{E}_{\mathcal{P} \sim \mathcal{Q}} \mathbb{E}_{\mathcal{P}^m} \|\theta_{\mathcal{P}} - \phi_m^*\|_2^2$$

$$\le 2 \mathbb{E}_{\mathcal{P} \sim \mathcal{Q}} \mathbb{E}_{\mathcal{P}^m} \left( \|\hat{\theta}_m - \theta_{\mathcal{P}}^*\|_2^2 + \|\hat{\theta}_m - \phi_m^*\|_2^2 \right)$$

$$\le \frac{4}{\alpha} \mathbb{E}_{\mathcal{P} \sim \mathcal{Q}} \mathbb{E}_{\mathcal{P}^m} \left( \ell_{\mathcal{P}}(\hat{\theta}_m) - \ell_{\mathcal{P}}(\theta_{\mathcal{P}}^*) \right) + 2V^2$$

$$\le \frac{16G^2}{\alpha^2 m} + 2V^2$$

$\square$

# G Experimental Details

Code is available at `https://github.com/mkhodak/ARUBA`.

## G.1 Reptile

For our Reptile experiments we use the code and default settings provided by Nichol et al. [44], except we tune the learning rate, which for ARUBA corresponds to $\varepsilon/\zeta$, and the coefficient $c$ in ARUBA++. In addition to the the parameters listed in the above tables, we set $\zeta = p = 1.0$ for all experiments. All evaluations are averages of three runs.

| Omniglot | 1-shot | | | | 5-shot | | | |
|---|---|---|---|---|---|---|---|---|
| **5-way** | evaluation setting | | hyperparameters | | evaluation setting | | hyperparameters | |
| | regular | transductive | $\eta = \frac{\varepsilon}{\zeta}$ | $c$ | regular | transductive | $\eta = \frac{\varepsilon}{\zeta}$ | $c$ |
| MAML (1) [23] | | $98.3 \pm 0.5$ | | | | $99.2 \pm 0.2$ | | |
| Reptile [44] | $95.39 \pm 0.09$ | $97.68 \pm 0.04$ | $1e-3$ | | $98.90 \pm 0.10$ | $99.48 \pm 0.06$ | $1e-3$ | |
| ARUBA | $94.57 \pm 1.04$ | $97.44 \pm 0.32$ | $1e-1$ | | $98.64 \pm 0.04$ | $99.29 \pm 0.07$ | $1e-2$ | |
| ARUBA++ | $94.80 \pm 1.10$ | $97.58 \pm 0.13$ | $1e-1$ | $10^3$ | $98.93 \pm 0.13$ | $99.46 \pm 0.02$ | $1e-2$ | $10^3$ |
| MAML (2) | | $98.7 \pm 0.4$ | | | | $99.9 \pm 0.1$ | | |
| Meta-SGD [38] | | $99.53 \pm 0.26$ | | | | $99.93 \pm 0.09$ | | |

Figure 4: Final learning rate $\eta_T$ across the layers of a convolutional network trained on 1-shot 5-way Omniglot (top) and 5-shot 5-way Omniglot (bottom) using Algorithm 2 applied to Reptile.

| Omniglot | 1-shot | | | | 5-shot | | | |
|---|---|---|---|---|---|---|---|---|
| 20-way | evaluation setting | | hyperparameters | | evaluation setting | | hyperparameters | |
| | regular | transductive | $\eta = \frac{\varepsilon}{\zeta}$ | $c$ | regular | transductive | $\eta = \frac{\varepsilon}{\zeta}$ | $c$ |
| MAML (1) [23] | | $95.8 \pm 0.3$ | | | | $98.9 \pm 0.2$ | | |
| Reptile [44] | $88.14 \pm 0.15$ | $89.43 \pm 0.14$ | $5e-4$ | | $96.65 \pm 0.33$ | $97.12 \pm 0.32$ | $5e-4$ | |
| ARUBA | $85.61 \pm 0.25$ | $86.67 \pm 0.17$ | $5e-3$ | | $96.02 \pm 0.12$ | $96.61 \pm 0.13$ | $5e-3$ | |
| ARUBA++ | $88.38 \pm 0.24$ | $89.66 \pm 0.3$ | $5e-3$ | $10^3$ | $96.99 \pm 0.35$ | $97.49 \pm 0.28$ | $5e-3$ | 10 |
| MAML (2) | | $95.8 \pm 0.3$ | | | | $98.9 \pm 0.2$ | | |
| Meta-SGD [38] | | $95.93 \pm 0.38$ | | | | $98.97 \pm 0.19$ | | |

Figure 5: Final learning rate $\eta_T$ across the layers of a convolutional network trained on 1-shot 20-way Omniglot (top) and 5-shot 20-way Omniglot (bottom) using Algorithm 2 applied to Reptile.

| Mini-ImageNet | 1-shot | | | | 5-shot | | | |
|---|---|---|---|---|---|---|---|---|
| **5-way** | evaluation setting | | hyperparameters | | evaluation setting | | hyperparameters | |
| | regular | transductive | $\eta = \frac{\varepsilon}{\zeta}$ | $c$ | regular | transductive | $\eta = \frac{\varepsilon}{\zeta}$ | $c$ |
| MAML (1) [23] | | $48.07 \pm 1.75$ | | | | $63.15 \pm 0.91$ | | |
| Reptile [44] | $47.07 \pm 0.26$ | $49.97 \pm 0.32$ | $1e-3$ | | $62.74 \pm 0.37$ | $65.99 \pm 0.58$ | $1e-3$ | |
| ARUBA | $47.01 \pm 0.37$ | $50.73 \pm 0.32$ | $5e-3$ | | $62.35 \pm 0.25$ | $65.69 \pm 0.61$ | $5e-3$ | |
| ARUBA++ | $47.25 \pm 0.61$ | $50.35 \pm 0.74$ | $5e-3$ | $10$ | $62.69 \pm 0.57$ | $65.89 \pm 0.34$ | $5e-3$ | $10^{-1}$ |
| MAML (2) | | $48.70 \pm 1.84$ | | | | $63.11 \pm 0.92$ | | |
| Meta-SGD [38] | | $50.47 \pm 1.87$ | | | | $64.03 \pm 0.94$ | | |

Figure 6: Final learning rate $\eta_T$ across the layers of a convolutional network trained on 1-shot 5-way Mini-ImageNet (top) and 5-shot 5-way Mini-ImageNet (bottom) using Algorithm 2 applied to Reptile.

## G.2  FedAvg

For FedAvg we train a 2-layer stacked LSTM model with 256 hidden units, 8-dimensional trained character embeddings, with a maximum input string size of 80 characters; these settings are used to match those of McMahan et al. [41]. Similarly, we take their approach of only removing those actors from the Shakespeare dataset with fewer than two lines and split each user temporally into train/test sets with a training fraction of 0.8. Unlike McMahan et al. [41], we also split the users into meta-training and meta-testing sets, also with a fraction of 0.8, in order to evaluate meta-test performance. We run both algorithms for 500 rounds with a batch of 10 users per round and a within-task batch-size of 10, as in Caldas et al. [12]. For unmodified FedAvg we found that an initial learning rate of $\eta = 1.0$ worked well – this is similar to those reported in McMahan et al. [41] and Caldas et al. [12] – and for the tuned variant we found that a multiplicative decay of 0.99. At meta-test-time we tuned the refinement learning rate over $\{10^{-3}, 10^{-2}, 10^{-1}\}$. For ARUBA and its isotropic variant we set $\varepsilon = \zeta = 0.05$ and $p = 1.0$, so that $\eta = \varepsilon/\zeta = 1.0$ in our setting as well.

Figure 7: Final learning rate $\eta_T$ across the layers of an LSTM trained for next-character prediction on the Shakespeare dataset using Algorithm 2 applied to FedAvg.