[Reviews · NeurIPS 2019]

Reviewer 1



I would like to thank the authors for very clearly written paper. I have a few small points which could potentially improve the write-up. Page 3, 1. Generality, “the regret of OGD on m_t convex G-Lipschitz losses has a well-known upper bound of” It may be useful to have the reference to this well-known upper bound.. “The niceness of \hat{R_t(x_t)} makes the analysis tractable” May be the authors mean here the case of the OGD? Would the form of \hat{R_t(x_t)} be nice in general? On the line 108 -> 109, it’s not clear how \hat{\bar{R_{T}} \leq o(T) + min_x 1/T sum_t \hat{R_t} In experimental section, authors test their optimization method against Reptile with Adam. I think another sensible baseline would be Reptile+Meta-SGD, which makes the learning rate learnable, thus making it a more fair comparison to ARUBA which uses the adaptive learning rate. For FedAvg case, it would be interesting to see whether the improvement comes from the meta-learning treatment of the federated learning problem (where we optimize for the initialization) or the ARUBA algorithm itself (e.g. the fact that the learning rate is adaptable).

Reviewer 2



The paper presents a new way to analyze multi-task learning algorithms in the online setting by analysing average regret-upper-bound. The authors introduce a two-level algorithm that runs a mirror descent algorithm for each task and adjusts the parameters of the mirror descent before each new task by running another online learning algorithm on the level of tasks. This task-level algorithm is designed to minimize theoretical regret bounds for each task. The authors prove a general bound on the performance of the presented algorithm and show its corollaries for settings with static and dynamic comparators. A version of the algorithm for different task similarity measures is also presented, as well as, an online-to-batch conversion result for learning-to-learn setting with i.i.d. tasks. In the end, an empirical evaluation is presented that shows an improved performance when ARUBA applied to few-shot classification and federated learning settings. Pros: * A new approach for multi-task learning by analysing average regret-upper-bound * A general performance bound (Theorem 1) that can be applied to a variety of settings * Interesting decomposition of (2) into two online learning problem that can be studied separately * Proof in the static environment case for non-convex Bregman divergencies * Interesting extension with inter-task geometry (Section 4) Cons: * Complicated and overloaded notations: too many versions of regret and bounds with very similar symbols, the sequence of \psi_t for dynamic regret is not defined for notations of Theorem 3.1. * The paper is missing a detailed discussion/examples of the behaviour of V_\Psi, which makes it's hard to judge the sensibility of the proven bounds. AFTER FEEDBACK: I had only minor comments and authors addressed them. My evaluation stays the same and I recommend the paper for acceptance.

Reviewer 3



This paper proposed a new theoretical framework for analysing the regret of gradient based meta learning algorithms and derived a few variants of algorithms to learn the initialisation and learning rate. It provides rigorous analysis about the online regret and how it depends on the similarity of tasks, number of gradient descent steps, number of tasks, etc. It provides a nice tool to select the update rule for the model initialisation and choosing the step size. It also provides an approach to analyse the bound and convergence rate of the excess transfer risk. However, I'm not familiar with the literature on online learning and its application on the meta-learning setup. Therefore, I'm not able to comment on the advantage of the proposed method compared to existing approaches. In the experiment section, the improvement of ARUBA over Adam is relative small for the reptile algorithm while the improvement on the FedAvg algorithm is a lot more significant. Could the author comment what factor leads to the big difference between these two experiments?

[Author Response · NeurIPS 2019]

We thank all the reviewers for their thoughtful comments and positive feedback. We will first address two major points
raised by the reviewers and then answer individual questions.

**Major Points:**

1. **(Reviewers 1 and 3)** - "Providing more empirical results and refined baselines will improve the experiments section
and be useful to study the strength of this algorithm in practice." (paraphrase)
- We agree and will have more experiments in revision. Specifically, we plan to include few-shot learning evaluations
on MiniImageNet [47] and detailed numerical results for both the meta-learning and federated learning settings.
Furthermore, following the suggestion of Reviewer 1, we will investigate a comparison with Reptile+Meta-SGD
[39], which would also learn a per-coordinate learning rate. One caveat here is that Meta-SGD was designed for a
MAML-like approach of only taking one gradient step within-task, whereas Reptile takes many iterations; since
Meta-SGD uses higher-order differentiation, repeated application may slow it down. Note that we do discuss
(MAML+) Meta-SGD briefly in lines 305 -> 307.

2. **(Reviewer 2)** - "The paper is missing a detailed discussion/examples of the behaviour of $V_\Psi$, which makes it hard to
judge the sensibility of the proven bounds."
- We agree that examples of $V_\Psi$ are needed to understand the results, as it is the main measure of task-similarity
and if $V_\Psi$ is small then so is the average regret. For the case of a fixed comparator (Theorem 3.2), we do give a
simple example in lines 136 -> 137; here $V_\Psi$ is proportional to the empirical standard deviation of the optimal
task-parameters, so if they are close (i.e. tasks are similar) then $V_\Psi$ is small. As we will describe in more detail in
revision, the case when the comparator is varying is similar, as $V_\Psi$ is now proportional to the average deviation of
the optimal task-parameters from a shifting sequence of vectors. For example, if we see one task every day for
year, and $\Psi$ is a sequence that fixes a single comparator for each month, then $V_\Psi^2$ is roughly the average over the
months of the empirical variance of each month's thirty or so optimal task-parameters from that month's fixed
comparator. This can be very small if the variation between tasks is well-described by a seasonal trend.

**Reviewer #1:**

1. "It would be useful to include a reference for the regret of OGD." (paraphrase)
- We agree and will add a pointer to the Shalev-Shwartz survey [49, Theorem 2.15 for $R(w) = \frac{1}{2\eta}\|w - \phi\|_2^2$ ].

2. "Is the form of the regret-upper-bound $\hat{R}_t(x_t)$ nice in cases more general than just that of OGD?" (paraphrase)
- Yes! This is a main reason we expect this framework to be broadly applicable. In our paper, we show that several
results (specifically Theorems 3.1 and 3.2) hold for any algorithm in the OMD/FTRL family, which includes not
just OGD but also other classical methods such as exponentiated gradient/multiplicative weights. Even more
generally, regret guarantees often include terms that depend on some measure of distance from an initial state,
which are often amenable to study (e.g. because norms are convex). We will elaborate on this in the revision.

3. "On the line 108 -> 109, it's not clear how $\bar{\hat{R}}_T \le o(T) + \min_x \frac{1}{T}\sum_t \hat{R}_t$ ."
- We believe this asking how the right-hand side goes to zero. This is a typo - the first term should be $o_T(1)$, i.e.
sub-constant, not sub-linear. Similarly, the last term in the statement of Theorem 3.1 should be divided by $T$ (the
expressions in the proof are correct as-is). We apologize for both errors and will correct them in revision.

4. "For FedAvg case, does the improvement comes from the meta-learning treatment (where we optimize for the
initialization) or the ARUBA algorithm itself (e.g. the fact that the learning rate is adaptable)?" (paraphrase)
- In fact to get FedAvg+ARUBA we do not modify FedAvg except to adapt the learning rate - the global model is
still learned in the same way. This is possible because FedAvg is equivalent to Reptile with the outer-loop update
coefficient set to 1.0. So the improvement is indeed coming from the adaptivity.

**Reviewer #2:**

1. "Complicated and overloaded notations: too many versions of regret and bounds with very similar symbols, the
sequence of $\psi_t$ for dynamic regret is not defined for notations of Theorem 3.1."
- We will make sure that the mathematical presentation is as clean as possible in the revision. One way of reducing
the many variations on the regret notations is to depend less on accents and represent regret-upper-bounds by a
different capital letter (e.g. $\mathbf{U}$) to better distinguish from regret terms ($\mathbf{R}$). As for the sequence of $\psi_t$, this can be
any arbitrary sequence of vectors in the action space $\Theta$.

**Reviewer #3:**

1. "What factor leads to the big experimental difference between meta-learning, where the improvement over Adam is
relatively small, and federated learning, where the improvement over FedAvg is a lot more significant?" (paraphrase)
- Our explanation, which we will add in revision, is based on the nature of data. Whereas standard evaluation
datasets in few-shot learning consist of tasks with identical amounts of i.i.d. data, the Shakespeare benchmark
we use for federated learning has tasks with highly variable amounts of data (two different roles can have very
different numbers of lines) and data that is not i.i.d. (lines are not shuffled but split in their order of appearance
in the play). Our method may be better able to handle such data, for example by tempering noisy directions in
low-data tasks by learning which directions are important based on the distance traveled in high-data tasks.

[Meta-Review · NeurIPS 2019]

This paper addresses a new method for analyzing gradient-based meta-learning in the online setting, where the average regret-upper-bound analysis was presented. All of reviewers agree that a new theoretical framework in this paper has valuable contributions that can be applied to a variety of setting. While there were no further comments in the discussion period, I believe that the paper is deserved to be presented at NeurIPS.